# Synthetic Series-Symbol Data Generation for Time Series Foundation Models

**Wenxuan Wang**
School of Telecommunications Engineering
Xidian University
whenxuan@ieee.org

**Kai Wu**[*]
School of Artificial Intelligence
Xidian Univeristy
kwu@xidian.edu.cn

**Yujian Betterest Li**
School of Artificial Intelligence
Xidian University
bebetterest@outlook.com

**Dan Wang**[*]
School of Telecommunications Engineering
Xidian University
danwang@xidian.edu.cn

**Xiaoyu Zhang**
School of Cyber Engineering
Xidian University
xiaoyuzhang@xidian.edu.cn

## Abstract

Foundation models for time series analysis (TSA) have attracted significant attention. However, challenges such as training data scarcity and imbalance continue to hinder their development. Inspired by complex dynamic system theories, we design a series-symbol data generation mechanism, enabling the unrestricted creation of high-quality time series data paired with corresponding symbolic expressions. To leverage series-symbol data pairs with strong correlations, we develop `SymTime`, a pre-trained foundation model for enhancing time series representation using symbolic information. `SymTime` demonstrates competitive performance across five major TSA tasks when fine-tunes with downstream tasks, rivaling foundation models pre-trained on real-world datasets. This approach underscores the potential of series-symbol data generation and pretraining mechanisms in overcoming data scarcity and enhancing task performance. The code is available at https://github.com/wwhenxuan/SymTime.

## 1 Introduction

In recent years, with the rapid advancement of deep learning, foundation models for time series analysis (TSA) have garnered widespread attention due to their superior generalization capabilities, scalability and advantages in few-shot learning [1, 2]. Coupled with issues of data privacy [3, 4], existing time series datasets are smaller compared to those in the fields of computer vision (CV) and natural language processing (NLP). Besides, current large-scale time series datasets face significant data imbalance issues, with certain types such as finance and healthcare still being relatively scarce (see Appendix B.4). According to scaling laws [5], this can lead to performance bias in the time series foundation models, reducing their generalization capabilities on out-of-distribution data [6, 7].

To mitigate the issue of training data scarcity and imbalance, this paper, starting from Takens' theorem [8, 9], posits that time series are representations of complex dynamical systems [10, 11, 12]. Based on

---

[*]Corresponding author

symbolic dynamics [13], complex systems can be expressed abstractly using mathematical symbols and formulas [14], with ordinary differential equations (ODE) and partial differential equations (PDE) being the most common methods for modeling complex systems [15, 16]. In an ideal scenario, continuously constructing diverse symbolic expressions allows us to cover a broader range of complex dynamical systems. As a result, the time series generated from these symbolic expressions exhibit rich and varied properties. To this end, we provide a series-symbol ($S^2$) dual-modality data generation mechanism. Simulation experiments demonstrate that this approach effectively mitigates the problem of training data scarcity. To encapsulate our work, the contributions are as follows:

- **Addressing Data Scarcity:** Our approach overcomes the challenge of limited training data when building foundation models. The observation that the size of the $S^2$ dataset directly correlates with model performance on downstream tasks validates this point.

- **Introducing SymTime:** We present `SymTime`, a scalable and efficient foundation model for time series analysis that leverages symbolic information to enhance representations. Pretrained on the constructed $S^2$ dataset, `SymTime` offers broader task generality compared to existing foundation models that are typically limited to zero-shot forecasting.

## 2 Related Work

Pre-trained foundation models (PTFMs) [17, 18, 19, 20, 21] have been demonstrated to adapt to a variety of downstream tasks after fine-tuning on specific datasets, exhibiting excellent generalization and scalability [22, 23]. Inspired by this, recent years have seen significant progress in PTFMs for TSA [24, 25], with the emergence of various pre-training methods. Moirai, through masked time series modeling (MTM) and reconstruction [26, 27], has been pre-trained on large datasets ($27B$), yielding a universal forecasting model with zero-shot advantages [28]. Timer, after generative pre-training on large datasets ($1B$), has performed well in forecasting [29]. TimeGPT trained a encoder-decoder Transformer with $100B$ data [30]. COMET, using multi-level contrastive learning on a large ECG dataset, has obtained a medical time series PTFMs with few-shot advantages [4].

As discussed in Appendix B.4, these baseline models still face challenges related to data scarcity and data imbalance. In the next section, we introduce the proposed data generation mechanism and the corresponding dual-modality foundation model designed to mitigate these issues. The review of other topics can be found in Appendix D.

## 3 Main Methods

**Definition 1 Time Series Foundation Model.** *It is a deep neural network pre-trained in a self-supervised or unsupervised manner on large-scale, diverse time series data. By learning generalizable time series representations, it can then rapidly adapt—via few-shot or transfer learning—to efficiently solve a wide range of downstream time series tasks.*

**Theorem 1 Takens' Theorem.** *This theorem demonstrates that through phase space reconstruction [31, 32, 16, 33], a univariate time series, as a low-dimensional projection of a high-dimensional complex system, can completely preserve the dynamic topology of the original system, thereby forming an effective external representation of the complex system [8, 9, 34, 35, 36, 37, 38, 39].*

**Theorem 2 Symbolic Dynamics.** *This theory encodes the evolution of continuous systems into finite symbolic expressions by discretizing the state space of complex systems, establishing an isomorphic relationship between symbolic expressions and system behaviors [13, 41, 42, 43, 44]. This means that any complex system can be modeled using symbolic expressions [45, 46].*

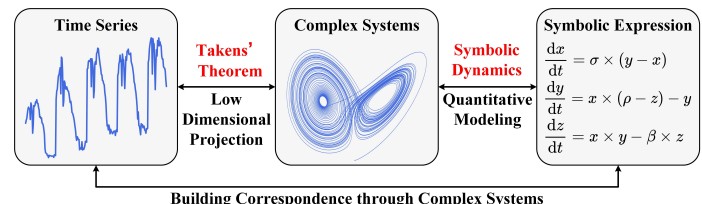

Figure 1: The connection between time series and symbolic expressions (taking the Lorentz system as an example) [40].

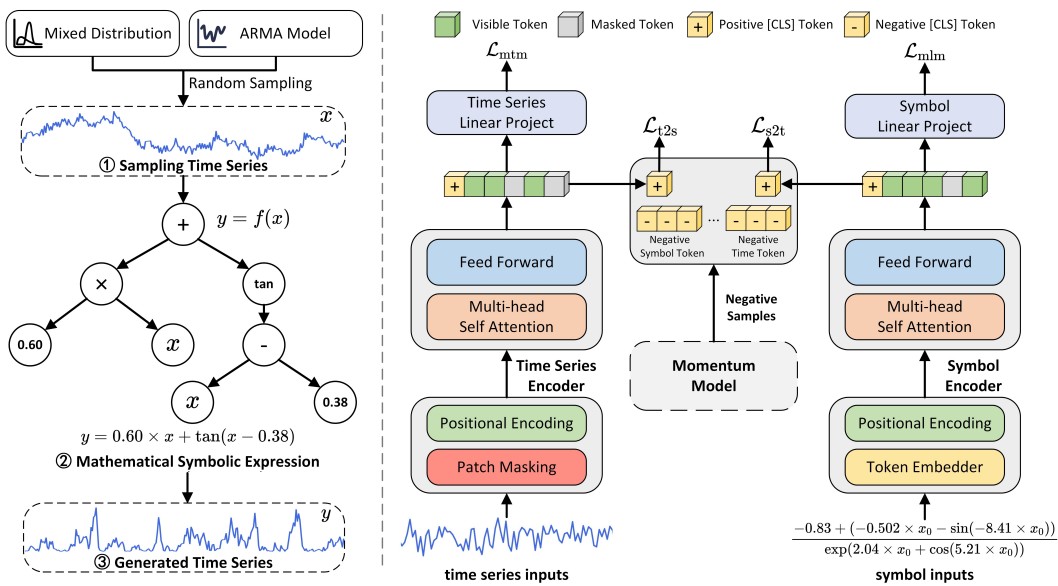

Figure 2: $S^2$ dataset generation mechanism (**left**) and `SymTime` network architecture (**right**).

As show in Figure 1, the two aforementioned theorems, using complex dynamical systems as a conceptual bridge, fundamentally connect time series and symbolic expressions. They provide rigorous theoretical support for the semantic correspondence between temporal patterns and symbolic representations. Compared with previous time series data generation methods [47, 22], the $S^2$ data generation mechanism proposed in this paper is more in line with the nature of time series generation.

## 3.1 Series-Symbol ($S^2$) Dataset Generation

The pre-training of `SymTime` relies on a large synthetic series-symbol ($S^2$) dataset[2]. The specific generation process is shown in Figure 2 (**left**). Firstly, we construct a multivariate input-output symbolic expression $f(\cdot)$ through random sampling [48]. Then, we use the randomly generated sampling series $X \in \mathbb{R}^{M \times L}$ to forward propagate through the symbolic expression to obtain the generated series $Y = f(X) \in \mathbb{R}^{N \times L}$, where $N$ and $M$ represent the dimensions of the input and output series respectively, and $L$ is the length of the series. The mathematical symbols and their explanations in this section are shown in Table 9. In Appendix B.3, we present an analysis of the statistical characterization of the $S^2$ data. In Appendix B.7, we demonstrate that the time complexity of generating the $S^2$ data scales linearly with the series length $L$, approximating $\mathcal{O}(L)$.

### 3.1.1 Sampling of Functions

Mathematical expressions can usually be represented using a tree structure, where constants and variables are the root nodes, binary operators are nodes with two children, and unary operators are nodes with one child. Therefore, we (1) build a binary tree over input variables using binary operators, (2) randomly insert constants and variables into the tree, and (3) add unary operator and perform affine transformation.

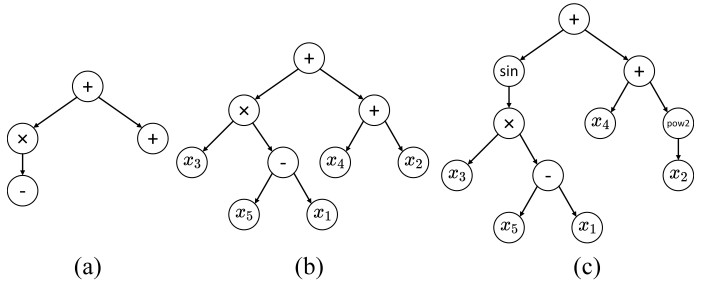

Figure 3: The process of building a binary tree when sampling symbolic expressions. (a) tree construction; (b) variable assignment to leaf nodes; (c) unary operator insertion.

The three key steps are illustrated in Figure 3.

---

[2]The code for $S^2$ data generation is available at `https://github.com/wwhenxuan/S2Generator`.

**Input/Output Dimension Selection.** Instead of randomly sampling input/output dimensions $M \sim \mathcal{U}(1, M_{\max})$ and $N \sim \mathcal{U}(1, N_{\max})$ as in prior work [12, 14, 49], we exhaustively traverse $M \in [1, M_{\max}], N \in [1, N_{\max}]$ (with $M_{\max} = 6$, $N_{\max} = 12$) to fully cover multivariate time series representations. An input dimension $M$ defines $M$ variable nodes $(x_1, \ldots, x_M)$, while the output dimension yields $N$ generated series $y_i = f_i(x_1, \ldots, x_M), i = 1, \ldots, N$ [48].

**Binary Operator Selection.** We sample the number of binary operators $b \sim \mathcal{U}(b_{\min}, b_{\max})$ to define the root nodes of the expression tree. Each node's operator is then drawn uniformly from $\mathcal{U}\{+, -, \times\}$, enhancing the diversity and complexity of the generated expressions [14, 12, 45, 46].

**Tree Construction and Leaf Assignment.** We randomly combine the $b$ binary operators to construct a binary tree to form the basic framework of the mathematical expressions (Figure 3a). Then, we randomly select $m$ variables from the $M$ set of variables $[x_1, \ldots, x_M]$ and insert them into the symbolic expression consisting of binary operators, where $m \sim \mathcal{U}(1, M)$ (Figure 3b). If the inserted expression does not form a full binary tree, a random constant node is added to make it full.

**Unary Operator Insertion.** After inserting the leaf nodes to form a complete binary tree, we select the number of unary operators $u$ from $\mathcal{U}(u_{\min}, u_{\max})$ and insert unary operators at random positions in the binary tree. The available unary operators include $\{\text{inv}, \text{abs}, \text{pow2}, \text{pow3}, \text{sqrt}, \text{sin}, \text{cos}, \text{tan}, \text{arctan}, \text{log}, \text{exp}\}$. This process is shown in Figure 3c. In Appendix B.8 we discussed the choice of unary operators.

**Affine Transformation.** To further diversify the symbolic expressions, we perform random affine transformations on each random variable $x_d$ and unary operator $u_d$ in the binary tree. Specifically, we replace $x_d$ and $u_d$ with $ax_d + b$ and $au_d + b$, respectively, where $a$ and $b$ are random constants [12, 14]. For example, $x_1 \to ax_1 + b$ and $\tan(\cdot) \to a\tan(\cdot) + b$ with contants $a, b$ sampled randomly.

### 3.1.2 Generating Inputs and Outputs Series

After obtaining symbolic expressions $f_i$, we sample $X \in \mathbb{R}^{M \times L}$ from mixed distributions [12, 14, 48] and random-parameter ARMA$(p, q)$ processes [50, 51], then compute $Y \in \mathbb{R}^{N \times L}, y_i = f_i(X)$. The ARMA$(p, q)$ model consists of moving average (MA) and autoregressive (AR) processes [52], which can be expressed as:

$$Y_t = \phi_1 Y_{t-1} + \phi_2 Y_{t-2} + \cdots + \phi_p Y_{t-p} + e_t - \theta_1 e_{t-1} - \theta_2 e_{t-2} - \cdots - \theta_q e_{t-q}, \qquad (1)$$

where $p$ and $q$ represent the orders of the AR and MA models, respectively, $\phi_p$ and $\theta_q$ are the parameters of the AR and MA processes [51], and $e_t \sim \mathcal{N}(0, 1)$ denotes the observed white noise sequence. Since ARMA possess both the temporal correlation of the AR process and the randomness of the MA process, series obtained from mixed distributions and ARMA sampling better reflect the characteristics of time series.

**Sampling Strategy.** Each input series $X \in \mathbb{R}^{M \times L}$ is drawn either from a mixture of $k \sim \mathcal{U}(1, k_{\max})$ distributions (with weights $w_j \sim \mathcal{U}(0, 1)$ normalized to $\sum_j w_j = 1$, and each component chosen as $\mathcal{N}(\mu_j, \sigma_j^2)$, $\mu_j \sim \mathcal{N}(0, 1)$, $\sigma_j \sim \mathcal{U}(0, 1)$, or $\mathcal{U}(0, \mu_j)$) with probability $P \leq 0.5$, or from an ARMA$(p, q)$ process ($p \sim \mathcal{U}(1, p_{\max})$, $q \sim \mathcal{U}(1, q_{\max})$, parameters $\phi_i, \theta_j \sim \mathcal{U}(-1, 1)$, and stationarity enforced by $\sum_i \phi_i < 1$, $|\phi_p| < 1$) otherwise.

**Series Generation and Curation.** We normalize each $X$ per channel, compute $Y = f(X)$ via symbolic expressions, and discard any $X$ outside $f$'s domain or $|Y| > 10^4$ [12, 53]. For each random seed, we traverse all input/output channels, sampling each expression once. The resulting $S^2$ dataset contains $40M$ series–symbol pairs ($50B$ total length); series are patched for the time series encoder [54] and expressions tokenized for the symbolic encoder [55].

## 3.2 Model Architecture of `SymTime`

As shown in Figure 2 **(right)**, `SymTime` comprises a time series encoder, a symbol encoder, and momentum encoders, each trained with distinct objectives.

**Time Series Encoder and Masked Time Series Modeling (MTM).** An input time series is first divided into non-overlapping patches $P=\{p_1, p_2, \cdots, p_n\}$ using a sliding window approach [54, 17]. A 6-layer Transformer encodes non-overlapping patches $P$ with random masking and reconstructs them via

$$\mathcal{L}_{\text{mtm}} = \frac{1}{|\mathbf{M}_T|} \sum_{j \in \mathbf{M}_T} \|p_j - \hat{p}_j\|^2, \tag{2}$$

where $\mathbf{M}_T$ is the set of masked patch indices, and $\hat{p}_j$ represents the patch reconstructed by the time series encoder and linear projection [56, 57]. Then, we obtain the corresponding embedded sequence $T=\{t_{\text{cls}}, t_1, t_2, \cdots, t_n\}$, where $t_{\text{cls}}$ is the `[CLS]` token added by the time series encoder [58].

**Symbolic Encoder and Masked Language Modeling (MLM).** We treat symbolic expression data as natural language and use the 6-layer DistilBERT [55] as a symbol encoder to learn the representation of symbol through natural language mask modeling [19]. The loss optimized in this part is

$$\mathcal{L}_{\text{mlm}} = \frac{1}{|\mathbf{M}_S|} \sum_{j \in \mathbf{M}_S} \mathcal{H}(y_j, p_j^{\text{mask}}), \tag{3}$$

where $\mathbf{M}_S$ are masked symbol positions, $\mathcal{H}$ is cross-entropy loss, $p^{\text{mask}}(\hat{s})$ denote the model's predicted probability for the masked token $\hat{s}$, and $y_j$ is a one-hot vocabulary distribution with a probability of **1** for the ground-truth token. Then, we obtain the embedded sequence: $S = \{s_{\text{cls}}, s_1, s_2, \ldots, s_m\}$, where $s_{\text{cls}}$ is the `[CLS]` token added by the symbol encoder.

**Series–Symbol Contrastive Learning.** To leverage series-symbol data pairs with strong correlations (The correspondence between positive and negative samples is shown in Appendix C.2), we employ contrastive learning to enhance time series representation using symbolic information. Using momentum encoders [59], we project `[CLS]` embeddings via linear projections $g_t, g_s$ and define $\text{sim}(t, s) = g_t(t_{\text{cls}})^\top g_s'(s_{\text{cls}}')$, where $g_s'(s_{\text{cls}}')$ is the normalized symbol features generated by the momentum model. Similarly, $\text{sim}(s, t) = g_s(s_{\text{cls}})^\top g_t'(t_{\text{cls}}')$. We compute:

$$p^{t2s}(t) = \frac{\exp(\text{sim}(t, s_m)/\tau)}{\sum_m \exp(\text{sim}(t, s_m)/\tau)}, p^{s2t}(s) = \frac{\exp(\text{sim}(s, t_m)/\tau)}{\sum_m \exp(\text{sim}(s, t_m)/\tau)}, \tag{4}$$

where $\tau$ is a learnable temperature parameter [4, 59]. Let $y^{t2s}(t)$ and $y^{s2t}(s)$ represent the one-hot similarity, with positive pairs having a probability of **1** and negative pairs having **0** [59]. We optimize

$$\mathcal{L}_{\text{tsc}} = \frac{1}{2} \mathbb{E}\big[\mathcal{H}(y^{t2s}, p^{t2s}) + \mathcal{H}(y^{s2t}, p^{s2t})\big]. \tag{5}$$

### 3.3 Momentum Distillation for Masked Data Learning

Inspired by ALBEF [60], we treat random masking as noise and use momentum distillation to align the output representation of our encoder with its momentum counterpart. Let the similarity functions generated by the momentum encoders be $\text{sim}'(t, s) = g_t(t_{\text{cls}}')^\top g_s(s_{\text{cls}}')$ and $\text{sim}'(s, t) = g_s(s_{\text{cls}}')^\top g_t(t_{\text{cls}}')$. We compute soft pseudo targets $q^{t2s}(t)$ and $q^{s2t}(s)$ by replacing sim with sim' in Equation 4. In addition to the contrastive loss $\mathcal{L}_{\text{tsc}}$ (Equation 5), we compute pseudo-targets $q^{t2s}, q^{s2t}$ from momentum-encoder similarities sim' and optimize

$$\mathcal{L}_{\text{tsc}}^{\text{mod}} = \frac{1}{2} \mathbb{E}\left[\mathbf{KL}\left(q^{t2s}(t)\|p^{t2s}(t)\right) + \mathbf{KL}\left(q^{s2t}(s)\|p^{s2t}(s)\right)\right]. \tag{6}$$

The total pre-training objective is

$$\mathcal{L} = \mathcal{L}_{\text{mtm}} + \mathcal{L}_{\text{mlm}} + \alpha \mathcal{L}_{\text{tsc}} + (1 - \alpha)\mathcal{L}_{\text{tsc}}^{\text{mod}}. \tag{7}$$

### 3.4 Fine-tuning for Downstream Tasks

We use the pre-trained time series encoder as backbone. After instance normalization [61], we apply the following two strategies:

- **Classification:** patch the series, encode, and classify via a linear head [54, 62].
- **Reconstruction (forecasting, imputation, anomaly detection) [63, 64]:** decompose each series into trend and periodic components; regress trend directly, patch and encode the periodic part, then recombine for the final output.

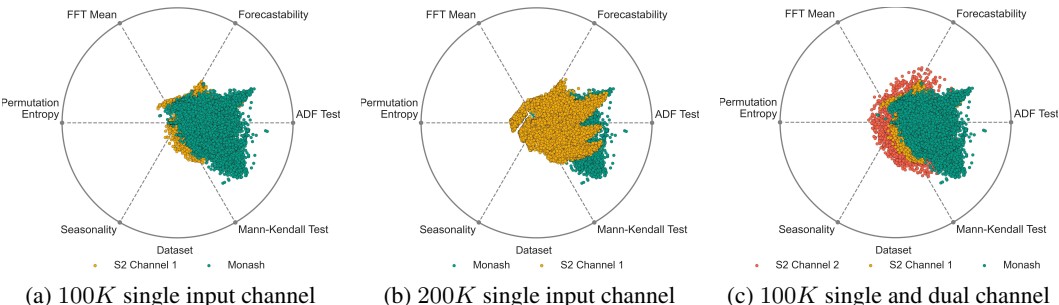

Figure 4: Radviz visualization of $S^2$ and Monash datasets.

## 4 Experiments

We explore multiple representation measures and conduct experimental verification on a variety of downstream task datasets to answer the following key questions:

- **RQ1:** Can the unrestrictedly generated $S^2$ dataset comprehensively cover diverse representation types of time series data?

- **RQ2:** Can `SymTime` pre-trained on the $S^2$ dataset achieve competitive results across five major TSA tasks (forecasting, classification, imputation and anomaly detection)?

- **RQ3:** Can `SymTime` learn fundamental representations of time series data on the synthetic $S^2$ dataset to alleviate the data scarcity in TSA?

- **RQ4:** Are the multiple pre-training objectives in `SymTime` effective, and can symbol expressions enhance TSA task performance?

- **RQ5:** How to demonstrate that `SymTime` learns semantic information of symbols?

### 4.1 Statistical Characterization and Representation Coverage of $S^2$ Dataset (RQ1)

**Target.** We quantify the range of representations that the $S^2$ dataset can cover through statistical metrics (including stationarity (ADF Test) [65], forecastability [66], frequency domain (FFT mean), seasonality [67], trend (Mann-Kendall Test) [68] and prmutation entropy [69]) (See Appendix B.5 for full descriptions).

**Setup.** We use Radviz [70] to visualize high-dimensional statistical features of 256-length time series segments from our synthetic $S^2$ and the Monash datasets [71]. From Monash (covering weather, traffic, electricity, tourism, medicine, and energy) we sample $200K$ segments per domain. For $S^2$, we sample 100K single-channel segments (Figure 4a), then $200K$ single-channel segments (Figure 4b), and finally $100K$ mixed single- and dual-channel segments (Figure 4c).

**Results.** Radviz visualization confirms that $S^2$ closely matches the Monash dataset across key statistics (stationarity, predictability, frequency, complexity, seasonality, trend), validating its use for pretraining. Expanding from $100K$ to $200K$ samples further broadens $S^2$'s coverage—surpassing Monash in some regions. Moreover, combining single- and dual-input samples dramatically increases diversity, as multi-variable expressions $f(x_1, \cdots, x_n)$ generate richer dynamics. These findings demonstrate that our infinitely scalable $S^2$ dataset covers the entire time series representation space.

### 4.2 Validation of `SymTime` in Five Time Series Analysis Tasks (RQ2)

**Setup.** We pre-trained `SymTime` on the $50B$-scale $S^2$ dataset using Equation 7 as the pre-training objective, with the model architecture detailed in Table 16. Then, we evaluate `SymTime` on five TSA tasks: long-term forecasting, short-term forecasting, classification, imputation and anomaly detection, using the TimesNet benchmark [72]. We use mean squared error (MSE) and mean absolute error (MAE) as the metrics for long-term forecasting and imputation tasks; overall weighted average (OWA) for short-term forecasting, which is unique metrics for M4 benchmark [73]; accuracy for classification; precision, recall and F1 score for anomaly detection. Detailed descriptions of datasets

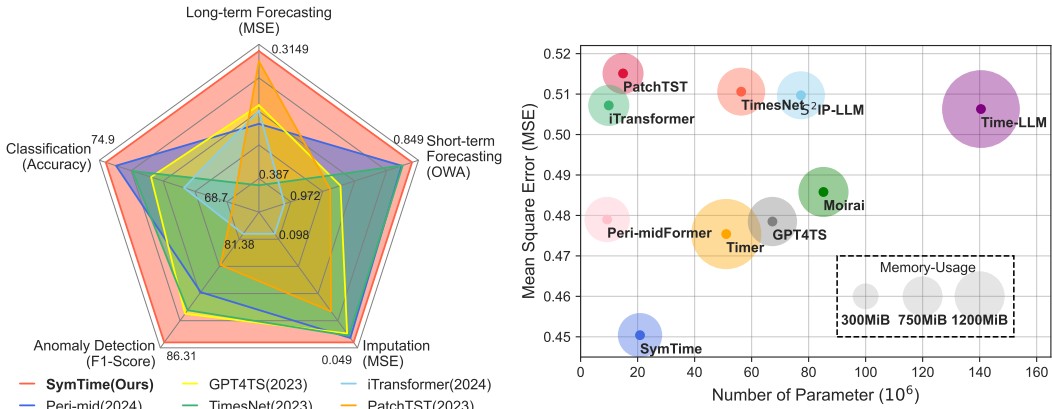

Figure 5: Model performance comparison with the state-of-the-art models in terms of five tasks (**left**). Complexity analysis on long time series forecasting tasks (ETTh1 dataset, forecasting length is 720 with 96 look-back windows) (**right**). Note that since the original backbone of Time-LLM [17] has too many parameters, we replaced it with GPT2 [94].

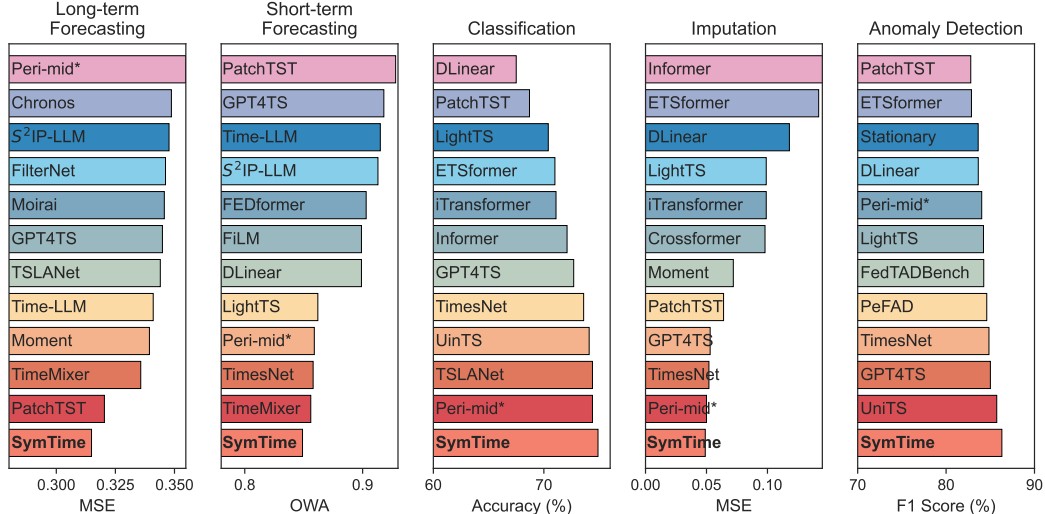

Figure 6: Validation of `SymTime` in 5 time series analysis tasks. We only show the average results of all datasets in this figure. See Appendix A for full results and analysis on different tasks. To ensure fair experimentation, we use the original model hyperparameters. For long-term prediction tasks, when a model is tested at multiple look-back windows, we select the best result.

and metrics for each task are provided in Appendix C.1 and C.3, while the pre-training and fine-tuning configurations for `SymTime` across downstream tasks are outlined in Appendix C.4 and C.5.

**Baselines.** We compare with various baselines including **Transformer-based models**: PatchTST [54], iTransformer [74], Autoformer [75], ETSformer [76], FEDformer [77], Non-stationary Transformer [78], Crossformer [79], Informer [80], Anomaly Transformer [81], Peri-midFormer [63]; **LLM-based models**: GPT4TS [2], Time-LLM [17], $S^2$IP-LLM [82]; **CNN-based models**: TimesNet [72], TSLANet [83], Rocket [84], InceptionTime (InTime) [85] and MICN [86]; **MLP-based models**: DLinear [87], LightTS [88], TimeMixer [89], FITS [90] and FilterNet [91]. We also compare with the **pre-trained foundation models**: Moirai [28], Timer [29], UniTS [92] and Moment [93]. Some models can be applied to all 5 TSA tasks, while others are suitable for only one or some specific tasks. For those pre-trained foundation models, we first load their pre-trained parameters and then fine-tune them in the same way.

**Main Results.** Figure 5 **(left)** compares `SymTime` with models of the same type, while Figure 6 presents `SymTime`'s performance against additional models across different tasks. These results

Table 1: Fine-tuning results of on long-term forecasting tasks with different pre-training dataset sizes. See Appendix F.8 for full results. The look-back window length for all experiments is **96**. **Red**: best, Blue: second best.

| Datasets | ETTm1 | | ETTm2 | | ETTh1 | | ETTh2 | | Weather | | Electircity | | Traffic | | Exchange | | Average | |
|---|---|---|---|---|---|---|---|---|---|---|---|---|---|---|---|---|---|---|
| Metrics | MSE | MAE | MSE | MAE | MSE | MAE | MSE | MAE | MSE | MAE | MSE | MAE | MSE | MAE | MSE | MAE | MSE | MAE |
| 0B | 0.401 | 0.409 | 0.293 | 0.339 | 0.487 | 0.474 | 0.376 | 0.412 | 0.257 | 0.289 | 0.193 | 0.284 | 0.471 | 0.310 | 0.383 | 0.415 | 0.358 | 0.366 |
| 1B | 0.376 | 0.398 | 0.292 | 0.331 | 0.461 | 0.459 | 0.403 | 0.419 | 0.257 | 0.282 | 0.199 | 0.285 | 0.473 | 0.303 | 0.370 | 0.410 | 0.354 | 0.361 |
| 10B | 0.376 | 0.393 | 0.281 | 0.329 | 0.444 | 0.444 | 0.376 | 0.408 | 0.250 | 0.279 | 0.196 | 0.286 | 0.473 | 0.294 | 0.368 | 0.407 | 0.345 | 0.355 |
| 25B | 0.378 | 0.393 | 0.278 | 0.325 | 0.434 | 0.438 | 0.371 | 0.405 | 0.253 | 0.282 | 0.195 | 0.288 | 0.467 | 0.299 | 0.357 | 0.401 | 0.342 | 0.354 |
| 50B | 0.371 | 0.390 | 0.274 | 0.321 | 0.430 | 0.436 | 0.365 | 0.402 | 0.247 | 0.276 | 0.187 | 0.276 | 0.457 | 0.291 | 0.359 | 0.401 | 0.336 | 0.349 |

Table 2: Fine-tuning results on short-term forecasting and imputation with different pre-training dataset sizes. See Appendix F.9 and F.11 for full results. **Red**: best, Blue: second best.

| tasks | Short-term Time Series Forecasting | | | | | Time Series Imputation | | | | | | | | | | | |
|---|---|---|---|---|---|---|---|---|---|---|---|---|---|---|---|---|---|
| Datasets | Yearly | Quartly | Monthly | Others | Avg | ETTm1 | | ETTm2 | | ETTh1 | | ETTh2 | | ECL | | Weather | |
| Metrics | Overall Weighted Average (OWA) | | | | | MSE | MAE | MSE | MAE | MSE | MAE | MSE | MAE | MSE | MAE | MSE | MAE |
| 0B | 0.782 | 0.913 | 0.964 | 1.097 | 0.887 | 0.042 | 0.122 | 0.038 | 0.106 | 0.112 | 0.230 | 0.065 | 0.160 | 0.058 | 0.155 | 0.036 | 0.053 |
| 1B | 0.784 | 0.911 | 0.893 | 1.082 | 0.861 | 0.039 | 0.119 | 0.031 | 0.097 | 0.113 | 0.217 | 0.066 | 0.160 | 0.057 | 0.152 | 0.033 | 0.050 |
| 10B | 0.783 | 0.905 | 0.896 | 1.055 | 0.859 | 0.038 | 0.119 | 0.030 | 0.095 | 0.107 | 0.213 | 0.063 | 0.158 | 0.056 | 0.151 | 0.033 | 0.048 |
| 25B | 0.788 | 0.909 | 0.877 | 1.061 | 0.856 | 0.037 | 0.118 | 0.028 | 0.093 | 0.104 | 0.207 | 0.059 | 0.154 | 0.055 | 0.152 | 0.030 | 0.043 |
| 50B | 0.786 | 0.872 | 0.872 | 1.045 | 0.849 | 0.036 | 0.117 | 0.026 | 0.088 | 0.095 | 0.201 | 0.058 | 0.148 | 0.054 | 0.151 | 0.028 | 0.038 |

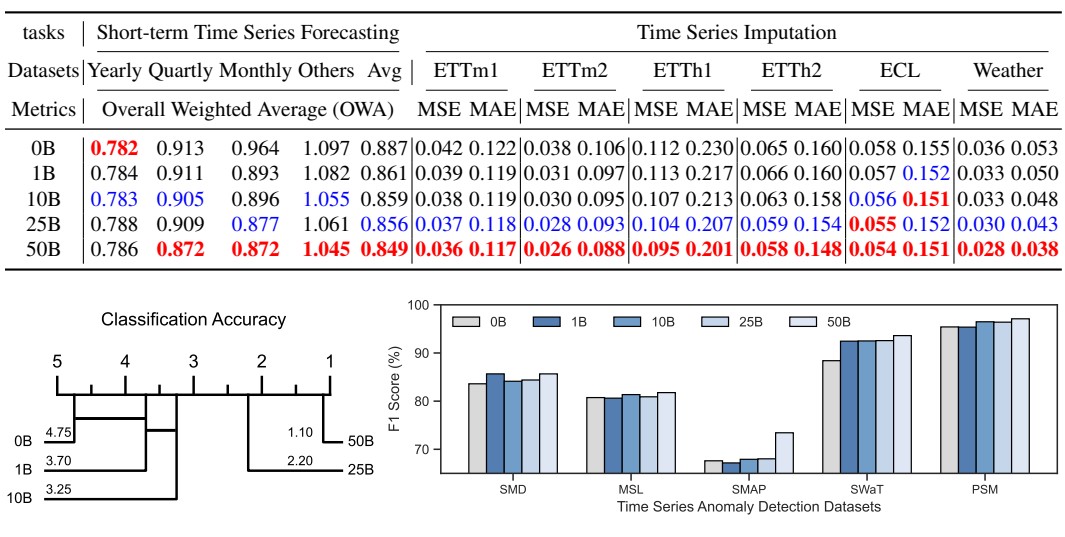

(a) Time Series Classification  (b) Time Series Anomaly Detection

Figure 7: Fine-tuning results of on classification and anomaly detection tasks with different pre-training dataset sizes. We use critical difference diagrams to measure the performance of classification in (a) [95], where the specific values represent the comprehensive ranking of the model on multiple datasets. See Appendix F.10 and F.12 for full results.

demonstrate that `SymTime`, pre-trained on the $S^2$ dataset, successfully learns fundamental representations of time series data and achieves competitive results when fine-tuned on downstream tasks. For each different task, the specific experimental settings and results are shown in: **long-term forecasting** (Appendix A.1), **short-term forecasting** (Appendix A.2), **classification** (Appendix A.3), **imputation** (Appendix A.4) and **anomaly detection** (Appendix A.5).

**Complexity Analysis.** We analyze the complexity of the model on the long-term forecasting ETTh1 dataset, with results shown in Figure 5 **(right)**. We consider the parameter count, the GPU memory required for forward and backward propagation when the batch size is 1, Using MSE as an evaluation metric, we find that `SymTime` achieves better performance with a smaller model parameter count and memory capacity than existing foundation models in forecasting task.

## 4.3 The Impact of Pre-training Dataset Size on `SymTime` Performance (RQ3)

**Setup.** Following the scaling laws of neural networks [6, 96], a large-scale and representationally comprehensive pre-training dataset is critical for building foundation models. In Section 3.1 and 4.1, we introduce and validate the unrestricted generation capability and comprehensive representational coverage of the $S^2$ dataset. Building on this, we proportionally divide the generated $S^2$ dataset into timestamp-based subsets $\{0B, 1B, 10B, 25B, 50B\}$ and pre-train `SymTime` under identical

configurations. To evaluate the effectiveness of pre-training, we fine-tune the models on five major TSA tasks (with $0B$ denoting direct fine-tuning without pre-training). The experimental results are summarized in Tables 1, Table 2 and Figure 7.

**Results.** The results reveal a clear trend: as the scale of the pre-trained $S^2$ dataset increases, `SymTime` shows progressively enhanced performance across diverse downstream tasks. Furthermore, pre-trained `SymTime` significantly outperforms its non-pre-trained counterpart. Given the unrestricted generation capability and comprehensive representational coverage of our generation methods, the $S^2$ dataset can theoretically scale to infinite size, progressively enhancing `SymTime`'s downstream task performance through pre-training. This demonstrates that the proposed $S^2$ mechanism enables models to learn the representations of time series while effectively mitigating performance degradation caused by data scarcity and imbalanced distributions.

### 4.4 Ablation Study on Different Pre-training Objectives (RQ4)

**Setup.** We conduct ablation studies on `SymTime`'s pre-training objectives using the ETTh1 and ETTh2 long-term forecasting datasets [80]. First, we establish 8 different control groups based on whether pre-training is performed, freezing the model and various pre-training losses: (1) Freeze, (2) Real-Data, (3) w/o Pre-train, (4) w/o MTM, (5) w/o MLM, (6) w/o T2S, (7) w/o S2T, (8) w/o Symbol and (9) w/o Distill. Specific explanations for the above control groups are provided in Appendix C.6 (Real-Data means we use real time series data of the same scale to pre-train the time series encoder only through the MTM.). Then, We adopt MSE as the metric, load the pre-trained parameters from each ablation configuration, and conduct multiple fine-tuning trials with varied random seeds on the ETT dataset. The average results with error are shown in Figure 8.

**Results.** As shown in Figure 8, pre-training with standard configurations through Equation 7 significantly enhances `SymTime`'s long-term forecasting performance compared to control groups, demonstrating the validity of its pre-training objectives and the effectiveness of $S^2$ data generation. It is worth noting that, whether on real time series data (Real-Data) or on synthetic $S^2$ data (w/o Symbol), removing the symbol encoder part

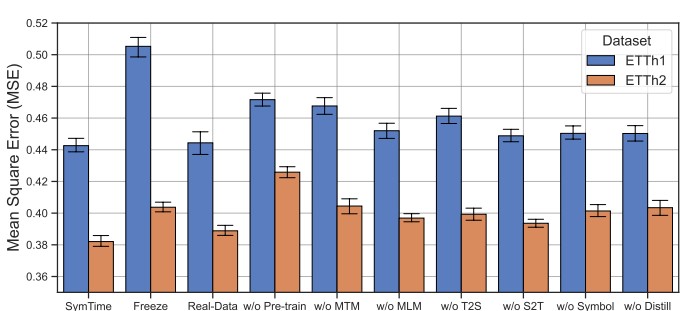

Figure 8: Ablation study on long-term forecasting task.

with contrastive learning and relying solely on MTM loss for temporal representation learning will degrade performance of `SymTime`. This suggests that semantic information provided by the symbolic encoder and series-symbol contrastive learning improves the temporal encoder's capability in long-term forecasting [97]. Similar findings are replicated in short-term forecasting tasks, as detailed in Appendix C.7.

**The Backbone in `SymTime`.** There is a time series encoder with a Transformer architecture and a symbol encoder consisting of a pre-trained LLM in `SymTime`. In Appendix C.8, we perform ablation experiments on the model architecture by replacing the backbone of `SymTime`. The results are shown in Figure 18. It is clear that replacing the backbone has no significant impact on downstream tasks. The improvement in model performance mainly comes from the dual-modal pre-training paradigm of time series and symbolic expressions (Equation 7).

### 4.5 Series-Symbol Representation Learning (RQ5)

We further analyze the data representations learned by `SymTime` through masked modeling and cross-modal contrastive loss. Given that contrastive learning aligns mutually positive samples in the representation space [98, 59], we pre-train `SymTime` on the $S^2$ dataset and extract its time series encoder and symbol encoder. Using generated univariate time series and unary symbolic expressions, we examine whether the representation spaces of encoders evolve during pre-training. Additionally,

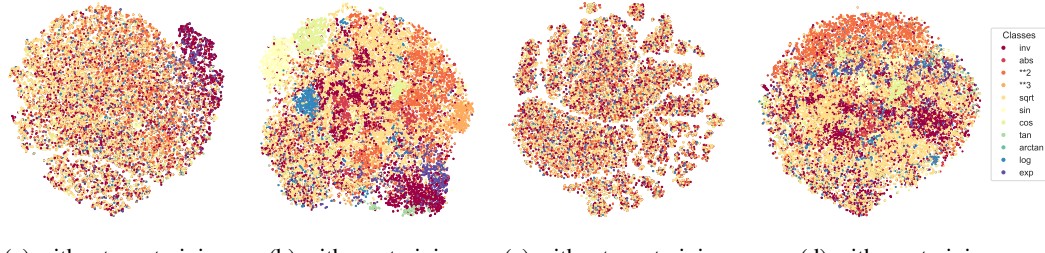

(a) without pre-training      (b) with pre-training      (c) without pre-training      (d) with pre-training

Figure 9: The t-SNE visualization of time series encoder and symbol encoder in `SymTime` representation space with 50 perplexity. (a)(b) time series encoder; (c)(d) symbol encoder.

we specifically evaluate the time series encoder's proficiency in temporal representation learning. Results demonstrate that the time series encoder, pre-trained via large-scale masked modeling [54, 27], achieves zero-shot imputation capability on both the $S^2$ dataset and real-world time series data. Detailed experimental analyses are provided in Appendix B.6.

### 4.5.1 Time Series Encoder Representation

**Setup.** We sample $20K$ single-channels series–symbol pairs from $S^2$, with each time series labeled by one unary operator (e.g., inv, sin, log). After patching, the time series segments are encoded by time series encoder of `SymTime` with and without series-symbol contrastive pretraining. The output embeddings are reduced via t-SNE [99].

**Results.** As shown in Figure 9 (a)(b), The untrained encoder produces entangled clusters (aside from outliers like inv, exp), whereas the pretrained encoder forms clear, operator-specific clusters (trigonometric: sin, cos; polynomial: pow2, pow3, sqrt; etc.), confirming that contrastive learning aligns symbolic semantics with series representations. The change in representation space before and after pre-training also proves that the time series encoder has mastered the semantic information of symbolic expressions. Ablations show both contrastive losses are essential.

### 4.5.2 Symbol Encoder Representation

**Setup.** We also use $20K$ series-symbol pairs to verify the representation space change of the symbol encoder. First, we tokenize the symbolic expression. Then, we input it into the encoder to obtain its `[CLS]` token [18, 60] and visualize it using the t-SNE [99].

**Results.** As shown in Figure 9 (c)(d), similar to the time series encoder, the pre-trained DistilBERT-based symbol encoder initially struggles to distinguish between different types of symbol [55]. However, after pre-training with MLM and contrastive learning, the encoder forms distinct clusters in the representation space for symbolic expressions of the same type [12]. Furthermore, the paired time-series data and their corresponding symbolic expression exhibit similar clustering characteristics, as shown in Figure 9 (b) and (d). This demonstrates that cross-modal representation learning effectively brings semantically related data points closer together in the representation space.

## 5 Conclusion

To mitigate the challenges of data scarcity and distribution imbalance in time series analysis, we introduce a dual-modality series-symbol ($S^2$) data generation mechanism that enables the unrestricted creation of high-quality time series data, along with corresponding symbolic representations. Leveraging this large-scale series-symbol synthetic dataset, we propose `SymTime`, a pre-trained foundation model that integrates both time series representations and symbolic semantic information through masked modeling and contrastive learning. Our pre-trained model demonstrates exceptional performance across five major TSA tasks, highlighting the effectiveness of both our data generation strategy and pre-training methodology. Looking ahead, we aim to scale up our approach by training larger models on synthetic datasets, further boosting performance on downstream tasks.

## Acknowledgement

This work was supported in part by Natural Science Basic Research Program of Shaanxi under Grant 2025JC-QYCX-060, in part by the National Natural Science Foundation of China under Grant 62206205, 62201419, 62472345, and 62471371, in part by the Young Talent Fund of Association for Science and Technology in Shaanxi, China under Grant 20230129, in part by the Guangdong High-level Innovation Research Institution Project under Grant 2021B0909050008, in part by the Guangzhou Key Research and Development Program under Grant 202206030003, and in part by the Fundamental Research Funds for the Central Universities under Grant QTZX25111. This research was supported by Huawei's Al Hundred Schools Program and was carried out using the Huawei Ascend AI technology stack.

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

# A  Main Results and Conclusions of the Five Tasks in Time Series Analysis

## A.1  Long-term Forecasting

**Setup.**   Time series forecasting, which analyzes historical data patterns to predict future trends, is crucial for financial market analysis, inventory management, energy demand and other fields [1, 100]. We adopt 8 real-world benchmark datasets for long-term forecasting, including ETTm1, ETTm2, ETTh1, ETTh2 [80], Weather [101], ECL [102], Traffic [103] and Exchange [104]. The forecasting lengths are set to $\{96, 192, 336, 720\}$. To ensure the fairness of the experiment, we set up three different look-back windows $\{96, 336, 512\}$ for the experiment, except Moirai and Timer are 672. For different models, we try our best to ensure that they are tested according to their original experimental configuration. For foundation models that can perform zero-shot forecasting and have been pre-trained, such as Moirai [28], Timer [29], Moment [93] and Chronos [22], we first load the pre-trained parameters of the model, and then perform supervised fine-tuning on it in the same way.

Table 3: Long-term forecasting task with **96** look-back windows. The results are averaged from four different series length $\{96, 192, 336, 720\}$. (* means former.) See Appendix F.1 for full results. **Red**: best, Blue: second best. The standard deviation is within 1%.

| Model | SymTime (Ours) | | Peri-mid* [63] | | Moirai [28] | | Timer [29] | | Time-LLM [17] | | TSLANet [83] | | $S^2$IP-LLM [82] | | GPT4TS [2] | | TimeMixer [89] | |
|---|---|---|---|---|---|---|---|---|---|---|---|---|---|---|---|---|---|---|---|
| Metrics | MSE | MAE | MSE | MAE | MSE | MAE | MSE | MAE | MSE | MAE | MSE | MAE | MSE | MAE | MSE | MAE | MSE | MAE |
| ETTm1 | 0.371 | **0.387** | 0.409 | 0.410 | 0.398 | 0.417 | 0.388 | 0.402 | **0.369** | 0.394 | 0.377 | 0.397 | 0.374 | 0.404 | 0.369 | 0.395 | 0.382 | 0.397 |
| ETTm2 | 0.274 | **0.321** | 0.290 | 0.328 | 0.296 | 0.348 | 0.405 | 0.408 | 0.275 | 0.324 | 0.283 | 0.327 | 0.266 | 0.325 | **0.264** | 0.328 | 0.279 | 0.325 |
| ETTh1 | **0.430** | **0.436** | 0.455 | 0.446 | 0.441 | 0.454 | 0.434 | 0.444 | 0.434 | 0.441 | 0.456 | 0.454 | 0.434 | 0.440 | 0.453 | 0.441 | | |
| ETTh2 | 0.365 | 0.402 | 0.400 | 0.416 | 0.402 | 0.411 | 0.428 | 0.441 | 0.369 | 0.407 | **0.355** | **0.391** | 0.362 | 0.405 | 0.359 | 0.403 | 0.388 | 0.408 |
| Weather | 0.247 | 0.276 | 0.262 | 0.283 | 0.265 | 0.299 | 0.329 | 0.358 | 0.247 | **0.269** | 0.259 | 0.352 | **0.243** | 0.274 | 0.265 | 0.285 | 0.253 | 0.280 |
| ECL | 0.187 | 0.276 | 0.178 | 0.267 | **0.167** | **0.252** | 0.177 | 0.267 | 0.180 | 0.269 | 0.199 | 0.283 | 0.191 | 0.283 | 0.206 | 0.291 | 0.185 | 0.274 |
| Traffic | 0.457 | 0.291 | 0.458 | 0.295 | 0.424 | 0.289 | 0.436 | **0.284** | 0.471 | 0.334 | 0.463 | 0.310 | **0.417** | 0.306 | 0.491 | 0.320 | 0.499 | 0.306 |
| Exchange | **0.359** | **0.401** | 0.388 | 0.417 | 0.373 | 0.417 | 0.382 | 0.425 | 0.376 | 0.414 | 0.368 | 0.414 | 0.472 | 0.478 | 0.370 | 0.411 | 0.403 | 0.423 |
| Average | **0.336** | **0.349** | 0.355 | 0.358 | 0.346 | 0.361 | 0.372 | 0.378 | 0.341 | 0.357 | 0.344 | 0.364 | 0.348 | 0.366 | 0.345 | 0.359 | 0.355 | 0.357 |

Table 4: Long-term forecasting task with **336** look-back windows. The results are averaged from four different series length $\{96, 192, 336, 720\}$. See Appendix F.2 for full results. **Red**: best, Blue: second best. The standard deviation is within 1%.

| Model | SymTime (Ours) | | PatchTST [54] | | TimeMixer [89] | | TimesNet [72] | | Autoformer [75] | | DLinear [87] | | iTransformer [74] | | TimeXer [62] | | FEDformer [77] | |
|---|---|---|---|---|---|---|---|---|---|---|---|---|---|---|---|---|---|---|---|
| Metrics | MSE | MAE | MSE | MAE | MSE | MAE | MSE | MAE | MSE | MAE | MSE | MAE | MSE | MAE | MSE | MAE | MSE | MAE |
| ETTm1 | **0.350** | 0.382 | 0.352 | 0.382 | 0.368 | 0.392 | 0.421 | 0.423 | 0.618 | 0.539 | 0.357 | **0.379** | 0.368 | 0.395 | 0.372 | 0.395 | 0.441 | 0.452 |
| ETTm2 | **0.256** | 0.316 | 0.258 | **0.315** | 0.262 | 0.318 | 0.282 | 0.334 | 0.400 | 0.420 | 0.291 | 0.353 | 0.272 | 0.329 | 0.262 | 0.317 | 0.325 | 0.377 |
| ETTh1 | **0.413** | 0.432 | 0.419 | **0.432** | 0.430 | 0.437 | 0.485 | 0.480 | 0.580 | 0.539 | 0.425 | 0.440 | 0.450 | 0.457 | 0.493 | 0.483 | 0.450 | 0.472 |
| ETTh2 | 0.341 | 0.390 | **0.331** | **0.379** | 0.396 | 0.425 | 0.409 | 0.440 | 0.663 | 0.604 | 0.490 | 0.476 | 0.390 | 0.416 | 0.375 | 0.410 | 0.430 | 0.467 |
| Weather | 0.238 | 0.273 | 0.258 | 0.292 | **0.235** | 0.273 | 0.250 | 0.286 | 0.441 | 0.450 | 0.245 | 0.298 | 0.238 | **0.272** | 0.287 | 0.290 | 0.313 | 0.363 |
| ECL | 0.164 | 0.258 | 0.165 | 0.294 | 0.169 | 0.260 | 0.197 | 0.297 | 0.236 | 0.346 | 0.170 | 0.269 | **0.163** | **0.257** | 0.172 | 0.267 | 0.214 | 0.327 |
| Traffic | 0.391 | **0.267** | 0.396 | 0.268 | 0.411 | 0.271 | 0.615 | 0.331 | 0.676 | 0.413 | 0.465 | 0.320 | 0.401 | 0.283 | 0.452 | 0.281 | 0.610 | 0.376 |
| Exchange | **0.367** | **0.406** | 0.385 | 0.420 | 0.415 | 0.438 | 0.548 | 0.532 | 1.053 | 0.807 | 0.448 | 0.462 | 0.392 | 0.427 | 0.409 | 0.500 | 0.376 | 0.427 |
| Average | **0.315** | **0.341** | 0.320 | 0.348 | 0.336 | 0.352 | 0.401 | 0.390 | 0.583 | 0.515 | 0.361 | 0.375 | 0.347 | 0.355 | 0.353 | 0.368 | 0.395 | 0.408 |

**Results.**   Tables 3, Table 4, and Table 5 respectively show the results of SymTime with look-back windows of length $\{96, 336, 512\}$. It can be seen that SymTime demonstrates excellent performance in time series long-term forecasting tasks. Our model surpasses Peri-midFormer, GPT4TS and TimesNet, which are foundation models for the five major tasks, as well as Moirai and Timer, two general forecasting models. Compared with Time-LLM and $S^2$IP-LLM, which use pre-trained large language models as backbone, SymTime achieves better results with a more lightweight Transformer encoder.

Table 5: Long-term forecasting task with **512** look-back windows. The results are averaged from four different series length {96, 192, 336, 720}. See Appendix F.3 for full results. **Red**: best, Blue: second best. The standard deviation is within 1%.

| Models | SymTime Ours | | PatchTST [54] | | TimeMixer [89] | | TimesNet [72] | | Autoformer [75] | | DLinear [87] | | iTransformer [74] | | TimeXer [62] | | FITS [90] | |
|---|---|---|---|---|---|---|---|---|---|---|---|---|---|---|---|---|---|---|
| Metrics | MSE | MAE | MSE | MAE | MSE | MAE | MSE | MAE | MSE | MAE | MSE | MAE | MSE | MAE | MSE | MAE | MSE | MAE |
| ETTm1 | 0.356 | **0.380** | 0.352 | 0.382 | 0.371 | 0.392 | 0.425 | 0.430 | 0.556 | 0.518 | 0.358 | 0.380 | 0.367 | 0.397 | 0.378 | 0.401 | 0.374 | 0.384 |
| ETTm2 | 0.265 | 0.320 | 0.256 | 0.317 | 0.263 | 0.323 | 0.294 | 0.344 | 0.371 | 0.416 | 0.275 | 0.340 | 0.273 | 0.331 | 0.274 | 0.329 | **0.254** | **0.313** |
| ETTh1 | 0.414 | **0.432** | 0.413 | 0.434 | 0.429 | 0.444 | 0.481 | 0.486 | 0.627 | 0.579 | 0.418 | 0.438 | 0.446 | 0.460 | 0.475 | 0.479 | 0.418 | 0.441 |
| ETTh2 | 0.365 | 0.405 | 0.357 | 0.409 | 0.373 | 0.410 | 0.397 | 0.432 | 0.687 | 0.609 | 0.499 | 0.478 | 0.388 | 0.417 | **0.354** | **0.400** | 0.363 | 0.408 |
| weather | 0.234 | 0.273 | 0.245 | 0.284 | **0.231** | **0.271** | 0.251 | 0.288 | 0.489 | 0.486 | 0.241 | 0.292 | 0.249 | 0.280 | 0.282 | 0.284 | 0.244 | 0.281 |
| ECL | 0.163 | 0.267 | 0.169 | 0.269 | 0.177 | 0.274 | 0.201 | 0.302 | 0.353 | 0.393 | 0.167 | 0.267 | **0.162** | **0.257** | 0.171 | 0.270 | 0.169 | 0.265 |
| Traffic | 0.395 | **0.268** | 0.399 | 0.272 | 0.410 | 0.270 | 0.624 | 0.334 | 0.705 | 0.435 | 0.433 | 0.305 | **0.383** | 0.273 | 0.466 | 0.287 | 0.420 | 0.287 |
| Exchange | **0.384** | **0.412** | 0.398 | 0.423 | 0.517 | 0.497 | 0.718 | 0.608 | 0.944 | 0.768 | 0.500 | 0.494 | 0.427 | 0.467 | 0.514 | 0.506 | 0.393 | 0.439 |
| Average | **0.322** | **0.345** | 0.324 | 0.349 | 0.346 | 0.360 | 0.424 | 0.403 | 0.591 | 0.525 | 0.361 | 0.374 | 0.337 | 0.360 | 0.364 | 0.369 | 0.329 | 0.352 |

## A.2 Short-term Forecasting

**Setup.** We adopt M4 benchmark [73] for short-term forecasting, which contains the yearly, quarterly and monthly collected univariate marketing data. Then, we use symmetric mean absolute error (SMAPE), mean absolute scaled error (MASE) and overall weighted average (OWA) to measure the forecasting performance, which are calculated as detailed in Appendix C.3.

**Results.** Table 6 indicates that SymTime after pre-training, surpasses TimeMixer, Peri-midFormer and TimesNet on the short-term forecasting tasks in terms of SMAPE, MASE and OWA metrics, achieving state-of-the-art performance. Specifically, SymTime performs well on Yearly, Quarterly and Monthly datasets, demonstrating its capability to capture not only the trends of annual variations but also the cyclic characteristics of seasonal and monthly encoding.

Table 6: Short-term forecasting task on M4. The prediction lenghs are {6, 48} and results are weighted averaged from several datasets under different sample intervals. (* means former, TMixer is TimeMixer, $S$-LLM is $S^2$IP-LLM, T-LLM is Time-LLM). See Appendix F.4 for full results. **Red**: best, Blue: second best. The standard deviation is within 1%.

| Models | SymTime (Ours) | Peri-mid* [63] | $S$-LLM [82] | T-LLM [17] | GPT4TS [2] | TMixer [89] | PatchTST [54] | iTrans* [74] | TimesNet [72] | DLinear [87] | LightTS [88] | FED* [77] | In* [80] |
|---|---|---|---|---|---|---|---|---|---|---|---|---|---|
| SMAPE | **11.785** | 11.897 | 12.514 | 12.584 | 12.367 | 11.885 | 12.866 | 13.233 | 11.888 | 12.500 | 11.962 | 12.605 | 15.018 |
| MASE | **1.584** | 1.607 | 1.726 | 1.763 | 1.767 | 1.598 | 1.734 | 1.850 | 1.607 | 1.678 | 1.609 | 1.677 | 2.096 |
| OWA | **0.849** | 0.859 | 0.913 | 0.915 | 0.918 | 0.856 | 0.928 | 0.972 | 0.858 | 0.899 | 0.862 | 0.903 | 1.102 |

## A.3 Classification

**Setup.** Time series classification is crucial for the identification and diagnosis of patterns in complex systems and plays a significant role in various fields such as financial analysis, medical diagnosis and industrial monitoring [95]. Using the experimental setup from TimesNet [72], we test SymTime's discriminative ability on 10 UEA multivariate time series classification datasets [105], including categories such as Industry, Face Detection, ECG, Voice and Transportation.

**Results.** As shown in Figure 10, SymTime achieves an average accuracy of 74.9%, surpassing all baselines, indicating that SymTime is competitive in classification tasks.

## A.4 Imputation

**Setup.** Sensors monitoring complex systems in the real world may experience distortions or malfunctions, leading to partial missing data in the collected time series. Therefore, time series imputation is crucial for the recovery of complete datasets. We verify SymTime's imputation capabilities on 6

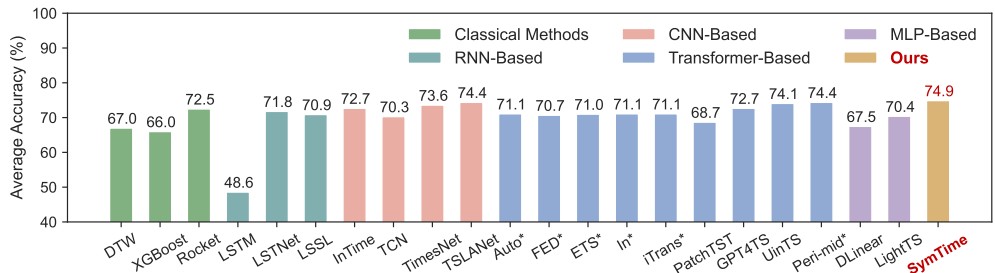

Figure 10: Comparison of the average accuracy of `SymTime` and other baselines on 10 UEA datasets. See Appendix F.5 for full results.

datasets: ETTm1, ETTm2, ETTh1, ETTh2 [80], Weather [101] and ECL [102]. To test the model's imputation ability under varying degrees of missing data, we add random masks at proportions of $\{12.5\%, 25\%, 37.5\%, 50\%\}$ in point level on time series of length 96. Since `SymTime` was pre-trained by randomly masking patches level for series reconstruction and masks are added randomly in point level in the imputation task. Considering the differences between these masking approaches and the potential disruption of the series's original trends and periodic features at higher mask rates, we adopt per-interpolation for the masked series from [63]. Analysis and ablation experiments regarding this method are presented in Appendix C.9. The results demonstrate that per-interpolation can be used as a model-independent feature engineering to improve the performance of downstream tasks.

Table 7: Imputation task, where we randomly mask {12.5%, 25%, 37.5%, 50%} time points of length-96 time series. The reuslts averaged from 4 different mask ratios. (* means former.) See Appendix F.6 for full results. **Red**: best, Blue: second best.

| Model | SymTime (Ours) | | GPT4TS [2] | | TimesNet [72] | | Peri-mid* [63] | | Moment [93] | | iTrans* [74] | | PatchTST [54] | | DLinear [87] | | LightTS [88] | |
|---|---|---|---|---|---|---|---|---|---|---|---|---|---|---|---|---|---|---|
| Metric | MSE | MAE | MSE | MAE | MSE | MAE | MSE | MAE | MSE | MAE | MSE | MAE | MSE | MAE | MSE | MAE | MSE | MAE |
| ETTm1 | 0.036 | 0.116 | 0.028 | 0.109 | **0.027** | **0.107** | 0.036 | 0.116 | 0.074 | 0.168 | 0.072 | 0.185 | 0.049 | 0.143 | 0.090 | 0.204 | 0.068 | 0.182 |
| ETTm2 | 0.026 | 0.088 | **0.022** | **0.088** | **0.022** | 0.089 | 0.026 | 0.087 | 0.031 | 0.108 | 0.082 | 0.191 | 0.030 | 0.101 | 0.102 | 0.212 | 0.068 | 0.176 |
| ETTh1 | 0.095 | 0.201 | 0.093 | 0.200 | **0.089** | 0.199 | 0.091 | **0.196** | 0.139 | 0.234 | 0.148 | 0.269 | 0.126 | 0.231 | 0.169 | 0.283 | 0.159 | 0.278 |
| ETTh2 | 0.058 | 0.148 | 0.052 | **0.147** | **0.050** | 0.148 | 0.057 | **0.147** | 0.061 | 0.159 | 0.139 | 0.254 | 0.066 | 0.164 | 0.163 | 0.273 | 0.143 | 0.258 |
| ECL | **0.054** | **0.151** | 0.093 | 0.212 | 0.094 | 0.211 | 0.063 | 0.169 | 0.094 | 0.211 | 0.099 | 0.224 | 0.078 | 0.192 | 0.128 | 0.256 | 0.108 | 0.238 |
| Weather | **0.028** | **0.038** | 0.032 | 0.058 | 0.030 | 0.056 | 0.029 | 0.041 | 0.035 | 0.075 | 0.052 | 0.114 | 0.033 | 0.057 | 0.053 | 0.116 | 0.047 | 0.106 |
| Average | **0.049** | **0.124** | 0.053 | 0.136 | 0.052 | 0.135 | 0.050 | 0.126 | 0.072 | 0.159 | 0.099 | 0.206 | 0.064 | 0.148 | 0.118 | 0.224 | 0.099 | 0.206 |

**Results.** Table 7 shows that `SymTime` outperforms Peri-midFormer, GPT4TS and TimesNet in overall performance establishing `SymTime` as the latest state-of-the-art approach. Although `SymTime`'s performance on the ETT series of datasets is not as strong as GPT4TS, it achieves more significant effects on datasets with a higher number of channels, such as ECL and Weather.

## A.5 Anomaly Detection

**Setup.** Time series anomaly detection is crucial for rapidly identifying anomalies in critical areas, aiding in risk prevention and decision optimization. Due to the difficulty in annotating time series anomalies, we focus primarily on unsupervised anomaly detection. We conduct experiments on 5 widely used anomaly detection datasets: SMD and SMAP [106], MSL [107], SWaT [108], PSM [109], encompassing service monitoring, space & earth exploration, and water treatment applications. We adopt the same data preprocessing method as the Anomaly Transformer [81], dividing the data into non-overlapping segments of length 100 for reconstruction. Specifically, normal data is used for model training and we employ a simple reconstruction loss to help the model learn the distribution of normal data [63]. In subsequent testing phases, reconstructed outputs exceeding a specified threshold are considered anomalies.

Table 8: Anomaly detection task, where we calculate the F1-score (as %) for each dataset. (* means former, TNet is TimesNet, PTST is PatchTST.) A higher value of F1-score indicates a better performance. See Appendix F.7 for full results. **Red**: best, Blue: second best.

| Model | SymTime (Ours) | UniTS [92] | Peri-mid* [63] | GPT4TS [2] | TNet [72] | PTST [54] | LightTS [88] | DLinear [87] | iTrans* [74] | Anomaly [81] | Stationary [78] | Cross* [79] | In* [80] | Auto* [75] |
|---|---|---|---|---|---|---|---|---|---|---|---|---|---|---|
| SMD | **85.66** | 83.69 | 84.08 | 84.49 | 84.37 | 84.62 | 82.53 | 79.76 | 80.19 | 85.49 | 82.97 | 77.22 | 77.88 | 71.17 |
| MSL | 81.77 | 81.16 | 80.68 | 82.03 | 81.14 | 78.70 | 78.95 | 81.87 | 72.47 | **83.31** | 76.68 | 80.59 | 81.07 | 82.22 |
| SMAP | 73.43 | **74.00** | 67.53 | 68.85 | 69.05 | 68.82 | 69.21 | 67.30 | 66.72 | 71.18 | 69.02 | 67.12 | 73.26 | 73.97 |
| SWaT | **93.61** | 92.51 | 91.64 | 92.60 | 92.61 | 85.72 | 93.33 | 92.66 | 92.64 | 83.10 | 92.24 | 90.22 | 80.35 | 79.19 |
| PSM | 97.10 | **97.31** | 96.21 | 97.09 | 97.06 | 96.08 | 97.15 | 96.64 | 94.88 | 79.40 | 97.23 | 92.52 | 90.43 | 88.24 |
| Avg F1 | **86.31** | 85.73 | 84.03 | 85.01 | 84.85 | 82.79 | 84.23 | 83.64 | 81.38 | 80.50 | 83.63 | 81.53 | 80.60 | 78.96 |

**Results.** The results in Table 8 show that `SymTime` surpasses all previous models such as TimesNet and GPT4TS in the time series anomaly detection task and becomes the latest SOTA model. Compared with UniTS [92] pre-trained on real time series data, `SymTime` is pre-trained on synthetic datasets and achieves better model performance.

# B   Analysis of Series-Symbol ($S^2$) Dataset and Model Pre-training

Table 9: Some symbols used in data generation and their explanations.

| Symbols | Explanation | Symbols | Explanation |
|---|---|---|---|
| $X$ | sampling series | $Y$ | generated series |
| $f(\cdot)$ | symbolic expression | $e_t$ | white noise sequence |
| $M$ | the input channels number | $N$ | the output channels number |
| $\mathcal{U}$ | uniform distribution | $\mathcal{N}$ | normal distribution |
| $P$ | probability of selecting sampling methods | $k$ | total number of mixed distributions |
| $p$ | the order of the AR process | $q$ | the order of the MA process |
| $\phi_p$ | the parameters of AR process | $\theta_q$ | the parameter of MA process |

Based on the viewpoint of complex dynamic system modeling, this paper proposes a bimodal series-symbol ($S^2$) generation mechanism to alleviate the problem of shortage of training data in the field of time series analysis. Table 9 shows the specific symbols we used in constructing symbolic expressions in Section 3.1. This part is mainly divided into the following sections:

- B.1 Series-Symbol Data Display: This section shows the symbolic expressions and time series data that can be generated through the $S^2$ data generation mechanism.

- B.2 Composition and Usage of the Series-Symbol Dataset: This section introduces the specific composition of the $S^2$ dataset and how to use it when training `SymTime`.

- B.3 Statistics Analysis: This section examines and analyzes the basic statistical representation of the $S^2$ data.

- B.4 Analysis of Existing Large-scale Datasets for Time Series Pre-training: This section analyzes the data shortage and distribution imbalance faced by large-scale time series pre-training datasets in TSA.

- B.5 $S^2$ Dataset Statistical Characterization Coverage Experiments: This section details the specific configuration of the characterization coverage experiments in the Section 4.1.

- B.6 Masked Time Series Modeling and Zero-shot Imputation for Representation Learning:

- B.7 Time Complexity Analysis of $S^2$ Data Generation Mechanism: In this section we analyze in detail the time complexity of the $S^2$ data generation mechanism.

- B.8 The Selection of the Unary Operators: In this section we analyze the unary operators used in $S^2$ data generation.

- B.9 The Limitations of $S^2$ Generation Mechanism.

## B.1 Series-Symbol Data Display

In Figure 11, we show the visualization of the generated series from 1 input channel and 1 output channel to 4 input channels and 4 output channels. We show two sets of cases for each input and output channel. The symbolic expressions $f(\cdot)$ for the generated series in (a), (c), (e) and (g) in Figure 11 are shown above.

The symbolic expressions with text format are shown as follow:

**Symbolic expression of Figure 11 (a)**

$y_1$ = (-0.795 add ((-0.675 mul ((0.999 add (-6.7 mul $x_1$)))**2) add ((-0.798 mul inv((-5.99 add (-0.751 mul $x_1$)))) sub (9.68 mul sqrt((-7.37 add (0.756 mul $x_1$)))))))

**Symbolic expression of Figure 11 (c)**

$y_1$ = (-3.39 add (((0.56 mul (inv((-98.9 add (58.2 mul $x_2$))) mul ((-19.7000 mul $x_1$) sub (31.9000 mul $x_2$)))) sub (40.4000 mul $x_1$)) add (0.71 mul (((7.13 mul $x_2$) sub (-1.68 mul ($x_1$ mul sqrt((-92.8000 add (0.054 mul ($x_2$ mul ((0.327 mul $x_2$) sub (2.3 mul $x_2$)))))))))) mul $x_1$))))

$y_2$ = (1.0 add ((68.9 mul $x_2$) sub (((80.9 mul ($x_1$ mul ($x_1$ mul ((6.1000 mul $x_2$) sub ((34.2 mul sqrt((64.4 add (29.2000 mul $x_1$)))) add (-5.24 mul $x_1$))))))) add (6.78 mul $x_2$)) sub (((0.5730 mul $x_1$) sub ((2.34 mul $x_2$) sub (-6.72 mul $x_1$))) add (0.966 mul sqrt((76.8000 add (-7.79 mul $x_1$)))))))))

**Symbolic expression of Figure 11 (e)**

$y_1$ = (0.795 add ((0.42 mul $x_3$) sub ((4.39 mul $x_1$) add (((0.1430 mul $x_2$) sub ((-5.28 mul $x_3$) add ((((-0.028 mul (((((1.27 mul $x_3$) sub (((0.331 mul $x_2$) sub ((2.99 mul $x_3$) add (-0.932 mul (((0.606 mul $x_1$) sub (0.967 mul $x_3$)) mul $x_3$)))) sub (-0.609 mul $x_3$))) add (-1.25 mul $x_1$)) mul $x_1$)) sub (77.3000 mul $x_1$)) sub (1.93 mul $x_3$)))) sub (16.7 mul $x_3$)))))

$y_2$ = (-9.2900 add ((0.398 mul (((((-49.7 mul $x_1$) sub ((5.93 mul sin((6.54 add (-0.045 mul $x_1$)))) add ((62.3000 mul inv(((0.138 mul $x_2$) add (29.0 mul $x_1$)))) add (8.75 mul $x_2$)))) add ((-0.9500 mul $x_3$) add (-8.1 mul $x_1$))) mul $x_3$)) add ((-9.74 mul $x_3$) add (((((-0.9 mul $x_3$) sub (4.45 mul sqrt((-0.373 add (-0.151 mul $x_3$)))))) add (-54.6 mul $x_3$)) sub (-0.758 mul ((85.3000 add (8.74 mul $x_3$)))**2)))))

$y_3$ = (-0.975 add ((-54.4000 mul sqrt((-0.722 add (-9.33 mul $x_2$)))) sub (1.45 mul ((66.4 add (-9.65 mul $x_1$)))**2)))

**Symbolic expression of Figure 11 (g)**

$y_1$ = (-7.17 add (0.537 mul $x_1$))

$y_2$ = (-0.843 add (48.8000 mul $x_1$))

$y_3$ = (57.3000 add ((((-0.449 mul $x_2$) add (-1.32 mul $x_3$)) add ((-0.9400 mul $x_4$) add (0.51 mul $x_1$)))))

$y_4$ = (-0.2040 add (((-6.6000 mul inv((0.88 add (58.1 mul $x_4$)))) sub ((-23.0 mul $x_4$) add ((-91.0 mul $x_3$) sub (-93.6000 mul $x_2$)))) sub ((-6.6000 mul $x_4$) sub (0.9580 mul (($x_3$ mul $x_3$) mul ((-0.45 mul $x_2$) sub (((((-9.09 mul $x_4$) sub ((8.93 mul sqrt(((-26.6 mul $x_4$) add (-0.907 mul $x_1$)))) add (-6.2 mul $x_4$))) sub (-0.078 mul $x_4$)) sub (-16.5 mul $x_2$)))))))))

## B.2 Composition and Usage of the Series-Symbol Dataset

We set the maximum number of input channels and the maximum number of output channels to 6 and 12 respectively to generate symbolic expressions and series. Each symbolic expression is sampled only once. We generated a total of 40M pairs of series and symbols. The cumulative series length is 50B. The data number of each input channel and output channel in the dataset is shown in Figure 12.

When pre-training SymTime with $S^2$ dataset, we start by combining the sampled and generated series and then segmenting them into patches using a sliding window [54, 17]. The sliding window's kernel size and step size are both set to 16. Due to the requirement for mask time series modeling (MTM) [27, 26], there is no overlap between adjacent patches. Given the varying number of input and output channels in the data, the series from the maximum input and output channels can be segmented into up to 288 patches ($18 \times 256/16$) [110]. For series with fewer than 288 patches, we pad them with zeros

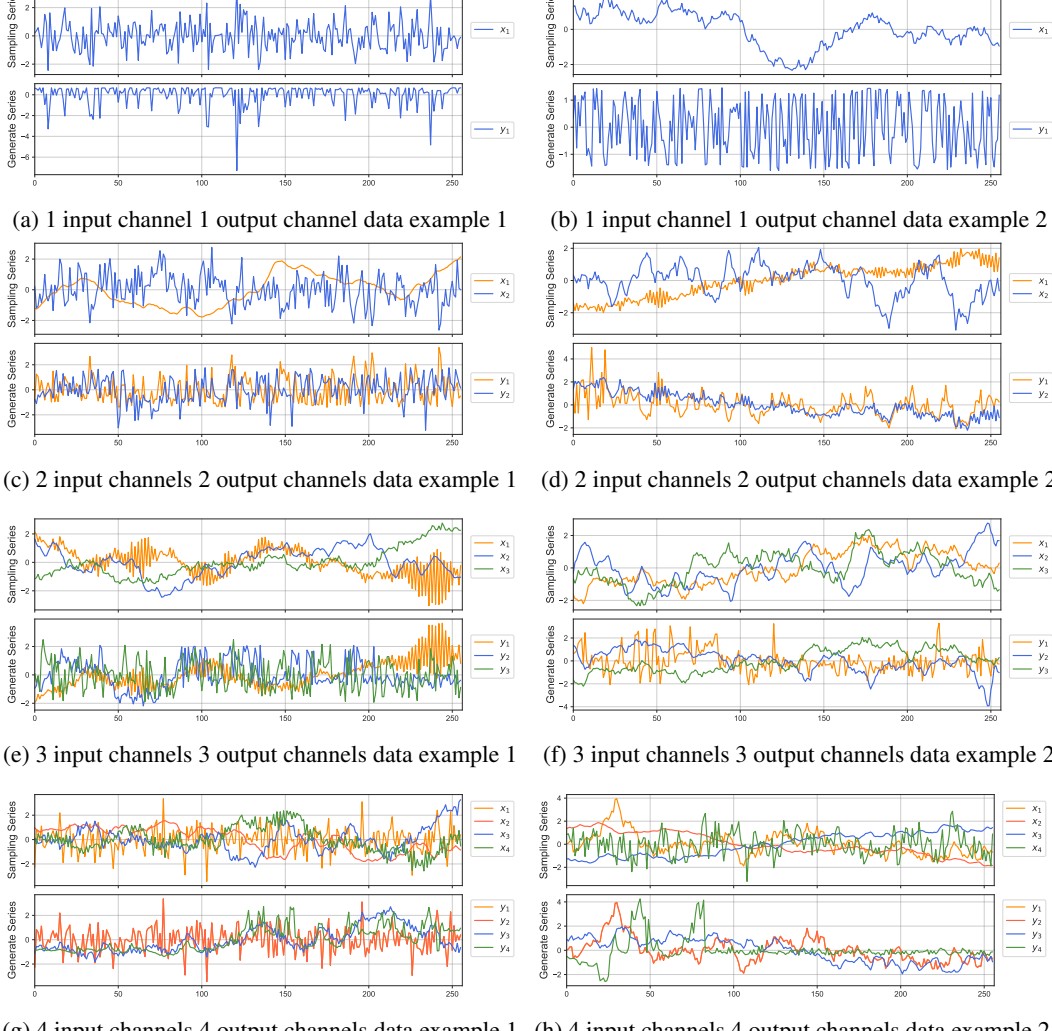

(a) 1 input channel 1 output channel data example 1    (b) 1 input channel 1 output channel data example 2

(c) 2 input channels 2 output channels data example 1    (d) 2 input channels 2 output channels data example 2

(e) 3 input channels 3 output channels data example 1    (f) 3 input channels 3 output channels data example 2

(g) 4 input channels 4 output channels data example 1    (h) 4 input channels 4 output channels data example 2

Figure 11: Visualization of series from 1 input channel 1 output channel to 4 input channels 4 output channels.

to align the length. Next, for symbolic expressions in natural language form [19, 12, 82, 75, 111], we set a maximum length of 512 characters and perform tokenization. Ultimately, the time series patches and natural language tokens are fed into the time series encoder and the LLM of the Transformer architecture, respectively.

### B.3   Statistics Analysis

**Setup.**   In Section B.2, we provide a detailed introduction to the generation process, composition and usage of the $S^2$ dataset [12, 14, 48]. In this section, we first conduct a random sampling analysis of the statistical characteristics of the $S^2$ dataset, including stationarity [65] and predictability [66, 29].

**Stationarity.**   Stationarity is one of the fundamental properties of time series [50, 51]. This attribute ensures that the statistical characteristics of time series data remain consistent across different time points, which is crucial for building effective predictive models and making reliable statistical inferences. To this end, we employ the Augmented Dickey-Fuller (ADF) [65] test to examine the stationarity of the data, thereby determining whether the generated $S^2$ dataset is suitable for deep neural networks (DNNs) to learn representations of time series.

**Forecastability.** The forecastability of a time series refers to the ability and accuracy to forecast future values based on historical data and statistical models [52, 29]. For certain specific time series and complex systems, such as stock markets, it is often challenging to predict their subsequent developments. Therefore, it is necessary to test whether the $S^2$ dataset is non-chaotic and learnable. Forecastability is calculated by subtracting the entropy of the series' Fourier decomposition as adopted from [66] and [29], where a higher forecastability value indicates better predictability. Please note that since the method provided by [66] is only applicable to multivariate time series, we merge the input channels and output channels together for calculation.

**Test Methods and Results.** For the multiple input-output channels presented in the Table 10, we randomly selected 1,000 samples to calculate their average ADF statistics, p-values, and Forecastability metrics. The results indicate that the average p-value from the ADF test across all samples is greater than 0.05, suggesting that the majority of the generated series in the $S^2$ dataset are non-stationary time series, posing a challenge in modeling and learning [65]. However, the Forecastability metric, which is greater than 0.3 for all tested samples, indicates that the generated series $Y$ is not produced by a chaotic system and is, overall, predictable.

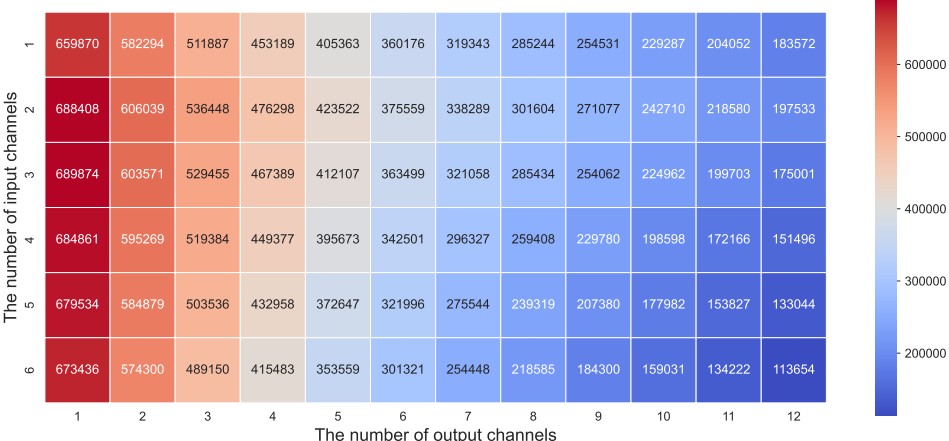

Figure 12: The number of samples in each part of the $S^2$ dataset.

Table 10: Results of the stationarity and forecastability tests for the $S^2$ dataset.

| inputs | outputs | ADF | p value | forecast | inputs | outputs | ADF | p value | Forecastability |
|--------|---------|-----|---------|----------|--------|---------|-----|---------|-----------------|
| 1 | 1 | -12.77 | 0.0538 | 0.3155 | 1 | 6 | -11.48 | 0.0619 | 0.3375 |
| 2 | 2 | -11.89 | 0.0568 | 0.3199 | 2 | 6 | -11.46 | 0.0733 | 0.3218 |
| 3 | 3 | -12.40 | 0.0544 | 0.3328 | 3 | 6 | -11.43 | 0.0625 | 0.3244 |
| 4 | 4 | -11.66 | 0.0617 | 0.3491 | 4 | 6 | -11.53 | 0.0640 | 0.3428 |
| 5 | 5 | -11.38 | 0.0628 | 0.3140 | 5 | 6 | -12.32 | 0.0597 | 0.3284 |
| 6 | 6 | -12.43 | 0.0625 | 0.3262 | 6 | 8 | -11.52 | 0.0555 | 0.3246 |
| 6 | 10 | -11.65 | 0.0619 | 0.3287 | 6 | 12 | -11.66 | 0.0520 | 0.3310 |

## B.4 Analysis of Existing Large-scale Datasets for Time Series Pre-training

**Large-scale datasets are crucial for building foundation models.** Almost all deep learning models today are data-driven, relying on training data [80, 85, 112, 113]. Therefore, when constructing a pre-trained foundation model for time series, a large-scale and comprehensively representative pre-training dataset is indispensable [28, 29, 20, 18, 98]. The scaling laws of neural networks indicate that the learning effectiveness of deep neural networks is primarily influenced by three factors: the number of model parameters, the size of the training dataset, and the amount of computational resources [5, 6, 96, 114]. Expanding the scale of the pre-training dataset can effectively improve the model's generalization capability and performance, and the performance gains from increasing data volume

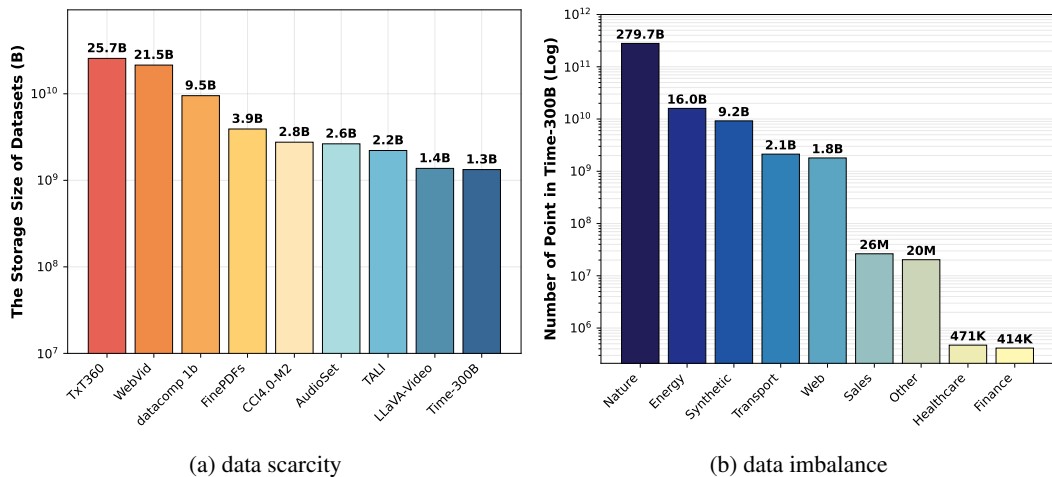

|          | (a) data scarcity | (b) data imbalance |
|----------|-------------------|--------------------|

Figure 13: The scarcity and imbalance of time series pre-training dataset (taking the largest open-source time series dataset Time-300$B$ as an example [7]). (a) time series datasets are data-scarce compared to text datasets in natural language processing and video understanding datasets in computer vision. (b) Large-scale time series pre-training datasets face serious distribution imbalance problems.

Table 11: Time-300B time series dataset from Time-MoE [7].

|        | Energy | Finance | Health   | Nature  | Sales  | Synthetic | Transport | Web    | Other  | Total   |
|--------|--------|---------|----------|---------|--------|-----------|-----------|--------|--------|---------|
| # Obs. | 15.98B | 413.70K | 471.04K  | 279.72B | 26.38M | 9.22B     | 2.13B     | 1.80B  | 20.32M | 309.09B |
| %      | 5.17%  | 5.17%   | 0.0001%  | 90.50%  | 0.008% | 2.98%     | 0.69%     | 0.58%  | 0.006% | 100%    |

Table 12: UTSD time series dataset from Timer [29], where Mise. means Multiple Sources.

|        | Energy | Environment | Health  | IoT     | Nature | Transport | Web    | Cloud  | Sales   | Finance | Mise.   |
|--------|--------|-------------|---------|---------|--------|-----------|--------|--------|---------|---------|---------|
| # Obs. | 16.86B | 70.45M      | 233.M   | 165M    | 201B   | 4.9B      | 157M   | 2.15B  | 198M    | 0.33M   | 56.52M  |
| %      | 7.461% | 0.031%      | 0.103%  | 0.073%  | 89%    | 2.17%     | 0.07%  | 0.95%  | 0.088%  | 0.00%   | 0.025%  |

Table 13: LOTSA time series dataset from Moirai [28].

|        | Energy | Transport | Climate | CloudOps | Web   | Sales | Nature  | Finance | Health | Total  |
|--------|--------|-----------|---------|----------|-------|-------|---------|---------|--------|--------|
| # Obs. | 16.36B | 4.90B     | 4.19B   | 1.52B    | 428M  | 198M  | 28.55M  | 24.92M  | 1.59M  | 27.65B |
| %      | 59.17% | 17.73%    | 15.15%  | 5.49%    | 1.55% | 0.72% | 0.09%   | 0.10%   | 0.01%  | 100%   |

Table 14: Time series datasets from neural scaling laws [6]

|        | Transport | Climate | Energy | CloudOps | Health | Sales | Web   | Total  |
|--------|-----------|---------|--------|----------|--------|-------|-------|--------|
| # Obs. | 4.82B     | 4.73B   | 2.34B  | 2.15B    | 240M   | 140M  | 600M  | 14.46B |
| %      | 33.31%    | 32.71%  | 16.15% | 14.86%   | 1.61%  | 0.96% | 0.40% | 100%   |

are independent of the model architecture and training methods [6, 115, 116, 117]. Consequently, an increasing number of models are adopting the approach of training larger-scale models on large-scale pre-training datasets to achieve better performance [118]. This paper surveys the pre-training datasets used by the three current mainstream pre-trained foundation models—Time-MoE [7], Moirai [28], and Timer [29]—as well as the datasets utilized in the study of time series scaling laws [6], which are shown in Tables 11, 12, 13 and 14. In Figure 13 (a) we demonstrate that the current largest time series datasets are still smaller than those in CV and NLP.

**Imbalanced domain distribution issues in large-scale time series datasets.** The distribution of data across various domains indicates that the four large-scale time series pre-training datasets all face issues with imbalanced domain data distribution. For instance, domains such as Nature, Energy and Transport have the most datasets [80], while others like Sales, IoT, Web, Finance and Multiple Sources suffer from extremely low data volumes due to difficulties in data collection or data privacy concerns as shown in Figure 13 (b). According to the scaling laws of neural networks, the imbalance in the pre-training dataset distribution can lead to significant performance biases in in-domain and out-of-domain forecasting tasks for the trained foundation models [6, 118], meaning there is a considerable performance gap between domains with less data and those with more data. To address this, this paper proposes an unrestricted method for generating high-quality time series data to alleviate the scarcity and imbalanced distribution of data in time series analysis domains.

## B.5  $S^2$ Dataset Statistical Characterization Coverage Experiments

**Metric.** To further examine the diversity of the artificially synthesized data in the $S^2$ dataset, we conduct a sampling assessment from six dimensions: stationarity, predictability, frequency domain characteristics, complexity, seasonality intensity, and trend characteristics. For each dimension, we select corresponding statistical indicators for dataset evaluation and quantification, as detailed below:

1. **Augmented Dickey-Fuller (ADF) Test:** Consistent with section B.3, we employ the ADF test to assess the stationarity of time series, using its test statistic as an indicator of time series stationarity [65, 29].

2. **Forecastability:** Based on [66] method, we determine whether a time series is chaotic or can be accurately predicted through machine learning models by using Fourier decomposition and entropy [29]. Note that since the method provided by [66] is only applicable to multivariate time series, we invert the sampled single-channel time series to form a dual-channel series to calculate the indicator.

3. **FFT Mean:** We utilize the average of the Fourier transform power spectrum to evaluate the frequency domain characteristics of time series. This indicator can be used to measure the overall intensity of time series and assess the energy distribution.

4. **Permutation Entropy:** This indicator assesses the dynamic complexity of a time series by analyzing its permutation patterns [69]. We set the embedding dimension $m = 3$ and time delay $\tau = 1$, and calculate its specific value using Shannon Entropy in Equation 8. See [69] for more detailed calculation.

5. **Seasonality:** We decompose the time series into trend, seasonal and residual components using the Seasonal-Trend Decomposition using LOESS (STL) algorithm [67]. Then, we calculate the intensity of the seasonal component in the time series according to Equation 9.

6. **Mann-Kendall Test:** This is a non-parametric statistical method used to detect monotonic trends in time series [68]. The basic principle is to compare the size relationship between each data point and other data points in the time series. Therefore, this method does not rely on a specific distribution of data and is not affected by outliers. We use the statistical test results of this method as the evaluation indicator, where -1 indicates a downward trend, 1 indicates an upward trend, and 0 indicates no obvious trend.

$$\text{Permutation} = -\sum_{j=1}^{K} P_j \times \ln P_j, \tag{8}$$

$$\begin{cases} Y_t = T_t + S_t + R_t \\ \text{Seasonality} = \max\left\{0, 1 - \frac{\text{Var}(R_t)}{\text{Var}(S_t + R_t)}\right\} \end{cases}, \tag{9}$$

where, $P_i$ represents the frequency of the $i$-th permutation model in the permutation entropy, and $K = m!$ is the total number of permutation patterns [69]. $Y_t$ represents the original time series, $T_t$, $S_t$ and $R_t$ are the trend, seasonal and residual components decomposed by the STL algorithm [67] respectively. $\text{Var}(\cdot)$ means calculating the variance of a series.

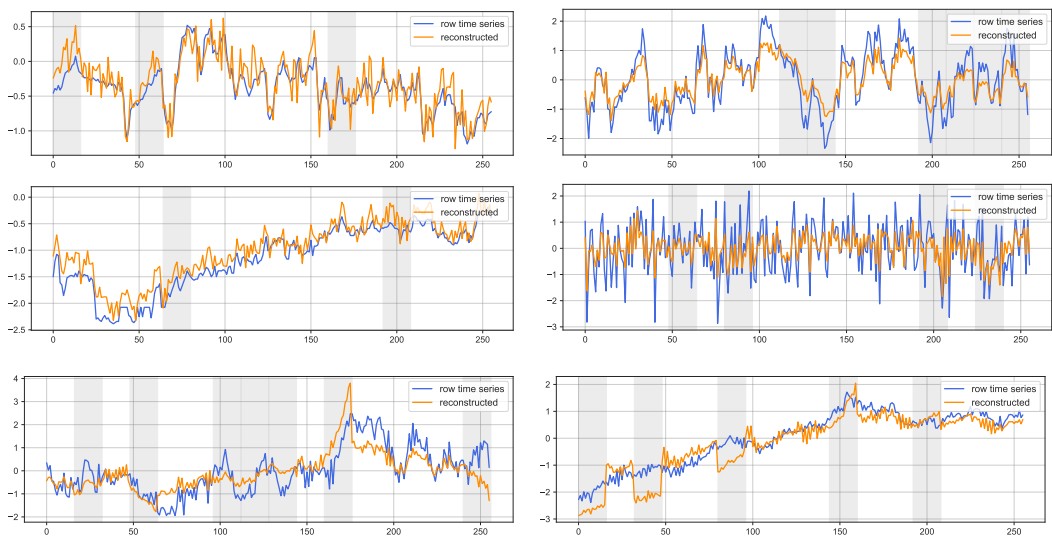

Figure 14: Zero-shot time series imputation in S² out-of-domain data.

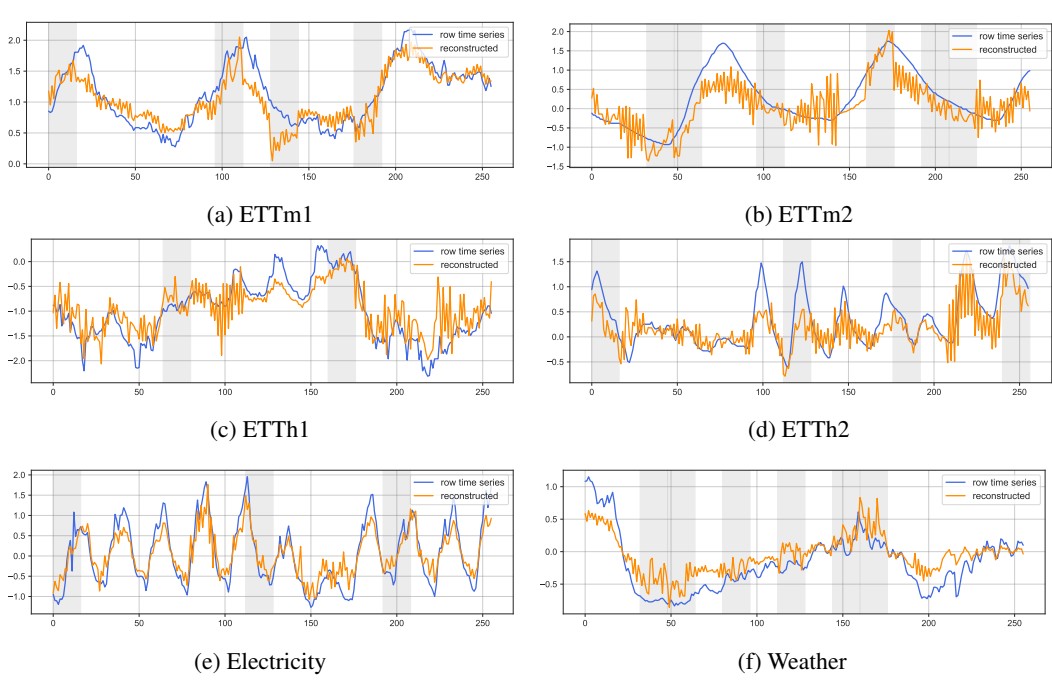

(a) ETTm1

(b) ETTm2

(c) ETTh1

(d) ETTh2

(e) Electricity

(f) Weather

Figure 15: Zero-shot time series imputation in real world time series dataset in ETTm1, ETTm2, ETTh1, ETTh2 [80], Electricity [102] and Weather [101].

### B.6 Masked Time Series Modeling and Zero-shot Imputation for Representation Learning

**Setup.** Since we incorporate MTM loss in the pre-training process of `SymTime`, in this section, we assess the specific learning effects of the time series encoder in `SymTime` through masked modeling [54, 27, 119, 89]. We test the model's performance using both pre-trained synthetic data not in the S² dataset and real datasets from time series imputation tasks [12, 14, 48]. As `SymTime` adds masks in units of patches of length 16 during pre-training, we also add masks in the form of 16-length patches. The reconstruction effect of the masked parts by the time series encoder is shown in Figure 14 and 15.

$S^2$ **Dataset Out-of-Domain Data.** In Figure 14, we generate new data using the method from the $S^2$ dataset and add masks to test the reconstruction ability of the time series encoder [12, 14, 48]. The gray sections represent the masked segments, while blue and orange represent the original and reconstructed series, respectively. We input time series outside the gray parts in patches and have the model reconstruct the gray sections based on the remaining information. Since we only calculate the MTM loss on the masked parts [27, 119], the visible reconstruction does not overlap with the original input series [29, 120, 93]. From the Figure 14, it can be observed that the time series encoder in `SymTime` performs well in fitting the fluctuations and trends of time series, demonstrating that our encoder successfully learned the fundamental representations of time series during pre-training [28, 121, 122].

**Real-world Time Series Data.** In Figure 15, we conduct representation learning tests on 6 real datasets: ETTm1, ETTm2, ETTh1, ETTh2 [80], Electricity [102], and Weather [101]. Since no real data are used for model pre-training, these datasets are also considered as out of domain data. We similarly add masks in patch units (gray sections). It can be observed that the time series encoder in `SymTime` also performs well in zero-shot reconstruction on real-world data [27, 123].

## B.7  Time Complexity Analysis of $S^2$ Data Generation Mechanism

We define the specific symbol and its explanation in Table 9. Then, we use the divide-and-conquer approach to anaylze the complexity of the $S^2$ data generation mechanism.

1. **Symbolic Expression Generation:** We construct symbolic expressions using a tree structure as a medium. When we have $b$ binary operators, we further insert $(b + 1)$ leaf nodes (the process from (a) to (b) in Figure 3 in our paper). Therefore, after inserting $u$ unary operators (Figure 3 (c)), the total number of nodes in the tree is $n = 2b + u + 1$. Because there are many ways to construct a tree, we consider the time complexity of constructing a balanced tree. Therefore, for $N$ symbols constructed, the specific complexity of this process is $\mathcal{O}(N \times n\log n)$.

2. **Sampling series generation:** When we want to generate a sampling time series with $M$ channels, each channel has a probability of $P$ to be sampled using a mixture distribution and a probability of $(1 - P)$ to be sampled using an ARMA model. When the sampling length of the series is $L$, the complexity of generating $k$ mixture distribution and ARMA ($p$, $q$) series is $O(kL)$ and $O(L(p + q))$. Therefore, the time complexity of this process can be quantified as $\mathcal{O}\left(ML \times [Pk + (1 - P)(p + q)]\right)$.

3. **Sampling through symbolic expressions and series:** We simplify the specific operational details of this process and only consider the time complexity of operations with variables. For a series of length L, we have $N$ symbolic expressions to be sampled, and each symbol has an average of $\frac{M+1}{2}$ variables (Each symbolic expression may contain any number of variables from 1 to M, so here we take $\frac{M+1}{2} = \frac{(1+2+\cdots+M)}{M}$ as the average probability). Then the process can be quantified as $\mathcal{O}(N \cdot \frac{M+1}{2} \cdot L)$.

To sum up, the symbolic expressions we construct and the parameters used in the sampling process are typically smaller than the length of the time series $L$. Therefore, we ignore the symbolic expression generation process and consider only the two processes of generating the sampling series and sampling using the symbolic expression. Since the number of channels, $M$ and $N$, as well as $k$, $p$ and $q$ are all smaller than $L$, we can intuitively assume that the time complexity of the $S^2$ data generation mechanism is linearly related to the length of the series $L$.

## B.8  The Selection of the Unary Operators

From the perspective of constructing symbolic expressions in the current $S^2$ data generation mechanism, binary operators primarily serve to connect multiple variables, while unary operators can increase the diversity of numerical values through specific operations. However, we choose not to use all symbolic and linear operations, but only use the unary operators $\{\mathrm{inv}, \mathrm{abs}, \mathrm{pow2}, \mathrm{pow3}, \mathrm{sqrt}, \sin, \cos, \tan, \arctan, \log, \exp\}$ for generation.

Although ignoring certain mathematical symbols will reduce the diversity of symbolic expressions, we have found in numerous experiments and tests that differential $\frac{\mathrm{d}y}{\mathrm{d}x}$, integral $\int$, power operations

$x^n$, and exponential operations with various bases $n^x$ will seriously affect our data generation to a certain extent. The specific reasons are as follows:

1. **Value explosion:** To maintain quality, we cap large magnitudes (Section 3.1). Integration, exponential and high-order powers readily cause overflow, so they were dropped; exp alone is retained for diversity.

2. **Numerical differentiation:** Symbolic differentiation introduces truncation/round-off trade-offs. Combined with reciprocal and absolute-value operators, functions such as $|x|$ or $\sin\left(\frac{1}{x}\right)$ because non-smooth or high-frequency vibration at $x = 0$, breaking differentiation.

3. **Numerical integration:** Randomly built expression trees often yield integrands with singularities (e.g., $\int_0^1 \frac{1}{\sqrt{x}}\mathrm{d}x$) or force costly oscillatory integrals (e.g., $\int_0^{100} sin(100x)\mathrm{d}x$); interval selection is non-trivial.

4. **Symbolic cost:** Differentiation and integration are slow and can trigger exponential memory growth. Many elementary functions lack closed-form antiderivatives (e.g., $\int \exp(-x^2)\mathrm{d}x$).

Considering factors like numerical stability, symbolic complexity, computational efficiency, and sampling success rate, we selectively omitted some symbolic operations. Nevertheless, the data generation framework proposed is essentially a complete theory. It already incorporates the vast majority of symbolic operations, and new operators or user-defined symbolic operations can be easily added. The omission of some operations due to the above factors does not undermine the validity of this framework.

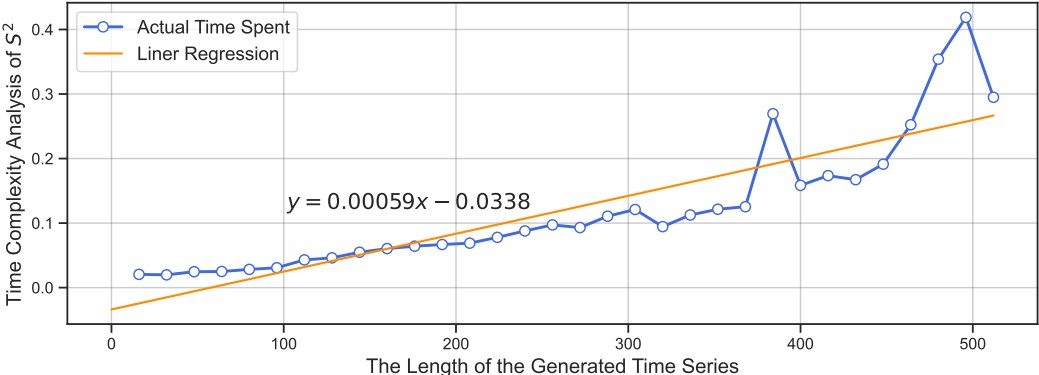

Figure 16: The time complexity analysis of $S^2$ Generation

To further demonstrate that the time complexity of the $S^2$ data generation mechanism is linearly related $\mathcal{O}(L)$ to the length of the generated time series, we start with a time series of length 16 and generate it every 16 lengths until 512. We switch to a different random seed for each generation and repeat the experiment 1280 times. The average time of each data generation is shown in Figure 16. The linear fit line of the result shows that the time complexity of data generation is linearly related to the sequence length when sampling failure is not considered.

## B.9 The Limitations of $S^2$ Generation Mechanism

As outlined in the abstract and introduction, to address the scarcity of training data for time series foundation models, this work proposes a dual-modal data generation mechanism grounded in complex dynamical systems theory (detailed in Section 3.1). This mechanism enables comprehensive coverage of time series representation spaces through unrestricted, high-quality generation. However, generating symbolic expressions (complex systems) via randomized binary tree algorithms occasionally results in oversized trees, leading to overly intricate symbolic systems. This issue narrows the domain of symbolic functions $f(\cdot)$, hinders sampling of stimulus-driven time series $X$, and reduces sampling efficiency. Notably, differential and integral operations—complex linear transformations—severely degrade sampling speed. Consequently, these operations are excluded from the current $S^2$ dataset generation. Future work will explore integrating ordinary and partial differential equations into the

$S^2$ framework to enrich symbolic expression diversity (complex systems) and further enhance the representational capacity of generated time series data.

## C  Implementation Details

In this section, we first provide a detailed introduction to the datasets and evaluation metrics used for the five TSA tasks. Subsequently, we elaborate on the training details of our experiments, including how we pre-trained `SymTime` on the $S^2$ dataset and how we fine-tuned it on downstream task datasets. All experiments and deep neural networks training are implemented in PyTorch on 8 NVIDIA A6000 48GB GPU.

### C.1  Downstream Tasks Datasets Details

We conduct experiments using the TimesNet benchmark [72], with a detailed description of the dataset provided in Table 15. Specifically, we utilize 8 datasets including ETTh1, ETTh2, ETTm1, ETTm2 [80], Electricity [102], Traffic [103], Weather [101], and Exchange [104] to conduct long-term time series forecasting experiments. Our model, `SymTime`, employ input series of lookback lengths 96 and 512, with forecast horizons of 96, 192, 336, and 720. For short-term forecasting experiments, we employ the M4 benchmark dataset, predicting data of various frequencies [73]. In the time series imputation task, we test on 6 datasets—ETTh1, ETTh2, ETTm1, ETTm2 [80], Electricity [102], and Weather [101]—with mask rates of 12.5%, 25%, 37.5%, and 50%. For time series classification, we utilize ten UEA multivariate time series classification benchmark datasets [105]. For anomaly detection in time series, we experiment with five datasets: SMD [106], MSL [107], SMAP [107], SWaT [108], and PSM [109].

### C.2  Correspondence between Positive and Negative Samples in `SymTime`

In our $S^2$ data generation mechanism, we randomly construct a symbolic expression $f(\cdot)$ and an input stimulus time series $X$, and then forward propagate the symbolic expression to obtain the complex system's response time series $Y$. Therefore, the generated time series $Y$ has a natural generating-being relationship with the symbolic expression $f(\cdot)$. Since symbolic expressions can model complex systems, and different symbolic expressions can correspond to different symbolic systems, the generated time series can also reflect the corresponding complex system's representation. When performing contrastive learning, we consider the time series and the symbolic expression that generated it as positive samples, and other unrelated symbols as negative samples.

### C.3  Metrics

We assess the five TSA tasks using various metrics. For long-term forecasting and imputation tasks, we employ mean squared error (MSE) and mean absolute error (MAE). For short-term forecasting, we utilize symmetric mean absolute percentage error (SMAPE), mean absolute scaled Error (MASE), and overall weighted average (OWA), with OWA being a metric unique to the M4 competition. For time series classification tasks, we use classification accuracy as the metric. For anomaly detection tasks, we adopt precision, recall, and F1-score as our evaluation metrics. The calculations for these metrics are as follows.

$$\text{MSE} = \sum_{i=1}^{n} (y_i - \hat{y}_i)^2, \tag{10}$$

$$\text{MAE} = \sum_{i=1}^{n} |y_i - \hat{y}_i|, \tag{11}$$

$$\text{SMAPE} = \frac{200}{T} \sum_{i=1}^{T} \frac{\left| \mathbf{X}_i - \hat{\mathbf{Y}}_i \right|}{|\mathbf{X}_i| + \left| \hat{\mathbf{Y}}_i \right|}, \tag{12}$$

$$\text{MAPE} = \frac{100}{T} \sum_{i=1}^{T} \frac{\left| \mathbf{X}_i - \hat{\mathbf{Y}}_i \right|}{|\mathbf{X}_i|}, \tag{13}$$

Table 15: Dataset descriptions. The dataset size is organized in (Train, Validation, Test).

| Tasks | Dataset | Dim | Series Length | Dataset Size | Information (Frequency) |
|---|---|---|---|---|---|
| Forecasting (Long-term) | ETTm1, ETTm2 | 7 | {96, 192, 336, 720} | (34465, 11521, 11521) | Electricity (15 mins) |
| | ETTh1, ETTh2 | 7 | {96, 192, 336, 720} | (8545, 2881, 2881) | Electricity (15 mins) |
| | Electricity | 321 | {96, 192, 336, 720} | (18317, 2633, 5261) | Electricity (Hourly) |
| | Traffic | 862 | {96, 192, 336, 720} | (12185, 1757, 3509) | Transportation (Hourly) |
| | Weather | 21 | {96, 192, 336, 720} | (36792, 5271, 10540) | Weather (10 mins) |
| | Exchange | 8 | {96, 192, 336, 720} | (5120, 665, 1422) | Exchange rate (Daily) |
| Forecasting (short-term) | M4-Yearly | 1 | 6 | (23000, 0, 23000) | Demographic |
| | M4-Quarterly | 1 | 8 | (24000, 0, 24000) | Finance |
| | M4-Monthly | 1 | 18 | (48000, 0, 48000) | Industry |
| | M4-Weakly | 1 | 13 | (359, 0, 359) | Macro |
| | M4-Daily | 1 | 14 | (4227, 0, 4227) | Micro |
| | M4-Hourly | 1 | 48 | (414, 0, 414) | Other |
| Imputation | ETTm1, ETTm2 | 7 | 96 | (34465, 11521, 11521) | Electricity (15 mins) |
| | ETTh1, ETTh2 | 7 | 96 | (8545, 2881, 2881) | Electricity (15 mins) |
| | Electricity | 321 | 96 | (18317, 2633, 5261) | Electricity (15 mins) |
| | Weather | 21 | 96 | (36792, 5271, 10540) | Weather (10 mins) |
| Classification (UEA) | EthanolConcentration | 3 | 1751 | (261, 0, 263) | Alcohol Industry |
| | FaceDetection | 144 | 62 | (5890, 0, 3524) | Face (250Hz) |
| | Handwriting | 3 | 152 | (150, 0, 850) | Handwriting |
| | Heartbeat | 61 | 405 | (204, 0, 205) | Heart Beat |
| | JapaneseVowels | 12 | 29 | (270, 0, 370) | Voice |
| | PEMS-SF | 963 | 144 | (267, 0, 173) | Transportation (Daily) |
| | SelfRegulationSCP1 | 6 | 896 | (268, 0, 293) | Health (256Hz) |
| | SelfRegulationSCP2 | 7 | 1152 | (200, 0, 180) | Health (256Hz) |
| | SpokenArabicDigits | 13 | 93 | (6599, 0, 2199) | Voice (11025Hz) |
| | UWaveGestureLibrary | 3 | 315 | (120, 0, 320) | Gesture |
| Anomaly Detection | SMD | 38 | 100 | (566724, 141681, 708420) | Server Machine |
| | MSL | 55 | 100 | (44653, 11664, 73729) | Spacecraft |
| | SMAP | 25 | 100 | (108146, 27037, 427617) | Spacecraft |
| | SWaT | 51 | 100 | (396000, 99000, 449919) | Infrastructure |
| | PSM | 25 | 100 | (105984, 26497, 87841) | Server Machine |

$$\text{MASE} = \frac{1}{T} \sum_{i=1}^{T} \frac{\left| \mathbf{X}_i - \hat{\mathbf{Y}}_i \right|}{\frac{1}{T-q} \sum_{j=q+1}^{T} \left| \mathbf{X}_j - \mathbf{X}_{j-q} \right|}, \tag{14}$$

$$\text{OWA} = \frac{1}{2} \left[ \frac{\text{SMAPE}}{\text{SMAPE}_{\text{Naïve2}}} + \frac{\text{MASE}}{\text{MASE}_{\text{Naïve2}}} \right], \tag{15}$$

Table 16: The model architecture of the time series and symbolic encoders in `SymTime`.

| Encoder | Layers | $d_{\mathrm{model}}$ | $d_{\mathrm{ff}}$ | Heads | Params |
|---|---|---|---|---|---|
| Time Series | 6 | 512 | 2048 | 8 | 19M |
| Symbol | 6 | 768 | 3072 | 12 | 67M |

where, $y_i$ is the ground true value, $\hat{y}_i$ is the model prediction, $q$ is the peridoicity of the time series data. $\mathbf{X}, \hat{\mathbf{Y}} \in \mathbb{R}^{T \times C}$ are the ground truth and prediction results of the future with $T$ time points and $C$ dimensions. $\mathbf{X}_i$ means the $i$-th future time point.

## C.4 Pre-training

**Model Architecture.** The model architecture of the time series and symbolic encoders in `SymTime` are shown in Table 16.

**Model Hyper-parameter.** The parameter configurations for the time series encoder and symbol encoder in `SymTime` are shown in Table 16. During model pre-training, we primarily set three hyperparameters: (1) the masking ratio of time series patches, (2) the masking ratio for natural language symbols, and (3) the proportion factor $\alpha$ used to balance pseudo-targets in momentum distillation. Based on the masked time series modeling pre-training experimental configuration of PatchTST [54] and SimMTM [27], we set the masking ratio for time series to 40%. Following the experimental configuration of BERT in masked language modeling [19, 55], we set the masking ratio for symbolic data to 15%. Based on the experimental configuration of momentum distillation in ALBEF [60, 124, 59], we set $\alpha$ to 0.6.

**Training Configurations.** During the pre-training of `SymTime`, we employ AdamW [125, 126] as the optimizer with the defult hyperparameter configuration for ($\beta_1$, $\beta_2$) as (0.9, 0.999). Then, we utilize the OneCycle policy to dynamically adjust the learning rate. We set the warmup epochs to 10, during which the learning rate gradually grows up to an initial value of $5 \times 10^{-5}$, and then adjust it dynamically using a cosine annealing schedule, with the minimum learning rate set at $1 \times 10^{-7}$. We conduct pre-training using data parallelism on a hardware setup consisting of 8 NVIDIA RTX A6000 GPUs with 48GB of memory each. We set the batch size to 128 and trained for a total of 85 epochs. Unlike SNIP [12], we do not generate data on-the-fly during training for pre-training. Instead, we prepare the data in advance and then load it into the device for pre-training. Due to the large size of our generat $S^2$ dataset, we load data into the GPU in batches during each epoch for pre-training.

## C.5 Fine-tuning

For the five major tasks in TSA, we conduct downstream task fine-tuning experiments using the configurations in Table 17. For all downstream task fine-tuning experiments, we employ the Adam optimizer [125, 126] with hyperparameters ($\beta_1$, $\beta_2$) set to $(0.9, 0.999)$. The LR in the table represents the initial learning rate and we utilize the dynamic learning rate adjustment strategy from TimesNet [72].

Table 17: Experiment configuration of `SymTime` fine-tuning.

| Tasks / Configurations | Model Parameter | | | Training Configurations | | | |
|---|---|---|---|---|---|---|---|
| | $d_{\mathrm{model}}$ | $d_{\mathrm{ff}}$ | Layers | LR | Loss | Batch Size | Epochs |
| Long-term Forecasting | | | 3, 6 | $10^{-4} - 5 \times 10^{-4}$ | MSE | 4-64 | 20 |
| Short-term Forecasting | | | 2, 3 | $10^{-4} - 2 \times 10^{-4}$ | SMAPE | 8-32 | 16 |
| Classification | 512 | 2048 | 1-6 | $10^{-4} - 5 \times 10^{-3}$ | Cross Entropy | 4-64 | 64 |
| Imputation | | | 2, 3, 6 | $10^{-4} - 5 \times 10^{-4}$ | MSE | 4-64 | 32 |
| Anomaly Detection | | | 3, 6 | $10^{-4} - 5 \times 10^{-4}$ | MSE | 4-64 | 12 |

## C.6 Ablation Experiments Details

**Ablation study on pre-training strategies and objectives.** To further verify the effectiveness of our series-symbol pre-training strategy and objectives, we establish 8 distinct ablation experiment groups and a control group. The specific configurations of these 8 ablation experiment groups are as follows.

1. **Freeze**: All parameters in the pre-trained time series encoder are frozen, with only the linear projection layer for outputting prediction results fine-tuned.

2. **Read-Data**: Since real time series data does not have matching symbolic expression information, we temporarily discarded the symbolic encoder and momentum model in this ablation experiment and only used real time series data for pre-training using the MTM method.

3. **w/o Pretrain**: No series-symbol pre-training is conducted; the time series encoder with initialized parameters is used for downstream task experiments.

4. **w/o MTM**: The masked time series modeling (MTM) is removed from the pre-training objectives.

5. **w/o MLM**: The masked language modeling (MLM) is removed from the pre-training objectives.

6. **w/o T2S**: The contrastive loss from time series to symbols is removed from the pre-training objectives.

7. **w/o S2T**: The contrastive loss from symbols to time series is removed from the pre-training objectives.

8. **w/o Symbol**: Only time series data from the $S^2$ dataset are used to pre-train the time series encoder via MTM, disregarding the correspondence with symbols.

9. **w/o Distill**: The contrastive loss in pre-training does not use the pseudo objective of momentum distillation.

## C.7 Ablation Experiments on Short-term Forecasting

**Setup.** We adopt the same experimental setup as in Section 4.4 to conduct ablation studies on short-term time series forecasting tasks. We first select the Yearly and Monthly sub-datasets from the M4 benchmark dataset [73] to perform ablation experiments on SymTime's pre-training objectives. We choose SMAPE as the evaluation metric and the average results with error bars are shown in Figure 17.

**Results.** Figure 17 (a) indicates that SymTime's performance drops sharply when the backbone encoder is frozen and no pre-training is conducted. When some pre-training objectives are removed, the model's performance in short-term time series forecasting also declines, but the sensitivity of performance degradation is not as pronounced as in long-term forecasting experiments. Figure 17 (a) shows that as the size of the pre-training dataset increases, SymTime's performance on the Quarterly dataset improves significantly.

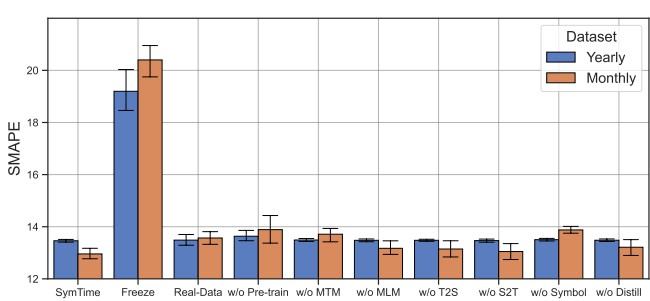

Figure 17: Ablation study on short-term forecasting task.

## C.8 The Ablation of Backbone in `SymTime`

**Setup.** `SymTime` is composed of two encoders with Transformer architectures [127]. The time series encoder is composed of a multi-layer Transformer encoder architecture. The symbolic expression encoder is pre-trained with a large language model. We conducted an ablation experiment on the

SymTime model architecture by changing the encoder structure through control variables. Specifically, we adjusted the number of parameters ($d_{\mathrm{model}}$ and $d_{\mathrm{ff}}$) of the time series encoder and the type of pre-trained LLM used by the symbolic encoder to set different control groups:

- **SymTime:** The original model architecture in Table 16 uses the DistilBERT [55].
- **SymTime$_{small}$:** Change the time series encoder to a 3-layer Transformer model with $d_{\mathrm{model}} = 386$ and $d_{\mathrm{ff}} = 1536$.
- **SymTime$_{large}$:** Change the time series encoder to a 6-layer Transformer model with $d_{\mathrm{model}} = 768$ and $d_{\mathrm{ff}} = 3072$.
- **BERT-base$_{110M}$:** Replace the pre-trained LLM with BERT-base$_{110M}$ [19].
- **BERT-large$_{340M}$:** Replace the pre-trained LLM with BERT-large$_{340M}$ [19].
- **GPT2-small$_{124M}$:** Replace the pre-trained LLM with GPT2-small$_{124M}$ [94].
- **GPT2-medium$_{335M}$:** Replace the pre-trained LLM with GPT2-medium$_{335M}$ [94].

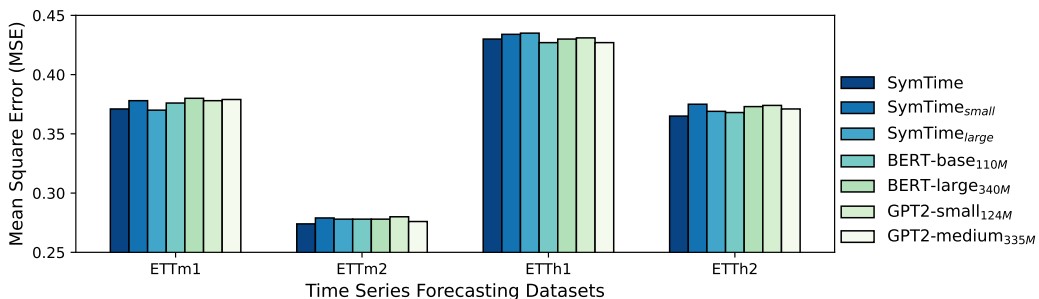

Figure 18: The ablation results of backbone in `SymTime`. We choose to conduct experimental verification on four ETT datasets [80] for long-term time series prediction.

**Results.** As shown in Figure 18, changing the backbone of `SymTime` does not significantly affect the experimental results in long-term time series forecasting tasks. Ablation experiments on pre-training objectives (Equation 7) reveal that the performance gains achieved during pre-training are primarily due to the pre-training paradigm of masked modeling of time series and symbolic expressions and contrastive learning. This pre-training approach is independent of the model backbone. Therefore, if the basic pre-training requirements are met (pre-training loss can be successfully optimized), a more lightweight model can be used for fine-tuning on downstream tasks.

### C.9 The Impact and Ablation of Pre-interpolation on Time Series Imputation Task

`SymTime` adds masks randomly at the patch level during pre-training for time series reconstruction. While in the imputation task, masks are added randomly at the data point level. Additionally, high masking rates may disrupt the original trend and periodic features of the time series. Therefore, we use Peri-midFormer's method to apply per-interpolation to the masked time series to restore the disrupted periodic features [63, 72, 89, 1, 128, 129]. It is important to note that this method is general and independent of deep learning models. The use of this method aims to further enhance the potential of deep learning models. To further verify the effectiveness and impact of the per-interpolation method, we conduct experiments on the ECL time series imputation dataset, with results shown in Table 18. Taking the ECL time series dataset 0.5 mask ratio as an example, the effect of pre-interpolation is shown in Figure 19. We perform experiments with masking rate of $\{0.125, 0.25, 0.375, 0.50\}$ and compare models such as Peri-midFormer [63], TimesNet [72], PatchTST [54], DLinear [87] and Pyraformer [130]. Per-interpolation represents the experimental results obtained using only linear interpolation. For a missing time series $x_t$ at time $t$, the method can be described as:

$$x_t = \begin{cases} \frac{x_{t-1}+x_{t+1}}{2}, & if\ (x_{t-1}) \neq \text{None}\ \&\ (x_{t+1} \neq \text{None}) \\ x_{t+1}, & if\ (x_{t-1}) = \text{None}\ \&\ (x_{t+1} \neq \text{None}) \\ x_{t-1}, & if\ (x_{t-1}) \neq \text{None}\ \&\ (x_{t+1} = \text{None}) \end{cases} \tag{16}$$

Table 18: Ablation Experiments of pre-interpolation in inputation task on ECL dataset. The per-interpolation results for Peri-midFormer, TimesNet, PatchTST, DLinear and Pyraformer are copied from [63].

| Methods | Metric | w/o per-interpolation | | | | with per-interpolation | | | |
|---|---|---|---|---|---|---|---|---|---|
| | | 0.125 | 0.25 | 0.375 | 0.5 | 0.125 | 0.25 | 0.375 | 0.5 |
| Per-interpolation | MSE | - | - | - | - | 0.086 | 0.110 | 0.149 | 0.206 |
| | MAE | - | - | - | - | 0.188 | 0.213 | 0.251 | 0.301 |
| **SymTime (Ours)** | MSE | 0.050 | 0.064 | 0.074 | 0.092 | 0.037 | 0.047 | 0.060 | 0.075 |
| | MAE | 0.145 | 0.169 | 0.181 | 0.206 | 0.122 | 0.139 | 0.160 | 0.181 |
| Peri-midFormer [63] | MSE | 0.073 | 0.092 | 0.107 | 0.122 | 0.047 | 0.053 | 0.067 | 0.085 |
| | MAE | 0.187 | 0.214 | 0.231 | 0.248 | 0.140 | 0.162 | 0.179 | 0.195 |
| TimesNet [72] | MSE | 0.088 | 0.092 | 0.096 | 0.102 | 0.081 | 0.083 | 0.086 | 0.091 |
| | MAE | 0.203 | 0.208 | 0.214 | 0.221 | 0.196 | 0.198 | 0.201 | 0.207 |
| PatchTST [54] | MSE | 0.061 | 0.072 | 0.082 | 0.097 | 0.050 | 0.059 | 0.070 | 0.087 |
| | MAE | 0.170 | 0.185 | 0.198 | 0.216 | 0.148 | 0.164 | 0.181 | 0.202 |
| DLinear [87] | MSE | 0.084 | 0.113 | 0.141 | 0.173 | 0.050 | 0.062 | 0.789 | 0.105 |
| | MAE | 0.206 | 0.243 | 0.273 | 0.303 | 0.144 | 0.164 | 0.189 | 0.225 |
| Pyraformer [130] | MSE | 0.297 | 0.294 | 0.296 | 0.299 | 0.165 | 0.165 | 0.171 | 0.173 |
| | MAE | 0.383 | 0.380 | 0.381 | 0.383 | 0.290 | 0.291 | 0.293 | 0.295 |

where, $x_{t-1}$ and $x_{t+1}$ represent the values at the previous and next time points, respectively, while None indicates a missing value. The results in Table 18 show that this method significantly improves the performance of all models in a model-independent manner.

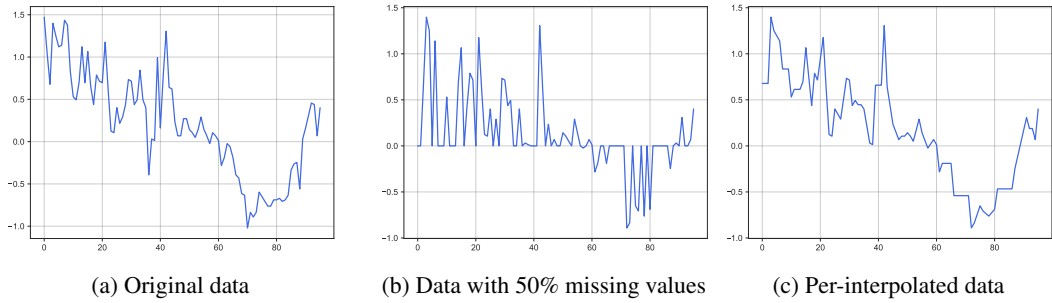

(a) Original data      (b) Data with 50% missing values      (c) Per-interpolated data

Figure 19: Visualization of original data, data with 50% missing values and pre-interpolated data of ECL dataset.

# D    Related Work

## D.1    Time Series Foundation Models

In CV and NLP [18, 131, 132], PTFMs have been demonstrated to adapt to a variety of downstream tasks after fine-tuning on specific datasets, exhibiting excellent generalization and scalability. Inspired by this, recent years have seen significant progress in PTFMs for TSA [24, 25, 133, 134], with the emergence of various pre-training methods. MOIRAI, through MTM and reconstruction, has been pre-trained on large datasets (27B), yielding a universal forecasting model with significant zero-shot advantages [28]. Timer, after generative pre-training on large datasets (1B), has performed well in forecasting [29]. TimeGPT trained a encoder-decoder Transformer with 100B data [30]. COMET, using multi-level contrastive learning on a large ECG dataset, has obtained a medical time series PTFMs with few-shot advantages [4].

As discussed in Appendix Section B.4, these baseline models still face challenges related to data scarcity and data imbalance. In the next section, we introduce the proposed data generation mechanism and the corresponding dual-modality foundation model designed to address these issues.

## D.2    Deep Learning and Symbolic Regression

The central thesis of this paper is to regard time series as representations of complex dynamical systems [135]. Traditionally, complex systems are modeled by observing time series utilizing ODE and PDE [15]. With the advancement of machine learning, symbolic regression (SR) [136], as a supervised learning method, can discover hidden mathematical expressions from numerical series. Although genetic algorithms (GAs) are the mainstream approach for SR [45, 46], deep learning-based methods have also made significant progress. [14] constructed an end-to-end SR model using Transformers, while SNIP built a large-scale pre-trained model through contrastive learning on symbolic expressions and numerical observations [12]. Both methods treat symbolic expressions as nature language and use deep neural networks to learn their features. Therefore, this paper employs a pre-trained LLM as a symbol encoder to learn the features of symbolic expressions and jointly trains a time series foundation model imbued with semantic information through contrastive learning [129].

## D.3    Time Series Forecasting Models Based on Synthetic Data

Unlike the representation pre-training conducted on the large synthetic $S^2$ dataset in this paper, previous TSA models trained on synthetic data were mainly based on Prior-data Fitted Networks (PFN) [137, 138, 139]. This model learns prior distributions from synthetic data using Bayesian methods, enabling zero-shot inference. ForecastPFN generated a large number of synthetic time series by separately modeling the seasonal trend, global trend and noise based on given constraint expressions [47]. Although PFN trained in this way offered certain zero-shot and few-shot advantages, this approach was limited to generating time series through sampling fixed expressions and performing linear combinations. In contrast, the $S^2$ data generation mechanism proposed in this paper can sample an infinite variety of symbolic expressions [12, 14, 48, 129]. TimePFN constructed synthetic datasets by filtering real time series with linear and periodic convolution kernels, training PFN for zero-shot inference [140]. However, this method depends on real-world time series for filtering and linear transformations between channels. Compared to the $S^2$ data generation mechanism, it can not create large-scale and fully representative synthetic datasets for model pre-training.

# E   Visualization

## E.1   Long-term Time Series Forecasting with 96 Prediction on ETTh1 (Figure 20) and ECL (Figure 21)

## E.2   Short-term Time Series Forecasting on M4 Weekly (Figure 22) and Monthly (Figure 23)

## E.3   Time Series Imputation with 50% mask rate on ETTh1 (Figure 24) and ETTm1 (Figure 25)

# F   Full Results

For the five downstream TSA tasks results, we use (1) Peri-midFormer for Peri-midFormer [63], (2) uni2ts for Moirai [28], (3) Large-Time-Series-Model for Timer [29], (4) Time-LLM for Time-LLM [17], (5) TSLANet for TSLANet [83], (6) S2IP-LLM for S2IP-LLM [82], (7) NeurIPS²023-One-Fits-All for GPT4TS [2], (8) UniTS for UniTS [92], (9) moment for Moment [93], (10) FilterNet for FilterNet [91], (11) RTSF for RLinear [141], and (12) Time-Series-Library for other models, such as TimesNet [72], PatchTST [54], TimeMixer [89], iTransformer [74], DLinear [87], Autoformer [75] and Informer [80], TimeXer [62], Chronos-forecasting for Chronos [22]. To ensure a fair comparison, we use the original experimental configuration in the project scripts.

## F.1   Time Series Long-term Forecasting with 96 look-back windows (Table 19, Table 20 and Table 21)

## F.2   Time Series Long-term Forecasting with 336 look-back windows (Table 22)

## F.3   Time Series Long-term Forecasting with 512 look-back windows (Table 23)

## F.4   Time Series Short-term Forecasting (Table 24 and Table 25)

## F.5   Time Series Classification (Table 26 and Table 27)

## F.6   Time Series Imputation (Table 28 and Table 29)

## F.7   Time Series Anomaly Detection (Table 30)

## F.8   Pre-training and Fine-tuning Results of Long-term Forecasting (Table 31)

## F.9   Pre-training and Fine-tuning Results of Short-term Forecasting (Table 32)

## F.10   Pre-training and Fine-tuning Results of classification (Table 33)

## F.11   Pre-training and Fine-tuning Results of Imputation (Table 34)

## F.12   Pre-training and Fine-tuning Results of Anomaly Detection (Table 35)

# G   Impact Statement

The potential value of this work lies in its ability to mitigate fundamental challenges in TSA, such as the lack of sufficient labeled data and the issue of imbalanced datasets. By generating rich, diverse, and high-quality synthetic data, our approach not only addresses these issues but also opens new avenues for improving model generalization across a wide range of applications. Furthermore, the dual-modality framework, which combines time series data with symbolic semantics, introduces a novel way of enriching the representation power of models, allowing them to better understand complex temporal dynamics and their underlying patterns.

We foresee that pre-training models on synthetic datasets, especially those that combine structured symbolic information with time series data, will become a key development trend in the TSA field. This could pave the way for more robust and scalable solutions in a variety of domains, including finance, healthcare, and climate modeling, where time series data is abundant, but labeled data is often scarce or hard to obtain.

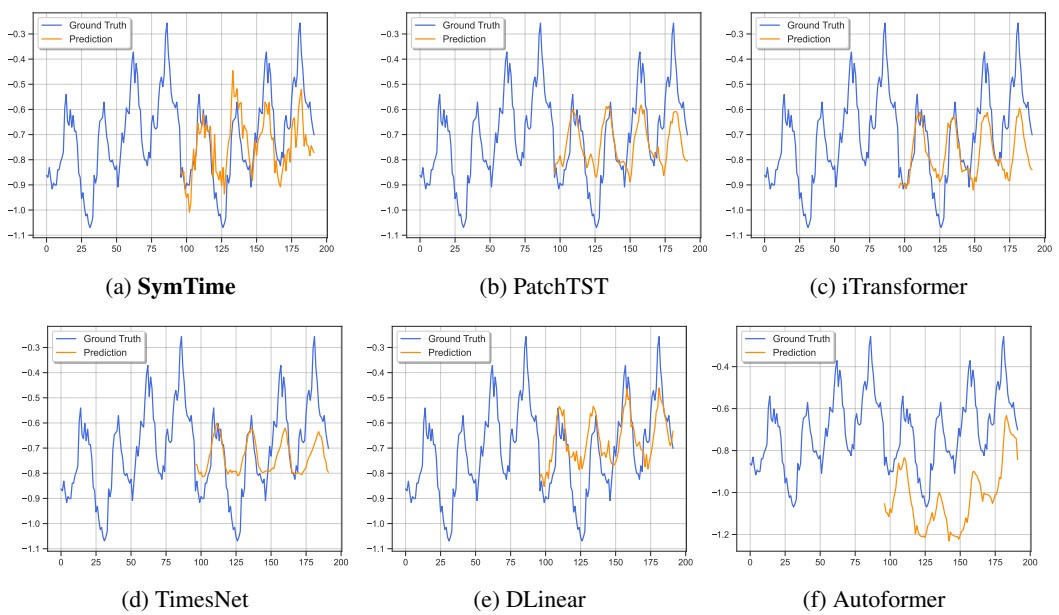

Figure 20: Visualization of long-term forecasting with 96 prediction length of ETTh1 dataset.

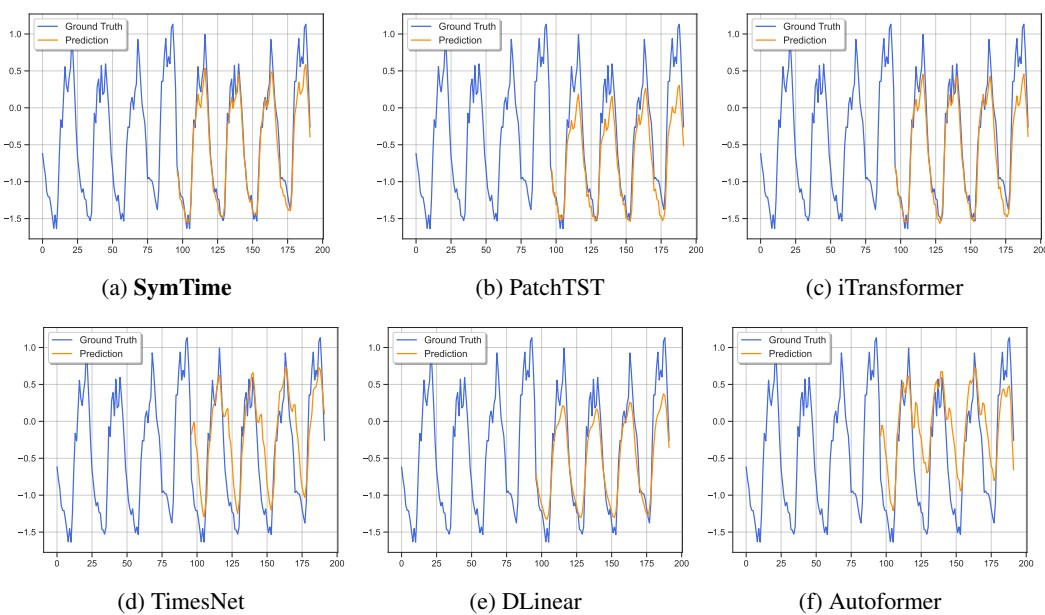

Figure 21: Visualization of long-term forecasting with 96 prediction length of Electricity dataset.

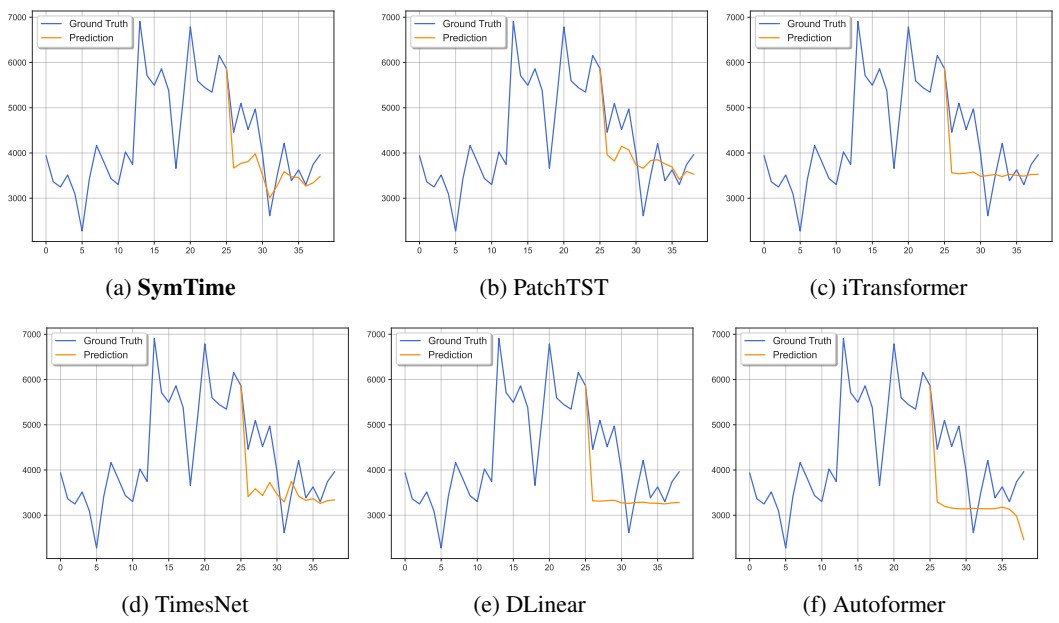

Figure 22: Visualization of time series short-term forecasting in M4 dataset Weekly.

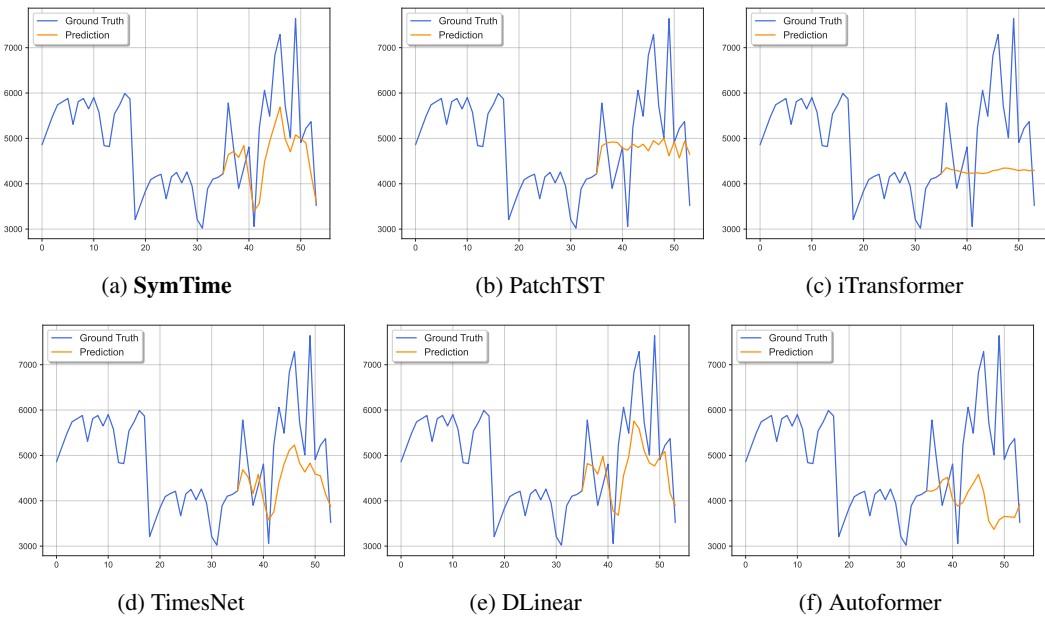

Figure 23: Visualization of time series short-term forecasting in M4 dataset Monthly.

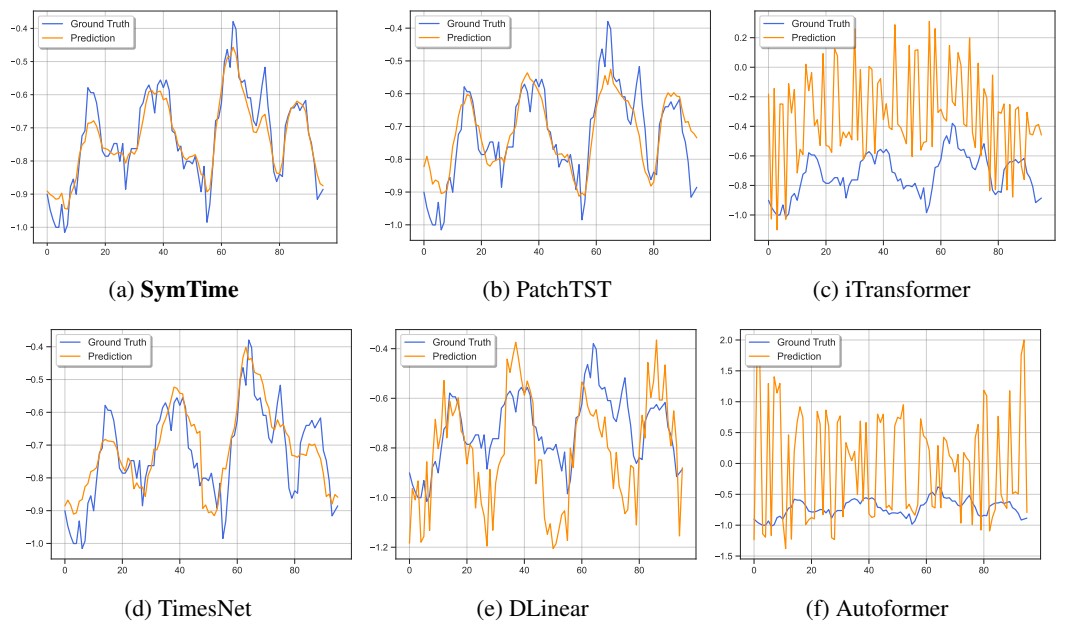

Figure 24: Visualization of time series imputation with 50% mask rate of ETTh1 dataset.

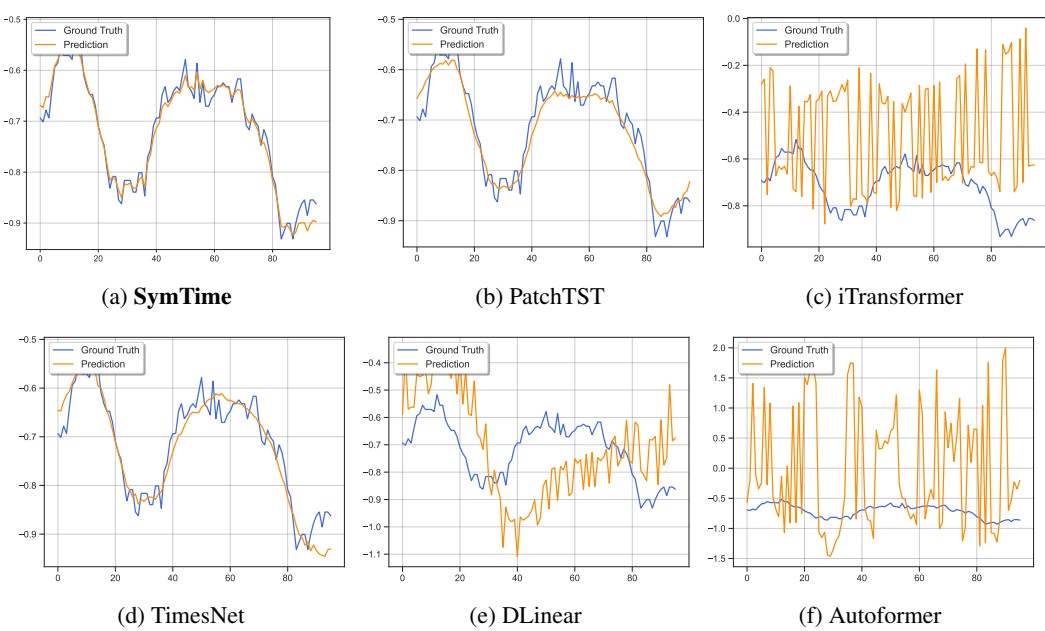

Figure 25: Visualization of time series imputation with 50% mask rate of ETTm1 dataset.

Table 19: Full results for the long-term forecasting task compared with Peri-midFormer [63], Moirai [28], Timer [29], Moment [93], Time-LLM [17], TSLANet [83], $S^2$IP-LLM [82] and GPT4TS [2]. (* means former, T-LLM is Time-LLM, S-LLM is $S^2$IP-LLM.) To ensure fairness in the comparison, we set the look-back window length of all models to **96**. Since the Timer and Moirai need to input a longer series to build a token, their windows are **672**. $S^2$IP-LLM has a gradient explosion when the window is **96**, so its look-back window is **512**. The standard deviation is within 0.5%. **Red**: best, Blue: second best.

| Methods | | SymTime (Ours) | | Peri-mid* [63] | | Moirai [28] | | Timer [29] | | Moment [93] | | T-LLM [17] | | TSLANet [83] | | S-LLM [82] | | GPT4TS [2] | |
|---|---|---|---|---|---|---|---|---|---|---|---|---|---|---|---|---|---|---|---|
| Metrics | | MSE | MAE | MSE | MAE | MSE | MAE | MSE | MAE | MSE | MAE | MSE | MAE | MSE | MAE | MSE | MAE | MSE | MAE |
| ETTm1 | 96 | 0.318 | 0.353 | 0.334 | 0.370 | 0.311 | 0.358 | 0.315 | 0.354 | 0.305 | 0.353 | 0.304 | 0.359 | 0.321 | 0.362 | 0.325 | 0.371 | 0.293 | 0.362 |
| | 192 | 0.362 | 0.380 | 0.382 | 0.391 | 0.381 | 0.402 | 0.369 | 0.378 | 0.369 | 0.386 | 0.368 | 0.396 | 0.361 | 0.383 | 0.361 | 0.397 | 0.374 | 0.392 |
| | 336 | 0.386 | 0.402 | 0.417 | 0.418 | 0.436 | 0.432 | 0.425 | 0.428 | 0.392 | 0.404 | 0.383 | 0.393 | 0.383 | 0.404 | 0.385 | 0.403 | 0.389 | 0.404 |
| | 720 | 0.419 | 0.423 | 0.501 | 0.461 | 0.466 | 0.476 | 0.442 | 0.447 | 0.425 | 0.436 | 0.420 | 0.429 | 0.445 | 0.437 | 0.426 | 0.446 | 0.421 | 0.423 |
| | Avg | 0.371 | **0.390** | 0.409 | 0.410 | 0.398 | 0.417 | 0.388 | 0.402 | 0.373 | 0.395 | **0.369** | 0.394 | 0.377 | 0.397 | 0.374 | 0.404 | 0.369 | 0.395 |
| ETTm2 | 96 | 0.174 | 0.257 | 0.174 | 0.255 | 0.179 | 0.267 | 0.168 | 0.254 | 0.170 | 0.264 | 0.177 | 0.269 | 0.179 | 0.261 | 0.174 | 0.263 | 0.171 | 0.265 |
| | 192 | 0.238 | 0.299 | 0.249 | 0.305 | 0.244 | 0.311 | 0.429 | 0.425 | 0.285 | 0.294 | 0.239 | 0.305 | 0.243 | 0.303 | 0.232 | 0.306 | 0.226 | 0.304 |
| | 336 | 0.295 | 0.337 | 0.319 | 0.349 | 0.335 | 0.371 | 0.476 | 0.457 | 0.275 | 0.329 | 0.301 | 0.340 | 0.308 | 0.345 | 0.300 | 0.344 | 0.288 | 0.345 |
| | 720 | 0.390 | 0.392 | 0.418 | 0.405 | 0.425 | 0.444 | 0.545 | 0.497 | 0.383 | 0.397 | 0.382 | 0.381 | 0.403 | 0.401 | 0.359 | 0.386 | 0.372 | 0.398 |
| | Avg | 0.274 | **0.321** | 0.290 | 0.328 | 0.296 | 0.348 | 0.405 | 0.408 | 0.278 | 0.321 | 0.275 | 0.324 | 0.283 | 0.327 | 0.266 | 0.325 | **0.264** | 0.328 |
| ETTh1 | 96 | 0.376 | 0.400 | 0.382 | 0.403 | 0.369 | 0.408 | 0.374 | 0.404 | 0.385 | 0.402 | 0.386 | 0.395 | 0.387 | 0.405 | 0.380 | 0.403 | 0.388 | 0.399 |
| | 192 | 0.428 | 0.431 | 0.436 | 0.435 | 0.441 | 0.450 | 0.430 | 0.438 | 0.449 | 0.450 | 0.421 | 0.424 | 0.448 | 0.436 | 0.410 | 0.427 | 0.425 | 0.429 |
| | 336 | 0.463 | 0.456 | 0.492 | 0.455 | 0.469 | 0.469 | 0.458 | 0.453 | 0.455 | 0.472 | 0.438 | 0.450 | 0.451 | 0.437 | 0.426 | 0.442 | 0.444 | 0.455 |
| | 720 | 0.450 | 0.458 | 0.508 | 0.490 | 0.486 | 0.490 | 0.475 | 0.480 | 0.480 | 0.503 | 0.506 | 0.510 | 0.505 | 0.485 | 0.610 | 0.543 | 0.479 | 0.477 |
| | Avg | **0.430** | **0.436** | 0.455 | 0.446 | 0.441 | 0.454 | 0.434 | 0.444 | 0.442 | 0.457 | 0.438 | 0.445 | 0.448 | 0.441 | 0.456 | 0.454 | 0.434 | 0.440 |
| ETTh2 | 96 | 0.293 | 0.347 | 0.312 | 0.358 | 0.288 | 0.350 | 0.315 | 0.360 | 0.285 | 0.343 | 0.307 | 0.369 | 0.289 | 0.345 | 0.292 | 0.353 | 0.292 | 0.351 |
| | 192 | 0.364 | 0.397 | 0.388 | 0.403 | 0.390 | 0.426 | 0.411 | 0.423 | 0.368 | 0.403 | 0.349 | 0.384 | 0.362 | 0.391 | 0.355 | 0.388 | 0.351 | 0.394 |
| | 336 | 0.385 | 0.423 | 0.443 | 0.443 | 0.441 | 0.435 | 0.465 | 0.467 | 0.380 | 0.421 | 0.394 | 0.420 | 0.350 | 0.389 | 0.368 | 0.417 | 0.380 | 0.421 |
| | 720 | 0.420 | 0.441 | 0.455 | 0.459 | 0.487 | 0.435 | 0.521 | 0.515 | 0.423 | 0.466 | 0.426 | 0.454 | 0.418 | 0.439 | 0.434 | 0.460 | 0.424 | 0.446 |
| | Avg | 0.365 | 0.402 | 0.400 | 0.416 | 0.402 | 0.411 | 0.428 | 0.441 | 0.364 | 0.408 | 0.369 | 0.407 | **0.355** | **0.391** | 0.362 | 0.405 | 0.359 | 0.403 |
| Weather | 96 | 0.166 | 0.213 | 0.157 | 0.201 | 0.156 | 0.206 | 0.289 | 0.331 | 0.168 | 0.228 | 0.172 | 0.221 | 0.177 | 0.216 | 0.162 | 0.213 | 0.184 | 0.224 |
| | 192 | 0.212 | 0.254 | 0.244 | 0.273 | 0.229 | 0.274 | 0.314 | 0.349 | 0.226 | 0.262 | 0.194 | 0.241 | 0.226 | 0.258 | 0.197 | 0.246 | 0.230 | 0.263 |
| | 336 | 0.267 | 0.294 | 0.283 | 0.303 | 0.282 | 0.316 | 0.339 | 0.363 | 0.257 | 0.303 | 0.286 | 0.282 | 0.279 | 0.588 | 0.281 | 0.299 | 0.285 | 0.302 |
| | 720 | 0.342 | 0.344 | 0.364 | 0.355 | 0.395 | 0.401 | 0.375 | 0.388 | 0.331 | 0.355 | 0.337 | 0.332 | 0.355 | 0.346 | 0.333 | 0.339 | 0.362 | 0.352 |
| | Avg | 0.247 | 0.276 | 0.262 | 0.283 | 0.265 | 0.299 | 0.329 | 0.358 | 0.245 | 0.287 | 0.247 | **0.269** | 0.259 | 0.352 | **0.243** | 0.274 | 0.265 | 0.285 |
| ECL | 96 | 0.162 | 0.253 | 0.151 | 0.245 | 0.137 | 0.221 | 0.150 | 0.244 | 0.153 | 0.247 | 0.149 | 0.242 | 0.176 | 0.261 | 0.149 | 0.251 | 0.186 | 0.272 |
| | 192 | 0.173 | 0.264 | 0.168 | 0.259 | 0.158 | 0.243 | 0.159 | 0.252 | 0.166 | 0.252 | 0.167 | 0.261 | 0.182 | 0.268 | 0.171 | 0.269 | 0.190 | 0.277 |
| | 336 | 0.194 | 0.285 | 0.184 | 0.268 | 0.167 | 0.255 | 0.190 | 0.271 | 0.172 | 0.269 | 0.188 | 0.270 | 0.199 | 0.285 | 0.199 | 0.291 | 0.205 | 0.292 |
| | 720 | 0.220 | 0.304 | 0.207 | 0.297 | 0.207 | 0.290 | 0.210 | 0.300 | 0.213 | 0.311 | 0.214 | 0.301 | 0.240 | 0.317 | 0.244 | 0.319 | 0.245 | 0.323 |
| | Avg | 0.187 | 0.276 | 0.178 | 0.267 | **0.167** | **0.252** | 0.177 | 0.267 | 0.176 | 0.270 | 0.180 | 0.269 | 0.199 | 0.283 | 0.191 | 0.283 | 0.206 | 0.291 |
| Traffic | 96 | 0.432 | 0.280 | 0.426 | 0.277 | 0.376 | 0.264 | 0.391 | 0.260 | 0.442 | 0.295 | 0.424 | 0.295 | 0.398 | 0.291 | 0.385 | 0.289 | 0.471 | 0.312 |
| | 192 | 0.444 | 0.287 | 0.440 | 0.283 | 0.410 | 0.279 | 0.426 | 0.271 | 0.452 | 0.301 | 0.455 | 0.315 | 0.430 | 0.307 | 0.403 | 0.308 | 0.478 | 0.312 |
| | 336 | 0.458 | 0.293 | 0.477 | 0.311 | 0.442 | 0.287 | 0.451 | 0.297 | 0.467 | 0.309 | 0.494 | 0.335 | 0.494 | 0.312 | 0.425 | 0.299 | 0.493 | 0.319 |
| | 720 | 0.492 | 0.303 | 0.487 | 0.308 | 0.470 | 0.328 | 0.475 | 0.307 | 0.489 | 0.316 | 0.513 | 0.394 | 0.528 | 0.332 | 0.454 | 0.326 | 0.523 | 0.335 |
| | Avg | 0.457 | 0.291 | 0.458 | 0.295 | 0.424 | 0.289 | 0.436 | **0.284** | 0.463 | 0.305 | 0.471 | 0.334 | 0.463 | 0.310 | **0.417** | 0.306 | 0.491 | 0.320 |
| Exchange | 96 | 0.084 | 0.201 | 0.083 | 0.199 | 0.089 | 0.211 | 0.098 | 0.228 | 0.091 | 0.214 | 0.090 | 0.209 | 0.082 | 0.200 | 0.147 | 0.279 | 0.087 | 0.218 |
| | 192 | 0.174 | 0.295 | 0.190 | 0.307 | 0.175 | 0.289 | 0.196 | 0.325 | 0.185 | 0.307 | 0.188 | 0.310 | 0.172 | 0.295 | 0.234 | 0.354 | 0.171 | 0.294 |
| | 336 | 0.331 | 0.416 | 0.401 | 0.458 | 0.345 | 0.423 | 0.359 | 0.433 | 0.345 | 0.414 | 0.342 | 0.427 | 0.329 | 0.415 | 0.403 | 0.474 | 0.349 | 0.418 |
| | 720 | 0.847 | 0.694 | 0.879 | 0.702 | 0.882 | 0.744 | 0.875 | 0.713 | 0.874 | 0.729 | 0.885 | 0.707 | 0.889 | 0.747 | 1.103 | 0.804 | 0.873 | 0.713 |
| | Avg | **0.359** | **0.401** | 0.388 | 0.417 | 0.373 | 0.417 | 0.382 | 0.425 | 0.374 | 0.416 | 0.376 | 0.414 | 0.368 | 0.414 | 0.472 | 0.478 | 0.370 | 0.411 |
| Average | | **0.336** | **0.349** | 0.355 | 0.358 | 0.346 | 0.361 | 0.372 | 0.378 | 0.339 | 0.357 | 0.341 | 0.357 | 0.344 | 0.364 | 0.348 | 0.366 | 0.345 | 0.359 |

Table 20: Full results for the long-term forecasting task compared with FilterNet [91], TimesNet [72], iTransformer [74], PatchTST [54], RLinear [141], DLinear [87] and TimeMixer [89]. (* means former, TNet is TimesNet, PTST is PatchTST, TMixer is TimeMixer) To ensure fairness in the comparison, we set the look-back window length of all models to **96**. The standard deviation is within 0.5%. **Red**: best, Blue: second best.

| Methods | | SymTime (Ours) | | FilterNet [91] | | Chronos [22] | | TNet [72] | | iTrans* [74] | | PTST [54] | | RLinear [141] | | DLinear [87] | | TMixer [89] | |
|---|---|---|---|---|---|---|---|---|---|---|---|---|---|---|---|---|---|---|---|
| Metrics | | MSE | MAE | MSE | MAE | MSE | MAE | MSE | MAE | MSE | MAE | MSE | MAE | MSE | MAE | MSE | MAE | MSE | MAE |
| ETTm1 | 96 | 0.318 | 0.353 | 0.321 | 0.361 | 0.324 | 0.371 | 0.331 | 0.372 | 0.343 | 0.377 | 0.324 | 0.365 | 0.355 | 0.376 | 0.345 | 0.372 | 0.323 | 0.361 |
| | 192 | 0.362 | 0.380 | 0.367 | 0.387 | 0.381 | 0.400 | 0.397 | 0.402 | 0.381 | 0.395 | 0.367 | 0.389 | 0.387 | 0.392 | 0.382 | 0.391 | 0.362 | 0.383 |
| | 336 | 0.386 | 0.402 | 0.401 | 0.409 | 0.402 | 0.415 | 0.427 | 0.427 | 0.419 | 0.418 | 0.400 | 0.409 | 0.424 | 0.415 | 0.414 | 0.414 | 0.388 | 0.403 |
| | 720 | 0.419 | 0.423 | 0.477 | 0.448 | 0.428 | 0.435 | 0.493 | 0.463 | 0.487 | 0.457 | 0.460 | 0.445 | 0.487 | 0.450 | 0.473 | 0.450 | 0.454 | 0.442 |
| | Avg | 0.371 | 0.390 | 0.392 | 0.401 | 0.384 | 0.405 | 0.412 | 0.416 | 0.407 | 0.412 | 0.388 | 0.402 | 0.413 | 0.408 | 0.403 | 0.407 | 0.382 | 0.397 |
| ETTm2 | 96 | 0.174 | 0.257 | 0.175 | 0.258 | 0.192 | 0.265 | 0.185 | 0.265 | 0.185 | 0.271 | 0.182 | 0.266 | 0.182 | 0.265 | 0.194 | 0.293 | 0.177 | 0.259 |
| | 192 | 0.238 | 0.299 | 0.240 | 0.301 | 0.268 | 0.320 | 0.256 | 0.310 | 0.254 | 0.314 | 0.250 | 0.311 | 0.246 | 0.304 | 0.283 | 0.360 | 0.245 | 0.306 |
| | 336 | 0.295 | 0.337 | 0.311 | 0.347 | 0.289 | 0.341 | 0.314 | 0.345 | 0.315 | 0.352 | 0.313 | 0.350 | 0.307 | 0.342 | 0.376 | 0.423 | 0.298 | 0.338 |
| | 720 | 0.390 | 0.392 | 0.414 | 0.405 | 0.392 | 0.410 | 0.424 | 0.412 | 0.413 | 0.407 | 0.417 | 0.412 | 0.407 | 0.398 | 0.529 | 0.509 | 0.395 | 0.396 |
| | Avg | 0.274 | 0.321 | 0.285 | 0.328 | 0.286 | 0.334 | 0.295 | 0.333 | 0.292 | 0.336 | 0.291 | 0.335 | 0.286 | 0.327 | 0.346 | 0.396 | 0.279 | 0.325 |
| ETTh1 | 96 | 0.376 | 0.400 | 0.382 | 0.402 | 0.408 | 0.402 | 0.409 | 0.425 | 0.394 | 0.409 | 0.381 | 0.400 | 0.386 | 0.395 | 0.396 | 0.411 | 0.385 | 0.400 |
| | 192 | 0.428 | 0.431 | 0.430 | 0.429 | 0.459 | 0.450 | 0.469 | 0.460 | 0.447 | 0.440 | 0.429 | 0.433 | 0.437 | 0.424 | 0.446 | 0.441 | 0.441 | 0.431 |
| | 336 | 0.463 | 0.456 | 0.472 | 0.451 | 0.445 | 0.437 | 0.507 | 0.478 | 0.491 | 0.464 | 0.475 | 0.460 | 0.479 | 0.446 | 0.490 | 0.468 | 0.482 | 0.450 |
| | 720 | 0.450 | 0.458 | 0.481 | 0.473 | 0.482 | 0.485 | 0.521 | 0.497 | 0.517 | 0.501 | 0.517 | 0.502 | 0.481 | 0.470 | 0.514 | 0.511 | 0.504 | 0.482 |
| | Avg | 0.430 | 0.436 | 0.441 | 0.439 | 0.449 | 0.444 | 0.476 | 0.465 | 0.462 | 0.454 | 0.451 | 0.449 | 0.446 | 0.434 | 0.461 | 0.458 | 0.453 | 0.441 |
| ETTh2 | 96 | 0.293 | 0.347 | 0.293 | 0.343 | 0.299 | 0.354 | 0.331 | 0.372 | 0.300 | 0.350 | 0.301 | 0.351 | 0.318 | 0.363 | 0.348 | 0.401 | 0.293 | 0.343 |
| | 192 | 0.364 | 0.397 | 0.374 | 0.396 | 0.356 | 0.390 | 0.429 | 0.423 | 0.380 | 0.399 | 0.374 | 0.398 | 0.401 | 0.412 | 0.473 | 0.474 | 0.376 | 0.396 |
| | 336 | 0.385 | 0.423 | 0.417 | 0.430 | 0.376 | 0.423 | 0.450 | 0.451 | 0.422 | 0.432 | 0.429 | 0.439 | 0.436 | 0.442 | 0.588 | 0.539 | 0.425 | 0.432 |
| | 720 | 0.420 | 0.441 | 0.449 | 0.460 | 0.439 | 0.467 | 0.459 | 0.466 | 0.429 | 0.447 | 0.443 | 0.461 | 0.442 | 0.454 | 0.829 | 0.656 | 0.457 | 0.459 |
| | Avg | 0.365 | 0.402 | 0.383 | 0.407 | 0.368 | 0.408 | 0.417 | 0.428 | 0.383 | 0.407 | 0.387 | 0.412 | 0.399 | 0.418 | 0.559 | 0.518 | 0.388 | 0.408 |
| Weather | 96 | 0.166 | 0.213 | 0.162 | 0.207 | 0.177 | 0.231 | 0.171 | 0.222 | 0.176 | 0.215 | 0.177 | 0.219 | 0.192 | 0.232 | 0.197 | 0.258 | 0.172 | 0.220 |
| | 192 | 0.212 | 0.254 | 0.210 | 0.250 | 0.221 | 0.263 | 0.234 | 0.273 | 0.226 | 0.258 | 0.222 | 0.258 | 0.240 | 0.271 | 0.237 | 0.296 | 0.227 | 0.259 |
| | 336 | 0.267 | 0.294 | 0.265 | 0.290 | 0.265 | 0.311 | 0.284 | 0.306 | 0.281 | 0.299 | 0.281 | 0.299 | 0.292 | 0.307 | 0.282 | 0.332 | 0.266 | 0.294 |
| | 720 | 0.342 | 0.344 | 0.342 | 0.340 | 0.339 | 0.347 | 0.358 | 0.352 | 0.359 | 0.350 | 0.356 | 0.348 | 0.364 | 0.353 | 0.347 | 0.385 | 0.346 | 0.347 |
| | Avg | 0.247 | 0.276 | 0.245 | 0.272 | 0.251 | 0.288 | 0.262 | 0.288 | 0.260 | 0.281 | 0.259 | 0.281 | 0.272 | 0.291 | 0.266 | 0.318 | 0.253 | 0.280 |
| ECL | 96 | 0.162 | 0.253 | 0.147 | 0.245 | 0.157 | 0.249 | 0.167 | 0.271 | 0.148 | 0.240 | 0.180 | 0.272 | 0.201 | 0.281 | 0.210 | 0.302 | 0.157 | 0.249 |
| | 192 | 0.173 | 0.264 | 0.160 | 0.250 | 0.193 | 0.288 | 0.186 | 0.288 | 0.165 | 0.256 | 0.188 | 0.279 | 0.201 | 0.283 | 0.210 | 0.305 | 0.170 | 0.261 |
| | 336 | 0.194 | 0.285 | 0.173 | 0.267 | 0.213 | 0.304 | 0.203 | 0.304 | 0.179 | 0.271 | 0.204 | 0.296 | 0.215 | 0.298 | 0.223 | 0.319 | 0.186 | 0.276 |
| | 720 | 0.220 | 0.304 | 0.210 | 0.309 | 0.255 | 0.336 | 0.227 | 0.322 | 0.209 | 0.298 | 0.246 | 0.328 | 0.257 | 0.331 | 0.258 | 0.350 | 0.227 | 0.311 |
| | Avg | 0.187 | 0.276 | 0.173 | 0.268 | 0.204 | 0.294 | 0.196 | 0.296 | 0.175 | 0.267 | 0.204 | 0.294 | 0.219 | 0.298 | 0.225 | 0.319 | 0.185 | 0.274 |
| Traffic | 96 | 0.432 | 0.280 | 0.430 | 0.294 | 0.420 | 0.294 | 0.589 | 0.316 | 0.393 | 0.268 | 0.461 | 0.298 | 0.649 | 0.389 | 0.696 | 0.429 | 0.479 | 0.299 |
| | 192 | 0.444 | 0.287 | 0.452 | 0.307 | 0.436 | 0.306 | 0.616 | 0.328 | 0.413 | 0.277 | 0.467 | 0.301 | 0.601 | 0.366 | 0.647 | 0.407 | 0.490 | 0.303 |
| | 336 | 0.458 | 0.293 | 0.470 | 0.316 | 0.491 | 0.315 | 0.628 | 0.333 | 0.424 | 0.283 | 0.483 | 0.308 | 0.609 | 0.369 | 0.653 | 0.410 | 0.493 | 0.304 |
| | 720 | 0.492 | 0.303 | 0.498 | 0.323 | 0.526 | 0.330 | 0.667 | 0.352 | 0.458 | 0.300 | 0.517 | 0.325 | 0.647 | 0.387 | 0.695 | 0.429 | 0.534 | 0.319 |
| | Avg | 0.457 | 0.291 | 0.463 | 0.310 | 0.468 | 0.312 | 0.625 | 0.332 | 0.422 | 0.282 | 0.482 | 0.308 | 0.627 | 0.378 | 0.673 | 0.419 | 0.499 | 0.306 |
| Exchange | 96 | 0.084 | 0.201 | 0.091 | 0.211 | 0.090 | 0.207 | 0.115 | 0.246 | 0.094 | 0.216 | 0.088 | 0.205 | 0.093 | 0.217 | 0.093 | 0.226 | 0.091 | 0.210 |
| | 192 | 0.174 | 0.295 | 0.186 | 0.305 | 0.190 | 0.316 | 0.213 | 0.335 | 0.185 | 0.307 | 0.189 | 0.309 | 0.184 | 0.307 | 0.184 | 0.324 | 0.185 | 0.304 |
| | 336 | 0.331 | 0.416 | 0.380 | 0.449 | 0.354 | 0.419 | 0.367 | 0.440 | 0.336 | 0.422 | 0.327 | 0.415 | 0.351 | 0.432 | 0.328 | 0.436 | 0.361 | 0.435 |
| | 720 | 0.847 | 0.694 | 0.896 | 0.712 | 0.892 | 0.716 | 0.978 | 0.753 | 0.893 | 0.716 | 0.886 | 0.706 | 0.886 | 0.714 | 0.880 | 0.705 | 0.974 | 0.741 |
| | Avg | 0.359 | 0.401 | 0.388 | 0.419 | 0.381 | 0.415 | 0.418 | 0.443 | 0.377 | 0.415 | 0.373 | 0.409 | 0.379 | 0.418 | 0.371 | 0.423 | 0.403 | 0.423 |
| Average | | 0.336 | 0.349 | 0.346 | 0.356 | 0.349 | 0.363 | 0.388 | 0.375 | 0.347 | 0.357 | 0.354 | 0.361 | 0.380 | 0.371 | 0.413 | 0.407 | 0.355 | 0.357 |

Table 21: Full results for the long-term forecasting task compared with Autoformer [75], Crossformer [79], FEDformer [77], ETSforemr [76], Stationary [78], LightTS [88], Informer [80]. (Stationary means Nonstationary Transformer. * means former) To ensure fairness in the comparison, we set the look-back window length of all models to **96**. The standard deviation is within 0.5%. **Red**: best, Blue: second best.

| Methods | | SymTime (Ours) | | Autoformer [75] | | Cross* [79] | | FED* [77] | | ETS* [76] | | Stationary [78] | | LightTS [88] | | In* [80] | |
|---|---|---|---|---|---|---|---|---|---|---|---|---|---|---|---|---|---|
| Metrics | | MSE | MAE | MSE | MAE | MSE | MAE | MSE | MAE | MSE | MAE | MSE | MAE | MSE | MAE | MSE | MAE |
| ETTm1 | 96 | 0.318 | 0.353 | 0.501 | 0.479 | 0.360 | 0.399 | 0.378 | 0.418 | 0.375 | 0.398 | 0.418 | 0.415 | 0.390 | 0.411 | 0.619 | 0.549 |
| | 192 | 0.362 | 0.380 | 0.578 | 0.510 | 0.422 | 0.449 | 0.438 | 0.449 | 0.408 | 0.410 | 0.506 | 0.454 | 0.425 | 0.436 | 0.760 | 0.645 |
| | 336 | 0.386 | 0.402 | 0.668 | 0.552 | 0.589 | 0.557 | 0.456 | 0.462 | 0.435 | 0.428 | 0.530 | 0.482 | 0.463 | 0.464 | 1.093 | 0.812 |
| | 720 | 0.419 | 0.423 | 0.602 | 0.524 | 0.838 | 0.706 | 0.530 | 0.498 | 0.499 | 0.462 | 0.610 | 0.525 | 0.547 | 0.520 | 1.114 | 0.806 |
| | Avg | **0.371** | **0.390** | 0.587 | 0.516 | 0.552 | 0.528 | 0.450 | 0.457 | 0.429 | 0.425 | 0.516 | 0.469 | 0.456 | 0.458 | 0.896 | 0.703 |
| ETTm1 | 96 | 0.174 | 0.257 | 0.245 | 0.323 | 0.274 | 0.268 | 0.196 | 0.284 | 0.189 | 0.280 | 0.240 | 0.308 | 0.226 | 0.323 | 0.467 | 0.533 |
| | 192 | 0.238 | 0.299 | 0.289 | 0.345 | 0.366 | 0.380 | 0.264 | 0.325 | 0.275 | 0.319 | 0.428 | 0.402 | 0.361 | 0.421 | 0.742 | 0.664 |
| | 336 | 0.295 | 0.337 | 0.342 | 0.378 | 0.437 | 0.453 | 0.324 | 0.363 | 0.314 | 0.357 | 0.521 | 0.449 | 0.474 | 0.488 | 1.184 | 0.825 |
| | 720 | 0.390 | 0.392 | 0.441 | 0.429 | 0.506 | 0.623 | 0.434 | 0.428 | 0.414 | 0.413 | 0.602 | 0.501 | 0.760 | 0.631 | 4.039 | 1.530 |
| | Avg | **0.274** | **0.321** | 0.329 | 0.369 | 0.396 | 0.431 | 0.305 | 0.350 | 0.298 | 0.342 | 0.448 | 0.415 | 0.455 | 0.466 | 1.608 | 0.888 |
| ETTh1 | 96 | 0.376 | 0.400 | 0.453 | 0.459 | 0.462 | 0.473 | 0.376 | 0.417 | 0.494 | 0.479 | 0.550 | 0.503 | 0.448 | 0.450 | 0.926 | 0.741 |
| | 192 | 0.428 | 0.431 | 0.481 | 0.470 | 0.495 | 0.484 | 0.431 | 0.454 | 0.538 | 0.504 | 0.655 | 0.569 | 0.503 | 0.483 | 0.968 | 0.757 |
| | 336 | 0.463 | 0.456 | 0.519 | 0.495 | 0.693 | 0.626 | 0.461 | 0.469 | 0.574 | 0.521 | 0.791 | 0.639 | 0.554 | 0.513 | 1.144 | 0.849 |
| | 720 | 0.450 | 0.458 | 0.510 | 0.508 | 0.668 | 0.599 | 0.502 | 0.499 | 0.562 | 0.535 | 0.797 | 0.652 | 0.627 | 0.578 | 1.214 | 0.880 |
| | Avg | **0.430** | **0.436** | 0.491 | 0.483 | 0.580 | 0.545 | 0.442 | 0.460 | 0.542 | 0.510 | 0.698 | 0.591 | 0.533 | 0.506 | 1.063 | 0.807 |
| ETTh2 | 96 | 0.293 | 0.347 | 0.383 | 0.416 | 0.367 | 0.347 | 0.346 | 0.390 | 0.340 | 0.391 | 0.417 | 0.432 | 0.417 | 0.448 | 3.132 | 1.425 |
| | 192 | 0.364 | 0.397 | 0.479 | 0.467 | 0.450 | 0.459 | 0.428 | 0.439 | 0.430 | 0.439 | 0.529 | 0.486 | 0.546 | 0.520 | 5.552 | 1.957 |
| | 336 | 0.385 | 0.423 | 0.476 | 0.481 | 0.532 | 0.521 | 0.469 | 0.474 | 0.485 | 0.479 | 0.591 | 0.517 | 0.619 | 0.554 | 4.926 | 1.873 |
| | 720 | 0.420 | 0.441 | 0.494 | 0.503 | 0.614 | 0.633 | 0.473 | 0.486 | 0.500 | 0.497 | 0.601 | 0.531 | 0.972 | 0.704 | 4.201 | 1.741 |
| | Avg | **0.365** | **0.402** | 0.458 | 0.467 | 0.491 | 0.490 | 0.429 | 0.447 | 0.439 | 0.452 | 0.534 | 0.491 | 0.639 | 0.556 | 4.453 | 1.749 |
| Weather | 96 | 0.166 | 0.213 | 0.276 | 0.343 | 0.174 | 0.243 | 0.218 | 0.299 | 0.197 | 0.281 | 0.184 | 0.233 | 0.174 | 0.235 | 0.357 | 0.415 |
| | 192 | 0.212 | 0.254 | 0.305 | 0.361 | 0.235 | 0.307 | 0.281 | 0.344 | 0.237 | 0.312 | 0.248 | 0.286 | 0.218 | 0.276 | 0.458 | 0.456 |
| | 336 | 0.267 | 0.294 | 0.372 | 0.405 | 0.277 | 0.342 | 0.337 | 0.375 | 0.298 | 0.353 | 0.337 | 0.349 | 0.267 | 0.316 | 0.520 | 0.501 |
| | 720 | 0.342 | 0.344 | 0.430 | 0.437 | 0.369 | 0.407 | 0.423 | 0.429 | 0.352 | 0.288 | 0.399 | 0.385 | 0.353 | 0.366 | 0.926 | 0.705 |
| | Avg | **0.247** | **0.276** | 0.346 | 0.387 | 0.264 | 0.325 | 0.315 | 0.362 | 0.271 | 0.309 | 0.292 | 0.313 | 0.253 | 0.298 | 0.565 | 0.519 |
| ECL | 96 | 0.162 | 0.253 | 0.198 | 0.313 | 0.146 | 0.249 | 0.202 | 0.314 | 0.187 | 0.304 | 0.167 | 0.270 | 0.211 | 0.313 | 0.342 | 0.423 |
| | 192 | 0.173 | 0.264 | 0.218 | 0.329 | 0.163 | 0.262 | 0.211 | 0.323 | 0.199 | 0.315 | 0.183 | 0.284 | 0.223 | 0.326 | 0.360 | 0.442 |
| | 336 | 0.194 | 0.285 | 0.253 | 0.352 | 0.198 | 0.296 | 0.222 | 0.335 | 0.212 | 0.329 | 0.194 | 0.295 | 0.243 | 0.346 | 0.365 | 0.445 |
| | 720 | 0.220 | 0.304 | 0.265 | 0.367 | 0.245 | 0.346 | 0.272 | 0.373 | 0.233 | 0.345 | 0.224 | 0.321 | 0.277 | 0.371 | 0.412 | 0.469 |
| | Avg | **0.187** | **0.276** | 0.233 | 0.340 | 0.188 | 0.288 | 0.227 | 0.337 | 0.208 | 0.323 | 0.192 | 0.292 | 0.239 | 0.339 | 0.370 | 0.445 |
| Traffic | 96 | 0.432 | 0.280 | 0.608 | 0.383 | 0.516 | 0.268 | 0.592 | 0.372 | 0.607 | 0.392 | 0.621 | 0.347 | 0.667 | 0.419 | 0.720 | 0.407 |
| | 192 | 0.444 | 0.287 | 0.630 | 0.397 | 0.541 | 0.283 | 0.598 | 0.371 | 0.621 | 0.399 | 0.643 | 0.355 | 0.662 | 0.425 | 0.738 | 0.414 |
| | 336 | 0.458 | 0.293 | 0.622 | 0.387 | 0.566 | 0.351 | 0.636 | 0.397 | 0.622 | 0.396 | 0.650 | 0.360 | 0.683 | 0.436 | 0.833 | 0.470 |
| | 720 | 0.492 | 0.303 | 0.689 | 0.396 | 0.610 | 0.403 | 0.639 | 0.395 | 0.632 | 0.396 | 0.670 | 0.365 | 0.700 | 0.455 | 0.854 | 0.491 |
| | Avg | **0.457** | **0.291** | 0.637 | 0.391 | 0.558 | 0.326 | 0.616 | 0.384 | 0.621 | 0.396 | 0.646 | 0.357 | 0.678 | 0.434 | 0.786 | 0.445 |
| Exchange | 96 | 0.084 | 0.201 | 0.191 | 0.318 | 0.276 | 0.383 | 0.162 | 0.291 | 0.085 | 0.204 | 0.132 | 0.254 | 0.128 | 0.266 | 0.896 | 0.761 |
| | 192 | 0.174 | 0.295 | 0.315 | 0.407 | 0.540 | 0.552 | 0.276 | 0.382 | 0.182 | 0.303 | 0.251 | 0.361 | 0.292 | 0.402 | 1.146 | 0.861 |
| | 336 | 0.331 | 0.416 | 0.480 | 0.519 | 1.229 | 0.873 | 0.442 | 0.488 | 0.348 | 0.428 | 0.467 | 0.507 | 0.500 | 0.536 | 1.628 | 1.017 |
| | 720 | 0.847 | 0.694 | 1.255 | 0.868 | 1.721 | 1.055 | 1.175 | 0.833 | 1.025 | 0.774 | 1.304 | 0.837 | 1.002 | 0.763 | 2.552 | 1.299 |
| | Avg | **0.359** | **0.401** | 0.560 | 0.528 | 0.942 | 0.716 | 0.514 | 0.498 | 0.410 | 0.427 | 0.538 | 0.490 | 0.480 | 0.492 | 1.555 | 0.984 |
| Average | | **0.336** | **0.349** | 0.455 | 0.435 | 0.496 | 0.456 | 0.456 | 0.412 | 0.402 | 0.398 | 0.483 | 0.427 | 0.467 | 0.444 | 1.412 | 0.818 |

Table 22: Full results for the long-term forecasting task compared with PatchTST [54], TimeMixer [89], TimesNet [72], Autoformer [75], DLinear [87], iTransformer [74], TimeXer [62], FEDformer [77]. (* means former, PTST is PatchTST, TMixer is TimeMixer.) To ensure fairness in the comparison, we set the look-back window length of all models to **336**. The standard deviation is within 0.5%. **Red**: best, Blue: second best.

| Methods | | SymTime (Our) | | PTST [54] | | TMixer [89] | | TimesNet [72] | | Auto* [75] | | DLinear [87] | | iTrans* [74] | | TimeXer [62] | | FED* [77] | |
|---|---|---|---|---|---|---|---|---|---|---|---|---|---|---|---|---|---|---|---|
| Metrics | | MSE | MAE | MSE | MAE | MSE | MAE | MSE | MAE | MSE | MAE | MSE | MAE | MSE | MAE | MSE | MAE | MSE | MAE |
| ETTm1 | 96 | 0.295 | 0.346 | 0.292 | 0.343 | 0.322 | 0.358 | 0.344 | 0.378 | 0.514 | 0.506 | 0.300 | 0.344 | 0.305 | 0.359 | 0.318 | 0.361 | 0.382 | 0.427 |
| | 192 | 0.328 | 0.365 | 0.331 | 0.369 | 0.342 | 0.375 | 0.456 | 0.426 | 0.576 | 0.520 | 0.335 | 0.365 | 0.345 | 0.382 | 0.354 | 0.383 | 0.393 | 0.434 |
| | 336 | 0.366 | 0.395 | 0.365 | 0.392 | 0.371 | 0.395 | 0.426 | 0.432 | 0.703 | 0.564 | 0.369 | 0.386 | 0.377 | 0.401 | 0.381 | 0.402 | 0.445 | 0.459 |
| | 720 | 0.411 | 0.422 | 0.420 | 0.425 | 0.437 | 0.440 | 0.459 | 0.455 | 0.678 | 0.568 | 0.425 | 0.420 | 0.444 | 0.439 | 0.434 | 0.433 | 0.543 | 0.490 |
| | Avg | 0.350 | 0.382 | 0.352 | 0.382 | 0.368 | 0.392 | 0.421 | 0.423 | 0.618 | 0.539 | 0.357 | 0.379 | 0.368 | 0.395 | 0.372 | 0.395 | 0.441 | 0.452 |
| ETTm2 | 96 | 0.165 | 0.255 | 0.165 | 0.255 | 0.176 | 0.259 | 0.184 | 0.272 | 0.349 | 0.400 | 0.169 | 0.266 | 0.174 | 0.265 | 0.169 | 0.254 | 0.260 | 0.336 |
| | 192 | 0.221 | 0.293 | 0.220 | 0.292 | 0.232 | 0.298 | 0.243 | 0.309 | 0.507 | 0.474 | 0.235 | 0.316 | 0.247 | 0.313 | 0.237 | 0.302 | 0.291 | 0.354 |
| | 336 | 0.275 | 0.329 | 0.278 | 0.329 | 0.280 | 0.329 | 0.310 | 0.351 | 0.328 | 0.375 | 0.305 | 0.366 | 0.294 | 0.345 | 0.284 | 0.333 | 0.325 | 0.366 |
| | 720 | 0.365 | 0.387 | 0.368 | 0.385 | 0.359 | 0.387 | 0.393 | 0.405 | 0.418 | 0.433 | 0.457 | 0.463 | 0.374 | 0.394 | 0.360 | 0.381 | 0.423 | 0.451 |
| | Avg | 0.256 | 0.316 | 0.258 | 0.315 | 0.262 | 0.318 | 0.282 | 0.334 | 0.400 | 0.420 | 0.291 | 0.353 | 0.272 | 0.329 | 0.262 | 0.317 | 0.325 | 0.377 |
| ETTh1 | 96 | 0.372 | 0.399 | 0.382 | 0.405 | 0.379 | 0.403 | 0.423 | 0.437 | 0.536 | 0.498 | 0.375 | 0.399 | 0.397 | 0.416 | 0.403 | 0.421 | 0.387 | 0.434 |
| | 192 | 0.409 | 0.427 | 0.414 | 0.421 | 0.415 | 0.423 | 0.481 | 0.481 | 0.562 | 0.533 | 0.413 | 0.424 | 0.442 | 0.448 | 0.440 | 0.440 | 0.431 | 0.458 |
| | 336 | 0.430 | 0.440 | 0.431 | 0.435 | 0.453 | 0.449 | 0.488 | 0.477 | 0.551 | 0.533 | 0.438 | 0.444 | 0.459 | 0.459 | 0.495 | 0.488 | 0.471 | 0.480 |
| | 720 | 0.440 | 0.463 | 0.449 | 0.466 | 0.473 | 0.473 | 0.548 | 0.523 | 0.670 | 0.590 | 0.475 | 0.495 | 0.503 | 0.506 | 0.633 | 0.583 | 0.512 | 0.516 |
| | Avg | 0.413 | 0.432 | 0.419 | 0.432 | 0.430 | 0.437 | 0.485 | 0.480 | 0.580 | 0.539 | 0.425 | 0.440 | 0.450 | 0.457 | 0.493 | 0.483 | 0.450 | 0.472 |
| ETTh2 | 96 | 0.271 | 0.341 | 0.274 | 0.336 | 0.284 | 0.350 | 0.378 | 0.421 | 0.509 | 0.527 | 0.307 | 0.370 | 0.307 | 0.363 | 0.313 | 0.365 | 0.394 | 0.457 |
| | 192 | 0.334 | 0.378 | 0.339 | 0.379 | 0.359 | 0.397 | 0.409 | 0.439 | 0.711 | 0.641 | 0.402 | 0.431 | 0.393 | 0.413 | 0.375 | 0.404 | 0.426 | 0.456 |
| | 336 | 0.359 | 0.403 | 0.331 | 0.381 | 0.388 | 0.422 | 0.410 | 0.439 | 0.574 | 0.569 | 0.489 | 0.485 | 0.428 | 0.437 | 0.400 | 0.429 | 0.420 | 0.461 |
| | 720 | 0.398 | 0.436 | 0.379 | 0.422 | 0.555 | 0.534 | 0.440 | 0.461 | 0.856 | 0.679 | 0.761 | 0.620 | 0.433 | 0.452 | 0.412 | 0.443 | 0.478 | 0.495 |
| | Avg | 0.341 | 0.390 | 0.331 | 0.379 | 0.396 | 0.425 | 0.409 | 0.440 | 0.663 | 0.604 | 0.490 | 0.476 | 0.390 | 0.416 | 0.375 | 0.410 | 0.430 | 0.467 |
| Weather | 96 | 0.149 | 0.199 | 0.227 | 0.273 | 0.175 | 0.224 | 0.170 | 0.228 | 0.268 | 0.343 | 0.174 | 0.234 | 0.162 | 0.210 | 0.169 | 0.203 | 0.217 | 0.296 |
| | 192 | 0.192 | 0.239 | 0.200 | 0.245 | 0.196 | 0.243 | 0.214 | 0.263 | 0.431 | 0.465 | 0.217 | 0.276 | 0.207 | 0.251 | 0.243 | 0.265 | 0.288 | 0.342 |
| | 336 | 0.245 | 0.282 | 0.259 | 0.298 | 0.243 | 0.280 | 0.272 | 0.301 | 0.559 | 0.498 | 0.262 | 0.313 | 0.257 | 0.291 | 0.322 | 0.318 | 0.340 | 0.382 |
| | 720 | 0.321 | 0.337 | 0.346 | 0.353 | 0.325 | 0.346 | 0.343 | 0.353 | 0.506 | 0.495 | 0.328 | 0.370 | 0.327 | 0.337 | 0.414 | 0.373 | 0.405 | 0.430 |
| | Avg | 0.238 | 0.273 | 0.258 | 0.292 | 0.235 | 0.273 | 0.250 | 0.286 | 0.441 | 0.450 | 0.245 | 0.298 | 0.238 | 0.272 | 0.287 | 0.290 | 0.313 | 0.363 |
| ECL | 96 | 0.133 | 0.230 | 0.131 | 0.361 | 0.144 | 0.244 | 0.174 | 0.278 | 0.205 | 0.322 | 0.147 | 0.249 | 0.132 | 0.227 | 0.155 | 0.235 | 0.193 | 0.308 |
| | 192 | 0.150 | 0.244 | 0.154 | 0.251 | 0.152 | 0.242 | 0.192 | 0.292 | 0.219 | 0.333 | 0.160 | 0.261 | 0.153 | 0.248 | 0.161 | 0.281 | 0.201 | 0.315 |
| | 336 | 0.163 | 0.262 | 0.164 | 0.262 | 0.172 | 0.261 | 0.198 | 0.299 | 0.227 | 0.340 | 0.176 | 0.278 | 0.173 | 0.267 | 0.187 | 0.279 | 0.214 | 0.329 |
| | 720 | 0.208 | 0.298 | 0.210 | 0.301 | 0.207 | 0.293 | 0.222 | 0.318 | 0.294 | 0.390 | 0.196 | 0.288 | 0.194 | 0.287 | 0.183 | 0.273 | 0.246 | 0.355 |
| | Avg | 0.164 | 0.258 | 0.165 | 0.294 | 0.169 | 0.260 | 0.197 | 0.297 | 0.236 | 0.346 | 0.170 | 0.269 | 0.163 | 0.257 | 0.172 | 0.267 | 0.214 | 0.327 |
| Traffic | 96 | 0.361 | 0.257 | 0.365 | 0.256 | 0.371 | 0.256 | 0.589 | 0.321 | 0.675 | 0.414 | 0.431 | 0.307 | 0.367 | 0.278 | 0.422 | 0.268 | 0.587 | 0.366 |
| | 192 | 0.382 | 0.258 | 0.383 | 0.258 | 0.401 | 0.271 | 0.605 | 0.322 | 0.672 | 0.411 | 0.443 | 0.312 | 0.414 | 0.284 | 0.433 | 0.280 | 0.604 | 0.373 |
| | 336 | 0.395 | 0.271 | 0.398 | 0.271 | 0.408 | 0.267 | 0.621 | 0.338 | 0.667 | 0.408 | 0.456 | 0.319 | 0.399 | 0.280 | 0.454 | 0.278 | 0.621 | 0.383 |
| | 720 | 0.424 | 0.281 | 0.438 | 0.288 | 0.464 | 0.292 | 0.647 | 0.344 | 0.689 | 0.421 | 0.528 | 0.343 | 0.422 | 0.290 | 0.498 | 0.299 | 0.626 | 0.382 |
| | Avg | 0.391 | 0.267 | 0.396 | 0.268 | 0.411 | 0.271 | 0.615 | 0.331 | 0.676 | 0.413 | 0.465 | 0.320 | 0.401 | 0.283 | 0.452 | 0.281 | 0.610 | 0.376 |
| Exchange | 96 | 0.085 | 0.204 | 0.093 | 0.214 | 0.092 | 0.217 | 0.218 | 0.343 | 0.967 | 0.778 | 0.099 | 0.235 | 0.099 | 0.224 | 0.234 | 0.312 | 0.168 | 0.285 |
| | 192 | 0.180 | 0.301 | 0.200 | 0.321 | 0.235 | 0.345 | 0.299 | 0.411 | 0.931 | 0.750 | 0.195 | 0.335 | 0.215 | 0.338 | 0.283 | 0.402 | 0.186 | 0.296 |
| | 336 | 0.335 | 0.419 | 0.373 | 0.448 | 0.370 | 0.441 | 0.468 | 0.527 | 1.051 | 0.809 | 0.380 | 0.474 | 0.378 | 0.454 | 0.433 | 0.343 | 0.250 | 0.342 |
| | 720 | 0.869 | 0.700 | 0.875 | 0.695 | 0.963 | 0.750 | 1.208 | 0.847 | 1.261 | 0.893 | 1.120 | 0.805 | 0.876 | 0.690 | 0.688 | 0.942 | 0.899 | 0.784 |
| | Avg | 0.367 | 0.406 | 0.385 | 0.420 | 0.415 | 0.438 | 0.548 | 0.532 | 1.053 | 0.807 | 0.448 | 0.462 | 0.392 | 0.427 | 0.409 | 0.500 | 0.376 | 0.427 |
| Average | | 0.315 | 0.341 | 0.320 | 0.348 | 0.336 | 0.352 | 0.401 | 0.390 | 0.583 | 0.515 | 0.361 | 0.375 | 0.347 | 0.355 | 0.353 | 0.368 | 0.395 | 0.408 |

Table 23: Full results for the long-term forecasting task compared with PatchTST [54], TimeMixer [89], TimesNet [72], Autoformer [75], DLinear [87], iTransformer [74], TimeXer [62], FITS [90]. (* means former, TMixer is TimeMixer.) To ensure fairness in the comparison, we set the look-back window length of all models to **512**. The standard deviation is within 0.5%. **Red**: best, Blue: second best.

| Methods | | SymTime (Ours) | | PatchTST [54] | | TMixer [89] | | TimesNet [72] | | Auto* [75] | | DLinear [87] | | iTrans* [74] | | TimeXer [62] | | FITS [90] | |
|---|---|---|---|---|---|---|---|---|---|---|---|---|---|---|---|---|---|---|---|
| Metrics | | MSE | MAE | MSE | MAE | MSE | MAE | MSE | MAE | MSE | MAE | MSE | MAE | MSE | MAE | MSE | MAE | MSE | MAE |
| ETTm1 | 96 | 0.313 | 0.348 | 0.290 | 0.344 | 0.315 | 0.354 | 0.358 | 0.388 | 0.495 | 0.499 | 0.304 | 0.347 | 0.310 | 0.364 | 0.332 | 0.374 | 0.306 | 0.349 |
| | 192 | 0.326 | 0.363 | 0.333 | 0.371 | 0.344 | 0.376 | 0.457 | 0.439 | 0.549 | 0.514 | 0.337 | 0.368 | 0.349 | 0.387 | 0.364 | 0.392 | 0.338 | 0.367 |
| | 336 | 0.376 | 0.390 | 0.369 | 0.392 | 0.385 | 0.397 | 0.412 | 0.427 | 0.536 | 0.509 | 0.366 | 0.385 | 0.376 | 0.401 | 0.387 | 0.407 | 0.421 | 0.384 |
| | 720 | 0.409 | 0.420 | 0.416 | 0.420 | 0.441 | 0.440 | 0.473 | 0.467 | 0.645 | 0.550 | 0.424 | 0.422 | 0.434 | 0.436 | 0.430 | 0.432 | 0.432 | 0.437 |
| | Avg | 0.356 | **0.380** | **0.352** | 0.382 | 0.371 | 0.392 | 0.425 | 0.430 | 0.556 | 0.518 | 0.358 | 0.380 | 0.367 | 0.397 | 0.378 | 0.401 | 0.374 | 0.384 |
| ETTm2 | 96 | 0.169 | 0.258 | 0.166 | 0.256 | 0.173 | 0.261 | 0.192 | 0.279 | 0.278 | 0.353 | 0.166 | 0.262 | 0.182 | 0.272 | 0.177 | 0.263 | 0.165 | 0.254 |
| | 192 | 0.232 | 0.301 | 0.223 | 0.296 | 0.223 | 0.298 | 0.264 | 0.325 | 0.318 | 0.383 | 0.225 | 0.304 | 0.242 | 0.312 | 0.241 | 0.312 | 0.219 | 0.291 |
| | 336 | 0.287 | 0.333 | 0.274 | 0.329 | 0.291 | 0.342 | 0.320 | 0.364 | 0.411 | 0.444 | 0.299 | 0.361 | 0.292 | 0.346 | 0.294 | 0.340 | 0.272 | 0.326 |
| | 720 | 0.372 | 0.387 | 0.362 | 0.385 | 0.366 | 0.391 | 0.402 | 0.410 | 0.476 | 0.484 | 0.412 | 0.432 | 0.378 | 0.396 | 0.382 | 0.399 | 0.359 | 0.391 |
| | Avg | 0.265 | 0.320 | 0.256 | 0.317 | 0.263 | 0.323 | 0.294 | 0.344 | 0.371 | 0.416 | 0.275 | 0.340 | 0.273 | 0.331 | 0.274 | 0.329 | **0.254** | **0.313** |
| ETTh1 | 96 | 0.372 | 0.389 | 0.370 | 0.400 | 0.380 | 0.408 | 0.442 | 0.465 | 0.556 | 0.534 | 0.368 | 0.397 | 0.394 | 0.420 | 0.397 | 0.424 | 0.372 | 0.396 |
| | 192 | 0.402 | 0.424 | 0.413 | 0.429 | 0.431 | 0.444 | 0.473 | 0.475 | 0.568 | 0.538 | 0.400 | 0.417 | 0.430 | 0.444 | 0.438 | 0.452 | 0.405 | 0.415 |
| | 336 | 0.433 | 0.445 | 0.422 | 0.440 | 0.468 | 0.472 | 0.505 | 0.501 | 0.606 | 0.573 | 0.430 | 0.442 | 0.447 | 0.459 | 0.466 | 0.472 | 0.440 | 0.468 |
| | 720 | 0.448 | 0.469 | 0.447 | 0.468 | 0.435 | 0.453 | 0.505 | 0.504 | 0.777 | 0.669 | 0.476 | 0.497 | 0.514 | 0.516 | 0.597 | 0.566 | 0.453 | 0.485 |
| | Avg | 0.414 | **0.432** | **0.413** | 0.434 | 0.429 | 0.444 | 0.481 | 0.486 | 0.627 | 0.579 | 0.418 | 0.438 | 0.446 | 0.460 | 0.475 | 0.479 | 0.418 | 0.441 |
| ETTh2 | 96 | 0.290 | 0.352 | 0.273 | 0.337 | 0.298 | 0.364 | 0.336 | 0.391 | 0.447 | 0.486 | 0.288 | 0.355 | 0.317 | 0.367 | 0.299 | 0.358 | 0.291 | 0.353 |
| | 192 | 0.376 | 0.409 | 0.341 | 0.382 | 0.361 | 0.399 | 0.393 | 0.425 | 0.587 | 0.568 | 0.394 | 0.427 | 0.388 | 0.411 | 0.370 | 0.408 | 0.350 | 0.395 |
| | 336 | 0.382 | 0.416 | 0.398 | 0.458 | 0.393 | 0.421 | 0.406 | 0.445 | 0.689 | 0.608 | 0.501 | 0.491 | 0.422 | 0.437 | 0.372 | 0.410 | 0.375 | 0.424 |
| | 720 | 0.414 | 0.442 | 0.416 | 0.458 | 0.442 | 0.458 | 0.451 | 0.466 | 1.027 | 0.772 | 0.813 | 0.638 | 0.424 | 0.454 | 0.376 | 0.424 | 0.437 | 0.459 |
| | Avg | 0.365 | 0.405 | 0.357 | 0.409 | 0.373 | 0.410 | 0.397 | 0.432 | 0.687 | 0.609 | 0.499 | 0.478 | 0.388 | 0.417 | **0.354** | **0.400** | 0.363 | 0.408 |
| Weather | 96 | 0.159 | 0.205 | 0.230 | 0.282 | 0.169 | 0.225 | 0.168 | 0.224 | 0.375 | 0.426 | 0.171 | 0.230 | 0.175 | 0.223 | 0.175 | 0.209 | 0.172 | 0.226 |
| | 192 | 0.203 | 0.260 | 0.194 | 0.242 | 0.191 | 0.242 | 0.217 | 0.266 | 0.471 | 0.491 | 0.213 | 0.269 | 0.213 | 0.256 | 0.246 | 0.267 | 0.216 | 0.262 |
| | 336 | 0.256 | 0.289 | 0.245 | 0.282 | 0.246 | 0.283 | 0.278 | 0.310 | 0.514 | 0.521 | 0.260 | 0.312 | 0.265 | 0.296 | 0.314 | 0.309 | 0.261 | 0.295 |
| | 720 | 0.319 | 0.339 | 0.312 | 0.332 | 0.316 | 0.333 | 0.342 | 0.352 | 0.596 | 0.506 | 0.320 | 0.358 | 0.342 | 0.347 | 0.391 | 0.353 | 0.326 | 0.342 |
| | Avg | 0.234 | 0.273 | 0.245 | 0.284 | **0.231** | **0.271** | 0.251 | 0.288 | 0.489 | 0.486 | 0.241 | 0.292 | 0.249 | 0.280 | 0.282 | 0.284 | 0.244 | 0.281 |
| ECL | 96 | 0.128 | 0.256 | 0.129 | 0.224 | 0.137 | 0.229 | 0.184 | 0.288 | 0.215 | 0.328 | 0.141 | 0.241 | 0.131 | 0.227 | 0.140 | 0.242 | 0.145 | 0.244 |
| | 192 | 0.149 | 0.244 | 0.158 | 0.258 | 0.152 | 0.250 | 0.187 | 0.291 | 0.224 | 0.333 | 0.154 | 0.254 | 0.155 | 0.250 | 0.157 | 0.256 | 0.153 | 0.250 |
| | 336 | 0.161 | 0.261 | 0.163 | 0.261 | 0.193 | 0.295 | 0.206 | 0.307 | 0.235 | 0.341 | 0.169 | 0.271 | 0.171 | 0.266 | 0.176 | 0.275 | 0.169 | 0.266 |
| | 720 | 0.216 | 0.308 | 0.225 | 0.331 | 0.228 | 0.324 | 0.226 | 0.323 | 0.738 | 0.570 | 0.204 | 0.304 | 0.191 | 0.285 | 0.211 | 0.306 | 0.208 | 0.298 |
| | Avg | 0.163 | 0.267 | 0.169 | 0.269 | 0.177 | 0.274 | 0.201 | 0.302 | 0.353 | 0.393 | 0.167 | 0.267 | **0.162** | **0.257** | 0.171 | 0.270 | 0.169 | 0.265 |
| Traffic | 96 | 0.365 | 0.257 | 0.369 | 0.262 | 0.368 | 0.254 | 0.600 | 0.321 | 0.697 | 0.428 | 0.412 | 0.294 | 0.351 | 0.257 | 0.428 | 0.271 | 0.398 | 0.277 |
| | 192 | 0.379 | 0.256 | 0.379 | 0.253 | 0.399 | 0.268 | 0.612 | 0.328 | 0.700 | 0.429 | 0.422 | 0.299 | 0.373 | 0.268 | 0.448 | 0.282 | 0.409 | 0.280 |
| | 336 | 0.401 | 0.275 | 0.410 | 0.280 | 0.404 | 0.264 | 0.631 | 0.338 | 0.707 | 0.437 | 0.431 | 0.304 | 0.386 | 0.274 | 0.473 | 0.289 | 0.418 | 0.285 |
| | 720 | 0.433 | 0.284 | 0.438 | 0.291 | 0.467 | 0.293 | 0.654 | 0.352 | 0.718 | 0.445 | 0.468 | 0.325 | 0.424 | 0.294 | 0.516 | 0.307 | 0.456 | 0.306 |
| | Avg | 0.395 | **0.268** | 0.399 | 0.272 | 0.410 | 0.270 | 0.624 | 0.334 | 0.705 | 0.435 | 0.433 | 0.305 | **0.383** | 0.273 | 0.466 | 0.287 | 0.420 | 0.287 |
| Exchange | 96 | 0.089 | 0.212 | 0.095 | 0.220 | 0.096 | 0.220 | 0.238 | 0.366 | 0.617 | 0.623 | 0.120 | 0.260 | 0.135 | 0.267 | 0.099 | 0.223 | 0.100 | 0.225 |
| | 192 | 0.176 | 0.298 | 0.215 | 0.336 | 0.197 | 0.318 | 0.439 | 0.497 | 0.810 | 0.739 | 0.241 | 0.376 | 0.322 | 0.417 | 0.207 | 0.331 | 0.201 | 0.326 |
| | 336 | 0.380 | 0.439 | 0.392 | 0.459 | 0.737 | 0.642 | 0.647 | 0.620 | 0.858 | 0.746 | 0.439 | 0.509 | 0.361 | 0.448 | 0.767 | 0.689 | 0.350 | 0.437 |
| | 720 | 0.893 | 0.698 | 0.890 | 0.680 | 1.038 | 0.808 | 1.550 | 0.948 | 1.491 | 0.965 | 1.199 | 0.831 | 0.888 | 0.736 | 0.984 | 0.780 | 0.920 | 0.728 |
| | Avg | **0.384** | **0.412** | 0.398 | 0.423 | 0.517 | 0.497 | 0.718 | 0.608 | 0.944 | 0.768 | 0.500 | 0.494 | 0.427 | 0.467 | 0.514 | 0.506 | 0.393 | 0.439 |
| Average | | **0.322** | **0.345** | 0.324 | 0.349 | 0.346 | 0.360 | 0.424 | 0.403 | 0.591 | 0.525 | 0.361 | 0.374 | 0.337 | 0.360 | 0.364 | 0.369 | 0.329 | 0.352 |

Table 24: Full results for the short-term forecasting task in the M4 dataset compared with Peri-midFormer [63], $S^2$IP-LLM(S-LLM) [82], Time-LLM(T-LLM) [17], GPT4TS [2], TimeMixer [89], PatchTST [54], iTransformer [74], TimesNet [72], DLinear [87], Informer [80]. (* means former.) The standard deviation is within 0.5%. **Red**: best, Blue: second best.

| Methods / Metric | SymTime (Ours) | Peri-mid* [63] | S-LLM [82] | T-LLM [17] | GPT4TS [2] | TimeMixer [89] | PatchTST [54] | iTrans* [74] | TimesNet [72] | DLinear [87] | In* [80] |
|---|---|---|---|---|---|---|---|---|---|---|---|
| **Yearly** SMAPE | **13.355** | 13.483 | 14.931 | 13.450 | 14.847 | 13.369 | 13.677 | 13.724 | 13.463 | 14.340 | 14.698 |
| **Yearly** MASE | **2.997** | 3.080 | 3.345 | 3.184 | 3.628 | 3.009 | 3.049 | 3.157 | 3.058 | 3.112 | 3.293 |
| **Yearly** OWA | **0.786** | 0.800 | 0.878 | 0.819 | 0.911 | 0.787 | 0.802 | 0.817 | 0.797 | 0.830 | 0.864 |
| **Quarterly** SMAPE | 10.060 | **10.037** | 10.655 | 10.671 | 10.389 | 10.131 | 10.922 | 13.473 | 10.069 | 10.510 | 16.172 |
| **Quarterly** MASE | 1.183 | **1.170** | 1.249 | 1.276 | 1.228 | 1.186 | 1.326 | 1.722 | 1.175 | 1.241 | 2.136 |
| **Quarterly** OWA | **0.872** | 0.882 | 0.939 | 0.950 | 0.919 | 0.893 | 0.979 | 1.240 | 0.886 | 0.930 | 1.513 |
| **Monthly** SMAPE | **12.608** | 12.795 | 13.012 | 13.416 | 12.907 | 12.762 | 14.200 | 13.674 | 12.760 | 13.382 | 15.446 |
| **Monthly** MASE | **0.925** | 0.948 | 0.973 | 1.045 | 0.954 | 0.940 | 1.111 | 1.068 | 0.947 | 1.007 | 1.247 |
| **Monthly** OWA | **0.872** | 0.889 | 0.909 | 0.957 | 0.896 | 0.884 | 1.015 | 0.976 | 0.887 | 0.937 | 1.122 |
| **Others** SMAPE | 4.941 | **4.912** | 5.540 | 4.973 | 5.266 | 5.085 | 5.658 | 5.598 | 4.995 | 5.122 | 6.839 |
| **Others** MASE | 3.327 | **3.260** | 8.426 | 3.412 | 3.595 | 3.403 | 3.626 | 3.957 | 3.346 | 3.608 | 4.536 |
| **Others** OWA | 1.045 | **1.031** | 3.792 | 1.059 | 1.121 | 1.072 | 1.167 | 1.213 | 1.053 | 1.108 | 1.435 |
| **Average** SMAPE | **11.785** | 11.897 | 12.514 | 12.584 | 12.367 | 11.885 | 12.866 | 13.233 | 11.888 | 12.500 | 15.018 |
| **Average** MASE | **1.584** | 1.607 | 1.726 | 1.763 | 1.767 | 1.598 | 1.734 | 1.850 | 1.607 | 1.678 | 2.096 |
| **Average** OWA | **0.849** | 0.859 | 0.913 | 0.915 | 0.918 | 0.856 | 0.928 | 0.972 | 0.858 | 0.899 | 1.102 |

Table 25: Full results for the short-term forecasting task in the M4 dataset compared with LightTS [88], Autoformer [75], Crossformer [79], FEDformer [77], ETSformer [76], Nonstationary Transformer (Stationary) [78], FiLM [142], MICN [86], Reformer [143], Pyraformer [130]. The standard deviation is within 0.5%. (* means former.) **Red**: best, Blue: second best.

| Methods / Metric | SymTime (Ours) | LightTS [88] | Auto* [75] | Cross* [79] | FED* [77] | ETS* [76] | Stationary [78] | FiLM [142] | MICN [86] | Re* [143] | Pyra* [130] |
|---|---|---|---|---|---|---|---|---|---|---|---|
| **Yearly** SMAPE | **13.355** | 13.444 | 17.764 | 79.308 | 13.508 | 18.009 | 13.717 | 14.076 | 14.557 | 13.752 | 14.594 |
| **Yearly** MASE | **2.997** | 3.022 | 3.919 | 18.692 | 3.051 | 4.487 | 3.078 | 3.017 | 3.380 | 3.088 | 3.269 |
| **Yearly** OWA | **0.786** | 0.792 | 1.037 | 4.778 | 0.797 | 1.115 | 0.807 | 0.810 | 0.871 | 0.809 | 0.858 |
| **Quarterly** SMAPE | **10.060** | 10.252 | 13.968 | 74.943 | 10.706 | 13.376 | 10.958 | 10.711 | 11.408 | 10.900 | 11.654 |
| **Quarterly** MASE | **1.183** | 1.183 | 1.754 | 13.133 | 1.263 | 1.906 | 1.325 | 1.292 | 1.384 | 1.316 | 1.392 |
| **Quarterly** OWA | **0.872** | 0.897 | 1.274 | 8.191 | 0.947 | 1.302 | 0.981 | 0.957 | 1.022 | 0.975 | 1.037 |
| **Monthly** SMAPE | **12.608** | 12.798 | 18.200 | 68.892 | 13.925 | 14.588 | 13.917 | 13.362 | 13.803 | 13.949 | 14.963 |
| **Monthly** MASE | **0.925** | 0.957 | 1.574 | 11.199 | 1.062 | 1.368 | 1.097 | 1.016 | 1.078 | 1.096 | 1.165 |
| **Monthly** OWA | **0.872** | 0.894 | 1.371 | 7.654 | 0.982 | 1.149 | 0.998 | 0.941 | 0.985 | 0.999 | 1.066 |
| **Others** SMAPE | 4.941 | 5.324 | 6.738 | 176.164 | **4.888** | 7.267 | 6.302 | 5.387 | 6.090 | 6.611 | 5.605 |
| **Others** MASE | 3.327 | 3.410 | 4.853 | 116.723 | **3.244** | 5.240 | 4.064 | 3.670 | 4.203 | 4.492 | 3.966 |
| **Others** OWA | 1.045 | 1.098 | 1.474 | 36.941 | **1.026** | 1.591 | 1.304 | 1.146 | 1.304 | 1.404 | 1.215 |
| **Average** SMAPE | **11.785** | 11.962 | 16.511 | 78.103 | 12.605 | 14.718 | 12.780 | 12.491 | 13.016 | 12.805 | 13.616 |
| **Average** MASE | **1.584** | 1.609 | 2.321 | 18.663 | 1.677 | 2.408 | 1.756 | 1.675 | 1.837 | 1.777 | 1.843 |
| **Average** OWA | **0.849** | 0.862 | 1.215 | 7.759 | 0.903 | 1.172 | 0.930 | 0.899 | 0.960 | 0.937 | 0.984 |

Table 26: Full results for time series classification task compared with (1) classical methods: DTW [144], XGBoost [145], Rocket [84]; (2) RNN-based methods: LSTM [146], LSTNet [104], LSSL [147]; (3) CNN-based methods: InceptionTime (InTime) [85], TCN [148], TimesNet [72], TSLANet [83]. We report the classification accuracy (%) as the result. **Red**: best, Blue: second best. The standard deviation is within 1%.

| Datasets / Methods | Classical Methods | | | RNN-based | | | CNN-based | | | | SymTime |
|---|---|---|---|---|---|---|---|---|---|---|---|
| | DTW [144] | XGBoost [145] | Rocket [84] | LSTM [146] | LSTNet [104] | LSSL [147] | InTime [85] | TCN [148] | TimesNet [72] | TSLANet [83] | (**Ours**) |
| EthanolConcentration | 32.3 | 43.7 | **45.2** | 32.3 | 39.9 | 31.1 | 39.1 | 28.9 | 35.7 | 30.4 | 37.3 |
| FaceDetection | 52.9 | 63.3 | 64.7 | 57.7 | 65.7 | 66.7 | 65.4 | 52.8 | 68.6 | 66.7 | **69.2** |
| Handwriting | 28.6 | 15.8 | **58.8** | 15.2 | 25.8 | 24.6 | 46.9 | 53.3 | 32.1 | 57.9 | 36.7 |
| Heartbeat | 71.7 | 73.2 | 75.6 | 72.2 | 77.1 | 72.7 | 75.2 | 75.6 | **78.0** | 77.5 | 74.1 |
| JapaneseVowels | 94.9 | 86.5 | 96.2 | 79.7 | 98.1 | 98.4 | 95.1 | **98.9** | 98.4 | 95.1 | 98.1 |
| PEMS-SF | 71.1 | **98.3** | 75.1 | 39.9 | 86.7 | 86.1 | 79.6 | 68.8 | 89.6 | 83.8 | 97.1 |
| SelfRegulationSCP1 | 77.7 | 84.6 | 90.8 | 68.9 | 84.0 | 90.8 | 87.2 | 84.6 | **91.8** | **91.8** | 89.8 |
| SelfRegulationSCP2 | 53.9 | 48.9 | 53.3 | 46.6 | 52.8 | 52.2 | 53.6 | 55.6 | 57.2 | 53.3 | **58.9** |
| SpokenArabicDigits | 96.3 | 69.6 | 71.2 | 31.9 | **100.0** | **100.0** | 96.3 | 95.6 | 99.0 | 98.0 | 98.9 |
| UWaveGestureLibrary | 90.3 | 75.9 | **94.4** | 41.2 | 87.8 | 85.9 | 92.4 | 88.4 | 85.3 | 89.4 | 89.4 |
| Average Accuracy | 67.0 | 66.0 | 72.5 | 48.6 | 71.8 | 70.9 | 73.1 | 70.3 | 73.6 | 74.4 | **74.9** |

Table 27: Full reuslts for time series classification task compared with (1) Transformer-based methods: Autoformer [75], FEDformer [77], ETSformer [76], Informer [80], iTransformer [74], PatchTST (Patch) [54], GPT4TS (GPT) [2], UniTS [92], Peri-midformer [63] and (2) MLP-based methods: DLinear [87], LightTS [88]. We report the classification accuracy (%) as the results. (* means former.) **Red**: best, Blue: second best. The standard deviation is within 1%.

| Datasets / Methods | Transformer-based | | | | | | | | | MLP-based | | SymTime |
|---|---|---|---|---|---|---|---|---|---|---|---|---|
| | Auto* [75] | FED* [77] | ETS* [76] | In* [80] | iTrans* [74] | Patch [54] | GPT [2] | UniTS [92] | Peri-mid* [63] | DLinear [87] | LightTS [88] | (**Ours**) |
| EthanolConcentration | 31.6 | 31.2 | 28.1 | 31.6 | 27.0 | 29.6 | 34.2 | 37.3 | **47.3** | 32.6 | 29.7 | 37.3 |
| FaceDetection | 68.4 | 66.0 | 66.3 | 67.0 | 67.0 | 67.8 | **69.2** | 67.5 | 68.7 | 68.0 | 67.5 | **69.2** |
| Handwriting | **36.7** | 28.0 | 32.5 | 32.8 | 27.2 | 23.2 | 32.7 | 27.0 | 31.5 | 27.0 | 26.1 | **36.7** |
| Heartbeat | 74.6 | 73.7 | 71.2 | 80.5 | 75.6 | 75.7 | 77.2 | 80.5 | **86.3** | 75.1 | 75.1 | 74.1 |
| JapaneseVowels | 96.2 | 98.4 | 95.9 | **98.9** | 97.6 | 94.0 | 98.6 | 97.8 | 96.8 | 96.2 | 96.2 | 98.1 |
| PEMS-SF | 82.7 | 80.9 | 86.0 | 81.5 | 85.5 | 80.9 | 87.9 | 93.1 | 88.2 | 75.1 | 88.4 | **97.1** |
| SelfRegulationSCP1 | 84.0 | 88.7 | 89.6 | 90.1 | **92.2** | 82.2 | 87.2 | 89.6 | 87.4 | 87.3 | 89.8 | 89.8 |
| SelfRegulationSCP2 | 50.6 | 54.4 | 55.0 | 53.3 | 54.4 | 53.6 | 59.4 | **61.1** | 55.4 | 50.5 | 51.1 | 58.9 |
| SpokenArabicDigits | **100.0** | **100.0** | **100.0** | **100.0** | 98.0 | 98.0 | 95.2 | 98.9 | 98.0 | 81.4 | **100.0** | 98.9 |
| UWaveGestureLibrary | 85.9 | 85.3 | 85.0 | 85.6 | 85.9 | 81.7 | 85.1 | 87.8 | 84.3 | 82.1 | 80.3 | **89.4** |

Table 28: Full results for time series imputation task, where we randomly mask {12.5%, 25%, 37.5%, 50%} time points of length-96 time series to compare the model performance under different missing degrees. We compare with GPT4TS [2], TimesNet [72], Peri-midFormer [63], Moment [93], iTransformer [74], PatchTST [54], DLinear [87] in this table. (* means former.) The standard deviation is within 0.5%. **Red**: best, Blue: second best.

| Models | | SymTime (Ours) | | GPT4TS [2] | | TimesNet [72] | | Peri-mid* [63] | | Moment [93] | | iTrans* [74] | | PatchTST [54] | | DLinear [87] | |
|---|---|---|---|---|---|---|---|---|---|---|---|---|---|---|---|---|---|
| Mask Ratio | | MSE | MAE | MSE | MAE | MSE | MAE | MSE | MAE | MSE | MAE | MSE | MAE | MSE | MAE | MSE | MAE |
| ETTm1 | 12.5% | 0.032 | 0.110 | 0.018 | 0.090 | 0.019 | 0.091 | 0.032 | 0.109 | 0.069 | 0.170 | 0.046 | 0.147 | 0.045 | 0.137 | 0.056 | 0.162 |
| | 25% | 0.034 | 0.113 | 0.024 | 0.102 | 0.024 | 0.101 | 0.034 | 0.112 | 0.071 | 0.169 | 0.060 | 0.171 | 0.046 | 0.139 | 0.077 | 0.191 |
| | 37.5% | 0.037 | 0.118 | 0.029 | 0.111 | 0.029 | 0.112 | 0.037 | 0.117 | 0.069 | 0.163 | 0.077 | 0.195 | 0.049 | 0.143 | 0.100 | 0.218 |
| | 50% | 0.041 | 0.126 | 0.042 | 0.132 | 0.036 | 0.124 | 0.042 | 0.126 | 0.086 | 0.169 | 0.104 | 0.228 | 0.055 | 0.152 | 0.129 | 0.247 |
| | Avg | 0.036 | 0.116 | 0.028 | 0.109 | **0.027** | **0.107** | 0.036 | 0.116 | 0.074 | 0.168 | 0.072 | 0.185 | 0.049 | 0.143 | 0.090 | 0.204 |
| ETTm2 | 12.5% | 0.024 | 0.084 | 0.018 | 0.081 | 0.019 | 0.081 | 0.023 | 0.081 | 0.032 | 0.108 | 0.052 | 0.151 | 0.026 | 0.094 | 0.067 | 0.171 |
| | 25% | 0.024 | 0.086 | 0.021 | 0.082 | 0.021 | 0.086 | 0.024 | 0.084 | 0.029 | 0.105 | 0.070 | 0.179 | 0.028 | 0.099 | 0.089 | 0.200 |
| | 37.5% | 0.027 | 0.089 | 0.023 | 0.090 | 0.023 | 0.091 | 0.026 | 0.089 | 0.032 | 0.109 | 0.091 | 0.204 | 0.031 | 0.104 | 0.112 | 0.226 |
| | 50% | 0.030 | 0.093 | 0.027 | 0.098 | 0.026 | 0.098 | 0.030 | 0.095 | 0.031 | 0.110 | 0.117 | 0.232 | 0.034 | 0.109 | 0.140 | 0.253 |
| | Avg | 0.026 | 0.088 | **0.022** | **0.088** | 0.022 | 0.089 | 0.026 | 0.087 | 0.031 | 0.108 | 0.082 | 0.191 | 0.030 | 0.101 | 0.102 | 0.212 |
| ETTh1 | 12.5% | 0.074 | 0.179 | 0.063 | 0.171 | 0.062 | 0.169 | 0.069 | 0.173 | 0.160 | 0.239 | 0.098 | 0.220 | 0.097 | 0.203 | 0.111 | 0.232 |
| | 25% | 0.082 | 0.190 | 0.080 | 0.190 | 0.081 | 0.191 | 0.079 | 0.185 | 0.142 | 0.238 | 0.125 | 0.249 | 0.115 | 0.221 | 0.149 | 0.269 |
| | 37.5% | 0.100 | 0.205 | 0.107 | 0.218 | 0.098 | 0.210 | 0.096 | 0.202 | 0.121 | 0.228 | 0.156 | 0.278 | 0.134 | 0.239 | 0.187 | 0.301 |
| | 50% | 0.123 | 0.230 | 0.121 | 0.221 | 0.116 | 0.227 | 0.122 | 0.226 | 0.132 | 0.231 | 0.213 | 0.327 | 0.160 | 0.260 | 0.229 | 0.332 |
| | Avg | 0.095 | 0.201 | 0.093 | 0.200 | **0.089** | 0.199 | 0.091 | **0.196** | 0.139 | 0.234 | 0.148 | 0.269 | 0.126 | 0.231 | 0.169 | 0.283 |
| ETTh2 | 12.5% | 0.051 | 0.138 | 0.041 | 0.129 | 0.040 | 0.132 | 0.051 | 0.139 | 0.051 | 0.150 | 0.095 | 0.210 | 0.058 | 0.153 | 0.109 | 0.223 |
| | 25% | 0.055 | 0.146 | 0.046 | 0.138 | 0.047 | 0.144 | 0.054 | 0.142 | 0.079 | 0.177 | 0.120 | 0.239 | 0.063 | 0.160 | 0.146 | 0.260 |
| | 37.5% | 0.059 | 0.152 | 0.060 | 0.160 | 0.054 | 0.154 | 0.058 | 0.148 | 0.056 | 0.155 | 0.149 | 0.266 | 0.068 | 0.167 | 0.180 | 0.290 |
| | 50% | 0.064 | 0.157 | 0.061 | 0.160 | 0.061 | 0.164 | 0.064 | 0.159 | 0.056 | 0.154 | 0.192 | 0.302 | 0.074 | 0.175 | 0.217 | 0.319 |
| | Avg | 0.058 | 0.148 | 0.052 | **0.147** | **0.050** | 0.148 | 0.057 | **0.147** | 0.061 | 0.159 | 0.139 | 0.254 | 0.066 | 0.164 | 0.163 | 0.273 |
| ECL | 12.5% | 0.037 | 0.122 | 0.080 | 0.195 | 0.088 | 0.203 | 0.047 | 0.140 | 0.095 | 0.211 | 0.073 | 0.190 | 0.061 | 0.170 | 0.084 | 0.206 |
| | 25% | 0.046 | 0.139 | 0.089 | 0.205 | 0.092 | 0.208 | 0.053 | 0.162 | 0.093 | 0.211 | 0.090 | 0.214 | 0.072 | 0.185 | 0.113 | 0.243 |
| | 37.5% | 0.060 | 0.160 | 0.094 | 0.217 | 0.096 | 0.214 | 0.067 | 0.179 | 0.094 | 0.211 | 0.107 | 0.235 | 0.082 | 0.198 | 0.141 | 0.273 |
| | 50% | 0.075 | 0.181 | 0.108 | 0.231 | 0.102 | 0.221 | 0.085 | 0.195 | 0.092 | 0.210 | 0.127 | 0.257 | 0.097 | 0.216 | 0.173 | 0.303 |
| | Avg | **0.054** | **0.151** | 0.093 | 0.212 | 0.094 | 0.211 | 0.063 | 0.169 | 0.094 | 0.211 | 0.099 | 0.224 | 0.078 | 0.192 | 0.128 | 0.256 |
| Weather | 12.5% | 0.025 | 0.035 | 0.026 | 0.047 | 0.026 | 0.049 | 0.025 | 0.037 | 0.033 | 0.073 | 0.038 | 0.087 | 0.028 | 0.049 | 0.039 | 0.091 |
| | 25% | 0.027 | 0.037 | 0.030 | 0.055 | 0.030 | 0.056 | 0.026 | 0.037 | 0.036 | 0.078 | 0.046 | 0.106 | 0.032 | 0.055 | 0.049 | 0.112 |
| | 37.5% | 0.029 | 0.039 | 0.033 | 0.061 | 0.032 | 0.058 | 0.029 | 0.041 | 0.034 | 0.075 | 0.055 | 0.122 | 0.035 | 0.059 | 0.057 | 0.125 |
| | 50% | 0.032 | 0.042 | 0.039 | 0.070 | 0.034 | 0.062 | 0.034 | 0.048 | 0.035 | 0.075 | 0.068 | 0.142 | 0.039 | 0.064 | 0.067 | 0.139 |
| | Avg | **0.028** | **0.038** | 0.032 | 0.058 | 0.030 | 0.056 | 0.029 | 0.041 | 0.035 | 0.075 | 0.052 | 0.114 | 0.033 | 0.057 | 0.053 | 0.116 |
| Average | | **0.049** | **0.124** | 0.053 | 0.136 | 0.052 | 0.135 | **0.050** | 0.126 | 0.072 | 0.159 | 0.099 | 0.206 | 0.064 | 0.148 | 0.118 | 0.224 |

Table 29: Full results for time series imputation task, where we randomly mask {12.5%, 25%, 37.5%, 50%} time points of length-96 time series to compare the model performance under different missing degrees. We compare with Stationary [78], LightTS [88], ETSformer [76], FEDformer [77], Informer [80], Reformer [143] and Pyraformer [130] in this table. (Stationary means Nonstationary Transformer.) The standard deviation is within 0.5%. **Red**: best, Blue: second best.

| Methods | | SymTime (Ours) | | Stationary [78] | | LightTS [88] | | ETSformer [76] | | FEDformer [77] | | Informer [80] | | Reformer [143] | | Pyraformer [130] | |
|---|---|---|---|---|---|---|---|---|---|---|---|---|---|---|---|---|---|
| Mask Ratio | | MSE | MAE | MSE | MAE | MSE | MAE | MSE | MAE | MSE | MAE | MSE | MAE | MSE | MAE | MSE | MAE |
| ETTm1 | 12.5% | 0.032 | 0.110 | 0.026 | 0.107 | 0.054 | 0.158 | 0.034 | 0.130 | 0.068 | 0.188 | 0.027 | 0.115 | 0.032 | 0.126 | 0.670 | 0.541 |
| | 25% | 0.034 | 0.113 | 0.032 | 0.119 | 0.061 | 0.173 | 0.053 | 0.162 | 0.097 | 0.230 | 0.040 | 0.140 | 0.042 | 0.146 | 0.689 | 0.553 |
| | 37.5% | 0.037 | 0.118 | 0.039 | 0.131 | 0.073 | 0.189 | 0.082 | 0.201 | 0.134 | 0.287 | 0.071 | 0.189 | 0.063 | 0.182 | 0.737 | 0.581 |
| | 50% | 0.041 | 0.126 | 0.047 | 0.145 | 0.086 | 0.207 | 0.130 | 0.257 | 0.188 | 0.323 | 0.091 | 0.208 | 0.082 | 0.208 | 0.770 | 0.605 |
| | Avg | **0.036** | **0.116** | **0.036** | 0.126 | 0.068 | 0.182 | 0.075 | 0.187 | 0.121 | 0.257 | 0.057 | 0.163 | 0.055 | 0.166 | 0.717 | 0.570 |
| ETTm2 | 12.5% | 0.024 | 0.084 | 0.021 | 0.088 | 0.051 | 0.150 | 0.061 | 0.169 | 0.109 | 0.239 | 0.196 | 0.326 | 0.108 | 0.228 | 0.394 | 0.470 |
| | 25% | 0.024 | 0.086 | 0.024 | 0.096 | 0.069 | 0.176 | 0.093 | 0.214 | 0.166 | 0.295 | 0.295 | 0.414 | 0.136 | 0.262 | 0.421 | 0.482 |
| | 37.5% | 0.027 | 0.089 | 0.027 | 0.103 | 0.074 | 0.185 | 0.137 | 0.253 | 0.237 | 0.356 | 0.155 | 0.293 | 0.175 | 0.300 | 0.478 | 0.521 |
| | 50% | 0.030 | 0.093 | 0.030 | 0.108 | 0.078 | 0.192 | 0.237 | 0.332 | 0.323 | 0.412 | 0.214 | 0.325 | 0.211 | 0.329 | 0.568 | 0.560 |
| | Avg | **0.026** | **0.088** | **0.026** | 0.099 | 0.068 | 0.176 | 0.132 | 0.242 | 0.209 | 0.326 | 0.215 | 0.340 | 0.157 | 0.280 | 0.465 | 0.508 |
| ETTh1 | 12.5% | 0.074 | 0.179 | 0.060 | 0.165 | 0.119 | 0.239 | 0.073 | 0.195 | 0.126 | 0.265 | 0.068 | 0.187 | 0.074 | 0.194 | 0.857 | 0.609 |
| | 25% | 0.082 | 0.190 | 0.080 | 0.189 | 0.144 | 0.266 | 0.105 | 0.234 | 0.169 | 0.305 | 0.096 | 0.220 | 0.102 | 0.227 | 0.829 | 0.672 |
| | 37.5% | 0.100 | 0.205 | 0.102 | 0.212 | 0.171 | 0.292 | 0.144 | 0.276 | 0.220 | 0.348 | 0.128 | 0.253 | 0.135 | 0.261 | 0.830 | 0.675 |
| | 50% | 0.123 | 0.230 | 0.133 | 0.240 | 0.201 | 0.317 | 0.200 | 0.327 | 0.298 | 0.403 | 0.166 | 0.287 | 0.179 | 0.298 | 0.854 | 0.691 |
| | Avg | 0.095 | **0.201** | **0.094** | **0.201** | 0.159 | 0.278 | 0.130 | 0.258 | 0.204 | 0.330 | 0.115 | 0.237 | 0.122 | 0.245 | 0.842 | 0.682 |
| ETTh2 | 12.5% | 0.051 | 0.138 | 0.042 | 0.133 | 0.094 | 0.208 | 0.134 | 0.251 | 0.187 | 0.319 | 0.271 | 0.384 | 0.163 | 0.289 | 0.976 | 0.754 |
| | 25% | 0.055 | 0.146 | 0.049 | 0.147 | 0.140 | 0.255 | 0.180 | 0.294 | 0.279 | 0.396 | 0.362 | 0.450 | 0.206 | 0.331 | 1.037 | 0.774 |
| | 37.5% | 0.059 | 0.152 | 0.056 | 0.158 | 0.159 | 0.274 | 0.243 | 0.341 | 0.402 | 0.465 | 0.401 | 0.469 | 0.252 | 0.370 | 1.107 | 0.800 |
| | 50% | 0.064 | 0.157 | 0.065 | 0.170 | 0.180 | 0.293 | 0.353 | 0.408 | 0.604 | 0.504 | 0.437 | 0.487 | 0.316 | 0.419 | 1.193 | 0.838 |
| | Avg | 0.058 | **0.148** | **0.053** | 0.152 | 0.143 | 0.258 | 0.228 | 0.324 | 0.368 | 0.421 | 0.368 | 0.448 | 0.234 | 0.352 | 1.079 | 0.792 |
| ECL | 12.5% | 0.037 | 0.122 | 0.093 | 0.210 | 0.077 | 0.198 | 0.185 | 0.323 | 0.197 | 0.324 | 0.152 | 0.279 | 0.190 | 0.308 | 0.297 | 0.383 |
| | 25% | 0.046 | 0.139 | 0.097 | 0.214 | 0.099 | 0.228 | 0.207 | 0.340 | 0.208 | 0.345 | 0.166 | 0.290 | 0.197 | 0.312 | 0.294 | 0.380 |
| | 37.5% | 0.060 | 0.160 | 0.102 | 0.220 | 0.120 | 0.252 | 0.226 | 0.355 | 0.219 | 0.337 | 0.178 | 0.297 | 0.203 | 0.315 | 0.296 | 0.381 |
| | 50% | 0.075 | 0.181 | 0.108 | 0.228 | 0.138 | 0.272 | 0.251 | 0.372 | 0.235 | 0.357 | 0.189 | 0.305 | 0.210 | 0.319 | 0.299 | 0.383 |
| | Avg | **0.049** | **0.151** | 0.100 | 0.218 | 0.108 | 0.238 | 0.217 | 0.347 | 0.215 | 0.341 | 0.171 | 0.293 | 0.200 | 0.313 | 0.297 | 0.382 |
| Weather | 12.5% | 0.025 | 0.035 | 0.027 | 0.051 | 0.039 | 0.092 | 0.042 | 0.103 | 0.057 | 0.141 | 0.040 | 0.108 | 0.031 | 0.076 | 0.140 | 0.220 |
| | 25% | 0.027 | 0.037 | 0.029 | 0.056 | 0.045 | 0.105 | 0.056 | 0.131 | 0.066 | 0.155 | 0.045 | 0.130 | 0.035 | 0.082 | 0.147 | 0.229 |
| | 37.5% | 0.029 | 0.039 | 0.033 | 0.062 | 0.049 | 0.110 | 0.081 | 0.180 | 0.083 | 0.180 | 0.049 | 0.101 | 0.040 | 0.091 | 0.156 | 0.240 |
| | 50% | 0.032 | 0.042 | 0.037 | 0.068 | 0.054 | 0.117 | 0.102 | 0.207 | 0.103 | 0.207 | 0.054 | 0.114 | 0.046 | 0.099 | 0.164 | 0.249 |
| | Avg | **0.028** | **0.038** | 0.032 | 0.059 | 0.047 | 0.106 | 0.071 | 0.155 | 0.077 | 0.171 | 0.047 | 0.113 | 0.038 | 0.087 | 0.152 | 0.235 |
| Average | | **0.049** | **0.124** | 0.057 | 0.143 | 0.099 | 0.206 | 0.142 | 0.252 | 0.199 | 0.308 | 0.162 | 0.265 | 0.134 | 0.241 | 0.592 | 0.528 |

Table 30: Full reuslts for time series anomaly detection task, where P, R and F1 represent the precision, recall and F1-score (%) respectively. F1-score is the harmonic mean of precision and recall. A higher value of P, R and F1 indicates a better performance. We compare with: Transformer [127], Reformer [143], Informer [80], Autoformer [75], Crossformer [79], iTransformer [74], Anomaly [81], Stationary [78], DLinear [87], LightTS [88], ETSformer [76], FEDformer [77], PatchTST [54], TimesNet [72], GPT4TS [2], Peri-midFormer [63], UniTS [92], where Anomaly means the Anomaly Transformer and Stationary means the Non-stationary Transformer. The standard deviation is within 1%. **Red**: best, Blue: second best.

| Datasets | SMD | | | MSL | | | SMAP | | | SWaT | | | PSM | | | Avg F1 |
|---|---|---|---|---|---|---|---|---|---|---|---|---|---|---|---|---|
| Metircs | P | R | F1 | P | R | F1 | P | R | F1 | P | R | F1 | P | R | F1 | (%) |
| LSTM [146] | 78.52 | 65.47 | 71.41 | 78.04 | 86.22 | 81.93 | 91.06 | 57.49 | 70.48 | 78.06 | 91.72 | 84.34 | 69.24 | 99.53 | 81.67 | 77.97 |
| LogTrans [127] | 83.46 | 70.13 | 76.21 | 73.05 | 87.37 | 79.57 | 89.15 | 57.59 | 69.97 | 68.67 | 97.32 | 80.52 | 63.06 | 98.00 | 76.74 | 76.60 |
| Transformer [127] | 78.44 | 65.26 | 71.24 | 89.85 | 73.71 | 80.99 | 90.77 | 61.76 | 73.50 | 96.82 | 66.41 | 79.76 | 99.31 | 83.18 | 90.53 | 79.20 |
| TCN [148] | 84.06 | 79.07 | 81.49 | 75.11 | 82.44 | 78.60 | 86.90 | 59.23 | 70.45 | 76.59 | 95.71 | 85.09 | 54.59 | 99.77 | 70.57 | 77.24 |
| LSSL [147] | 78.51 | 65.32 | 71.31 | 77.55 | 88.18 | 82.53 | 89.43 | 53.43 | 66.90 | 79.05 | 93.72 | 85.76 | 66.02 | 92.93 | 77.20 | 76.74 |
| Reformer [143] | 72.50 | 84.19 | 77.90 | 90.24 | 73.78 | 81.18 | 90.63 | 62.48 | 73.97 | 99.94 | 66.75 | 80.04 | 99.73 | 83.03 | 90.62 | 80.74 |
| Informer [80] | 72.51 | 84.13 | 77.88 | 90.10 | 73.68 | 81.07 | 90.57 | 61.51 | 73.26 | 99.83 | 67.24 | 80.35 | 99.03 | 83.21 | 90.43 | 80.60 |
| Autoformer [75] | 78.46 | 65.11 | 71.17 | 90.59 | 75.26 | 82.22 | 90.84 | 62.39 | 73.97 | 99.95 | 65.57 | 79.19 | 99.99 | 78.96 | 88.24 | 78.96 |
| Crossformer [79] | 71.89 | 83.41 | 77.22 | 90.32 | 72.74 | 80.59 | 89.68 | 53.63 | 67.12 | 98.00 | 83.59 | 90.22 | 97.49 | 88.02 | 92.52 | 81.53 |
| iTransformer [74] | 76.13 | 84.70 | 80.19 | 86.15 | 62.54 | 72.47 | 90.68 | 52.78 | 66.72 | 92.23 | 93.05 | 92.64 | 97.92 | 92.03 | 94.88 | 81.38 |
| Pyraformer [130] | 85.61 | 80.61 | 83.04 | 83.81 | 85.93 | 84.86 | 92.54 | 57.71 | 71.09 | 87.92 | 96.00 | 91.78 | 71.67 | 96.02 | 82.08 | 82.57 |
| Anomaly [81] | 88.91 | 82.23 | 85.49 | 79.61 | 87.37 | 83.31 | 91.85 | 58.11 | 71.18 | 72.51 | 97.32 | 83.10 | 68.35 | 94.72 | 79.40 | 80.50 |
| Stationary [78] | 78.51 | 87.98 | 82.97 | 86.86 | 68.63 | 76.68 | 90.62 | 55.74 | 69.02 | 89.26 | 95.42 | 92.24 | 98.17 | 96.30 | 97.23 | 83.63 |
| DLinear [87] | 75.91 | 84.02 | 79.76 | 89.68 | 75.31 | 81.87 | 89.87 | 53.79 | 67.30 | 92.26 | 93.05 | 92.66 | 98.65 | 94.70 | 96.64 | 83.64 |
| LightTS [88] | 87.10 | 78.42 | 82.53 | 82.40 | 75.78 | 78.95 | 92.58 | 55.27 | 69.21 | 91.98 | 94.72 | 93.33 | 98.37 | 95.97 | 97.15 | 84.23 |
| ETSformer [76] | 87.44 | 79.23 | 83.13 | 85.13 | 84.93 | 85.03 | 92.25 | 55.75 | 69.50 | 90.02 | 80.36 | 84.91 | 99.31 | 85.28 | 91.76 | 82.87 |
| FEDformer [77] | 72.82 | 81.68 | 76.99 | 90.72 | 75.41 | 82.36 | 90.47 | 58.10 | 70.76 | 99.95 | 65.55 | 79.18 | 99.98 | 81.92 | 90.05 | 79.46 |
| PatchTST [54] | 87.26 | 82.14 | 84.62 | 88.34 | 70.96 | 78.70 | 90.64 | 55.46 | 68.82 | 91.10 | 80.94 | 85.72 | 98.84 | 93.47 | 96.08 | 82.79 |
| TimesNet [72] | 88.07 | 80.97 | 84.37 | 88.83 | 74.68 | 81.14 | 89.98 | 56.02 | 69.05 | 91.99 | 93.24 | 92.61 | 98.46 | 95.70 | 97.06 | 84.85 |
| GPT4TS [2] | 87.68 | 81.52 | 84.49 | 82.09 | 81.97 | 82.03 | 90.12 | 55.70 | 68.85 | 92.12 | 93.09 | 92.60 | 98.36 | 95.85 | 97.09 | 85.01 |
| FedTADBench [149] | 87.31 | 80.06 | 83.53 | 77.69 | 69.37 | 84.09 | 90.49 | 57.44 | 70.27 | 90.63 | 84.43 | 87.42 | 97.67 | 94.41 | 96.01 | 84.26 |
| PeFAD [150] | 88.64 | 82.05 | 85.22 | 73.42 | 87.31 | 78.94 | 89.89 | 62.39 | 73.66 | 88.71 | 89.78 | 88.73 | 96.93 | 95.94 | 96.43 | 84.60 |
| InterFusion [151] | 84.06 | 83.52 | 83.78 | 84.83 | 78.49 | 81.54 | 90.66 | 58.11 | 70.82 | 96.76 | 78.45 | 86.65 | 83.61 | 83.45 | 83.53 | 81.26 |
| Peri-midFormer [63] | 86.97 | 81.37 | 84.08 | 88.66 | 74.02 | 80.68 | 90.02 | 54.03 | 67.53 | 90.74 | 92.55 | 91.64 | 98.46 | 94.06 | 96.21 | 84.03 |
| UniTS [92] | 82.42 | 84.99 | 83.69 | 91.32 | 73.04 | 81.16 | 90.58 | 62.55 | 74.00 | 92.60 | 92.42 | 92.51 | 98.45 | 96.19 | 97.31 | 85.73 |
| SymTime (**Ours**) | 88.08 | 83.37 | 85.66 | 89.46 | 75.31 | 81.77 | 91.06 | 61.51 | 73.43 | 95.94 | 91.39 | 93.61 | 98.90 | 95.36 | 97.10 | 86.31 |

Table 31: The full fine-tuning results of the time series long-term forecasting task under different sizes of pre-training datasets. The brief results of this experiment are shown in Table 1. **Red**: best, Blue: second best.

| Methods | | 0B | | 1B | | 10B | | 25B | | 50B | |
|---|---|---|---|---|---|---|---|---|---|---|---|
| Datasets \Horizon | | MSE | MAE | MSE | MAE | MSE | MAE | MSE | MAE | MSE | MAE |
| ETTm1 | 96 | 0.335 | 0.376 | 0.324 | 0.355 | 0.319 | 0.353 | 0.321 | 0.358 | 0.318 | 0.353 |
| | 192 | 0.404 | 0.382 | 0.363 | 0.385 | 0.374 | 0.382 | 0.369 | 0.383 | 0.362 | 0.380 |
| | 336 | 0.419 | 0.443 | 0.392 | 0.410 | 0.391 | 0.412 | 0.394 | 0.407 | 0.386 | 0.402 |
| | 720 | 0.446 | 0.435 | 0.426 | 0.440 | 0.422 | 0.423 | 0.426 | 0.423 | 0.419 | 0.423 |
| | Avg | 0.401 | 0.409 | 0.376 | 0.398 | 0.376 | 0.393 | 0.378 | 0.393 | 0.371 | 0.390 |
| ETTm2 | 96 | 0.181 | 0.271 | 0.195 | 0.257 | 0.183 | 0.265 | 0.177 | 0.264 | 0.174 | 0.257 |
| | 192 | 0.241 | 0.323 | 0.268 | 0.316 | 0.249 | 0.310 | 0.245 | 0.302 | 0.238 | 0.299 |
| | 336 | 0.334 | 0.361 | 0.310 | 0.350 | 0.295 | 0.346 | 0.297 | 0.338 | 0.295 | 0.337 |
| | 720 | 0.417 | 0.401 | 0.394 | 0.400 | 0.398 | 0.396 | 0.391 | 0.396 | 0.390 | 0.392 |
| | Avg | 0.293 | 0.339 | 0.292 | 0.331 | 0.281 | 0.329 | 0.278 | 0.325 | 0.274 | 0.321 |
| ETTh1 | 96 | 0.422 | 0.428 | 0.402 | 0.411 | 0.386 | 0.404 | 0.384 | 0.402 | 0.376 | 0.400 |
| | 192 | 0.457 | 0.459 | 0.447 | 0.462 | 0.430 | 0.432 | 0.434 | 0.436 | 0.428 | 0.431 |
| | 336 | 0.523 | 0.494 | 0.489 | 0.467 | 0.473 | 0.463 | 0.468 | 0.457 | 0.463 | 0.456 |
| | 720 | 0.547 | 0.515 | 0.507 | 0.495 | 0.486 | 0.478 | 0.481 | 0.479 | 0.450 | 0.458 |
| | Avg | 0.487 | 0.474 | 0.461 | 0.459 | 0.444 | 0.444 | 0.434 | 0.438 | 0.430 | 0.436 |
| ETTh2 | 96 | 0.298 | 0.350 | 0.302 | 0.355 | 0.295 | 0.357 | 0.301 | 0.352 | 0.293 | 0.348 |
| | 192 | 0.373 | 0.403 | 0.364 | 0.416 | 0.369 | 0.405 | 0.369 | 0.403 | 0.364 | 0.397 |
| | 336 | 0.401 | 0.434 | 0.458 | 0.442 | 0.387 | 0.424 | 0.389 | 0.426 | 0.385 | 0.423 |
| | 720 | 0.433 | 0.461 | 0.490 | 0.464 | 0.454 | 0.447 | 0.427 | 0.438 | 0.420 | 0.441 |
| | Avg | 0.376 | 0.412 | 0.403 | 0.419 | 0.376 | 0.408 | 0.371 | 0.405 | 0.365 | 0.402 |
| Weather | 96 | 0.185 | 0.221 | 0.173 | 0.219 | 0.170 | 0.218 | 0.175 | 0.217 | 0.166 | 0.213 |
| | 192 | 0.217 | 0.279 | 0.223 | 0.259 | 0.232 | 0.263 | 0.221 | 0.263 | 0.212 | 0.254 |
| | 336 | 0.276 | 0.304 | 0.285 | 0.294 | 0.266 | 0.296 | 0.271 | 0.297 | 0.267 | 0.294 |
| | 720 | 0.353 | 0.353 | 0.347 | 0.357 | 0.332 | 0.340 | 0.345 | 0.350 | 0.342 | 0.344 |
| | Avg | 0.257 | 0.289 | 0.257 | 0.282 | 0.250 | 0.279 | 0.253 | 0.282 | 0.247 | 0.276 |
| ECL | 96 | 0.162 | 0.254 | 0.168 | 0.263 | 0.166 | 0.262 | 0.175 | 0.272 | 0.162 | 0.253 |
| | 192 | 0.179 | 0.268 | 0.182 | 0.270 | 0.173 | 0.269 | 0.183 | 0.276 | 0.173 | 0.264 |
| | 336 | 0.217 | 0.288 | 0.196 | 0.286 | 0.209 | 0.300 | 0.205 | 0.295 | 0.194 | 0.285 |
| | 720 | 0.216 | 0.324 | 0.250 | 0.321 | 0.236 | 0.312 | 0.216 | 0.308 | 0.220 | 0.304 |
| | Avg | 0.193 | 0.284 | 0.199 | 0.285 | 0.196 | 0.286 | 0.195 | 0.288 | 0.187 | 0.276 |
| Traffic | 96 | 0.460 | 0.288 | 0.451 | 0.289 | 0.438 | 0.293 | 0.442 | 0.282 | 0.432 | 0.280 |
| | 192 | 0.452 | 0.287 | 0.461 | 0.291 | 0.450 | 0.292 | 0.452 | 0.289 | 0.444 | 0.287 |
| | 336 | 0.461 | 0.330 | 0.469 | 0.296 | 0.472 | 0.294 | 0.467 | 0.296 | 0.458 | 0.293 |
| | 720 | 0.510 | 0.336 | 0.510 | 0.336 | 0.508 | 0.314 | 0.502 | 0.309 | 0.492 | 0.303 |
| | Avg | 0.471 | 0.310 | 0.473 | 0.303 | 0.473 | 0.294 | 0.467 | 0.299 | 0.457 | 0.291 |
| Exchange | 96 | 0.120 | 0.235 | 0.087 | 0.203 | 0.098 | 0.208 | 0.087 | 0.205 | 0.084 | 0.201 |
| | 192 | 0.188 | 0.298 | 0.192 | 0.309 | 0.186 | 0.301 | 0.177 | 0.297 | 0.174 | 0.295 |
| | 336 | 0.346 | 0.428 | 0.341 | 0.423 | 0.337 | 0.420 | 0.326 | 0.414 | 0.331 | 0.416 |
| | 720 | 0.878 | 0.699 | 0.861 | 0.706 | 0.850 | 0.701 | 0.838 | 0.689 | 0.847 | 0.694 |
| | Avg | 0.383 | 0.415 | 0.370 | 0.410 | 0.368 | 0.407 | 0.357 | 0.401 | 0.359 | 0.401 |
| Average | | 0.358 | 0.366 | 0.354 | 0.361 | 0.345 | 0.355 | 0.342 | 0.354 | 0.336 | 0.349 |

Table 32: The full fine-tuning results of the time series short-term forecasting task under different sizes of pre-training datasets. The brief results of this experiment are shown in Table 2. **Red**: best, Blue: second best.

| Methods | Metric | 0B | 1B | 10B | 25B | 50B |
|---|---|---|---|---|---|---|
| Yearly | SMAPE | **13.291** | 13.341 | 13.332 | 13.380 | 13.355 |
| | MASE | **2.981** | 2.986 | 2.985 | 3.012 | 2.997 |
| | OWA | **0.782** | 0.784 | 0.783 | 0.788 | 0.786 |
| Quartly | SMAPE | 10.270 | 10.274 | 10.197 | 10.228 | **10.060** |
| | MASE | 1.224 | 1.218 | 1.212 | 1.219 | **1.183** |
| | OWA | 0.913 | 0.911 | 0.905 | 0.909 | **0.872** |
| Monthly | SMAPE | 13.545 | 12.811 | 12.833 | 12.662 | **12.608** |
| | MASE | 1.053 | 0.955 | 0.959 | 0.932 | **0.925** |
| | OWA | 0.964 | 0.893 | 0.896 | 0.877 | **0.872** |
| Others | SMAPE | 5.186 | 5.070 | 5.003 | 5.034 | **4.941** |
| | MASE | 3.498 | 3.479 | 3.350 | 3.372 | **3.327** |
| | OWA | 1.097 | 1.082 | 1.055 | 1.061 | **1.045** |
| Avg. | SMAPE | 12.283 | 11.937 | 11.924 | 11.862 | **11.785** |
| | MASE | 1.660 | 1.611 | 1.605 | 1.601 | **1.584** |
| | OWA | 0.887 | 0.861 | 0.859 | 0.856 | **0.849** |

Table 33: The full fine-tuning results of the time series classification task under different sizes of pre-training datasets. The brief results of this experiment are shown in Figure 7 (a). **Red**: best, Blue: second best.

| Datasets | 0B | 1B | 10B | 25B | 50B |
|---|---|---|---|---|---|
| EthanolConcentration | 33.08 | 30.04 | 34.22 | 35.74 | **37.30** |
| FaceDetection | 51.31 | 58.12 | 58.12 | 58.12 | **69.20** |
| Handwriting | 35.76 | 36.24 | 36.24 | 36.00 | **36.70** |
| Heartbeat | 70.24 | 71.71 | 72.20 | 73.17 | **74.15** |
| JapaneseVowels | 94.86 | 95.95 | 94.86 | 96.76 | **98.11** |
| PEMS-SF | 86.71 | 88.44 | 88.44 | 92.49 | **97.11** |
| SelfRegulationSCP1 | 86.69 | 85.67 | 87.71 | 88.05 | **89.76** |
| SelfRegulationSCP2 | 56.11 | 57.78 | 57.78 | 58.89 | **58.89** |
| SpokenArabicDigits | 89.77 | 93.91 | 95.91 | 97.04 | **98.86** |
| UWaveGestureLibrary | 78.75 | 85.63 | 85.63 | 87.19 | **89.38** |
| Average Accuracy | 68.33 | 70.35 | 71.11 | 71.34 | **74.90** |

Table 34: The full fine-tuning results of the time series imputation task under different sizes of pre-training datasets. The brief results of this experiment are shown in Table 2. **Red**: best, Blue: second best.

| Methods | | 0B | | 1B | | 10B | | 25B | | 50B | |
|---------|-------|------|------|------|------|------|------|------|------|------|------|
| Mask | Ratio | MSE | MAE | MSE | MAE | MSE | MAE | MSE | MAE | MSE | MAE |
| ETTm1 | 12.5% | 0.034 | 0.111 | 0.034 | 0.112 | 0.032 | 0.111 | 0.033 | 0.110 | 0.032 | 0.110 |
| | 25% | 0.050 | 0.125 | 0.042 | 0.116 | 0.041 | 0.118 | 0.036 | 0.116 | 0.034 | 0.113 |
| | 37.5% | 0.040 | 0.122 | 0.037 | 0.119 | 0.039 | 0.118 | 0.036 | 0.118 | 0.037 | 0.118 |
| | 50% | 0.044 | 0.128 | 0.042 | 0.129 | 0.042 | 0.128 | 0.042 | 0.126 | 0.041 | 0.126 |
| | Avg | 0.042 | 0.122 | 0.039 | 0.119 | 0.038 | 0.119 | 0.037 | 0.118 | **0.036** | **0.117** |
| ETTm2 | 12.5% | 0.030 | 0.088 | 0.027 | 0.087 | 0.028 | 0.087 | 0.026 | 0.086 | 0.024 | 0.084 |
| | 25% | 0.038 | 0.101 | 0.029 | 0.087 | 0.029 | 0.087 | 0.028 | 0.088 | 0.024 | 0.086 |
| | 37.5% | 0.040 | 0.120 | 0.032 | 0.107 | 0.028 | 0.096 | 0.027 | 0.091 | 0.027 | 0.089 |
| | 50% | 0.045 | 0.114 | 0.036 | 0.109 | 0.034 | 0.110 | 0.032 | 0.106 | 0.030 | 0.093 |
| | Avg | 0.038 | 0.106 | 0.031 | 0.097 | 0.030 | 0.095 | 0.028 | 0.093 | **0.026** | **0.088** |
| ETTh1 | 12.5% | 0.100 | 0.208 | 0.085 | 0.185 | 0.087 | 0.186 | 0.080 | 0.180 | 0.074 | 0.179 |
| | 25% | 0.120 | 0.238 | 0.116 | 0.222 | 0.106 | 0.221 | 0.103 | 0.207 | 0.082 | 0.190 |
| | 37.5% | 0.118 | 0.234 | 0.118 | 0.227 | 0.100 | 0.206 | 0.103 | 0.209 | 0.100 | 0.205 |
| | 50% | 0.140 | 0.241 | 0.133 | 0.235 | 0.137 | 0.240 | 0.130 | 0.232 | 0.123 | 0.230 |
| | Avg | 0.112 | 0.230 | 0.113 | 0.217 | 0.107 | 0.213 | 0.104 | 0.207 | **0.095** | **0.201** |
| ETTh2 | 12.5% | 0.061 | 0.158 | 0.061 | 0.158 | 0.057 | 0.151 | 0.054 | 0.146 | 0.051 | 0.138 |
| | 25% | 0.066 | 0.161 | 0.058 | 0.153 | 0.059 | 0.155 | 0.058 | 0.154 | 0.055 | 0.146 |
| | 37.5% | 0.065 | 0.158 | 0.070 | 0.164 | 0.067 | 0.160 | 0.059 | 0.155 | 0.059 | 0.152 |
| | 50% | 0.068 | 0.164 | 0.073 | 0.167 | 0.068 | 0.166 | 0.066 | 0.161 | 0.064 | 0.157 |
| | Avg | 0.065 | 0.160 | 0.066 | 0.160 | 0.063 | 0.158 | 0.059 | 0.154 | **0.057** | **0.148** |
| ECL | 12.5% | 0.046 | 0.123 | 0.039 | 0.123 | 0.039 | 0.123 | 0.038 | 0.123 | 0.037 | 0.122 |
| | 25% | 0.046 | 0.148 | 0.049 | 0.140 | 0.048 | 0.139 | 0.047 | 0.139 | 0.046 | 0.139 |
| | 37.5% | 0.062 | 0.168 | 0.063 | 0.163 | 0.062 | 0.161 | 0.060 | 0.161 | 0.060 | 0.160 |
| | 50% | 0.076 | 0.181 | 0.076 | 0.182 | 0.075 | 0.182 | 0.076 | 0.183 | 0.075 | 0.181 |
| | Avg | 0.058 | 0.155 | 0.057 | 0.152 | 0.056 | **0.151** | **0.055** | 0.152 | **0.055** | **0.151** |
| Weather | 12.5% | 0.029 | 0.044 | 0.030 | 0.046 | 0.025 | 0.036 | 0.026 | 0.038 | 0.025 | 0.035 |
| | 25% | 0.038 | 0.054 | 0.030 | 0.044 | 0.036 | 0.052 | 0.029 | 0.042 | 0.027 | 0.037 |
| | 37.5% | 0.038 | 0.058 | 0.037 | 0.057 | 0.034 | 0.052 | 0.032 | 0.045 | 0.029 | 0.039 |
| | 50% | 0.040 | 0.057 | 0.036 | 0.054 | 0.036 | 0.052 | 0.034 | 0.046 | 0.032 | 0.042 |
| | Avg | 0.036 | 0.053 | 0.033 | 0.050 | 0.033 | 0.048 | 0.030 | 0.043 | **0.028** | **0.038** |
| Average | | 0.058 | 0.138 | 0.057 | 0.132 | 0.055 | 0.131 | 0.052 | 0.128 | **0.049** | **0.124** |

Table 35: The full fine-tuning results of the time series classification task under different sizes of pre-training datasets. The brief results of this experiment are shown in Figure 7 (b). **Red**: best, Blue: second best.

| Datasets | 0B | | | 1B | | | 10B | | | 25B | | | 50B | | |
|---|---|---|---|---|---|---|---|---|---|---|---|---|---|---|---|
| Metrics | P | R | F1 | P | R | F1 | P | R | F1 | P | R | F1 | P | R | F1 |
| SMD | 86.70 | 80.72 | 83.60 | 88.08 | 83.37 | 85.66 | 87.46 | 81.05 | 84.13 | 86.89 | 82.02 | 84.39 | 88.08 | 83.37 | **85.66** |
| MSL | 89.28 | 73.68 | 80.74 | 89.13 | 73.59 | 80.62 | 89.48 | 74.59 | 81.36 | 89.34 | 73.91 | 80.90 | 89.46 | 75.31 | **81.77** |
| SMAP | 89.97 | 54.16 | 67.61 | 90.06 | 53.56 | 67.17 | 90.08 | 54.51 | 67.92 | 90.11 | 54.63 | 68.02 | 91.06 | 61.51 | **73.43** |
| SWaT | 91.57 | 85.45 | 88.40 | 92.21 | 92.70 | 92.45 | 92.23 | 92.77 | 92.50 | 92.28 | 92.87 | 92.57 | 95.94 | 91.39 | **93.61** |
| PSM | 98.69 | 94.24 | 96.41 | 98.53 | 94.30 | 96.37 | 98.65 | 94.41 | 96.48 | 98.79 | 94.13 | 96.40 | 91.39 | 93.61 | **98.90** |
| Avg. | 91.24 | 77.65 | 83.40 | 91.60 | 79.50 | 84.45 | 91.58 | 79.47 | 84.48 | 91.48 | 79.51 | 84.46 | 95.36 | 97.10 | **86.31** |

