# OpenReview forum: "Synthetic Series-Symbol Data Generation for Time Series Foundation Models"
_NeurIPS.cc/2025/Conference — NeurIPS 2025 poster_

### Official Review · Reviewer_SPwj · 2025-06-23

**Clarity:** 3
**Significance:** 3
**Originality:** 3
**Rating:** 5
**Confidence:** 4

**Summary:**

This paper introduces a novel mechanism for generating unlimited synthetic time series data paired with their underlying symbolic (mathematical) expressions—based on complex dynamical systems theory—to overcome real-world data scarcity and imbalance. Leveraging this dual-modality dataset, the authors pretrain SymTime, a foundation model that fuses masked time-series and symbolic-language modeling with contrastive learning, achieving state-of-the-art performance across five major time-series analysis tasks.

**Questions:**

- Would the synthetic data generation process be able to reproduce phenomena observed in real-world domain specific time-series datasets, e.g. volatility clustering in finance?

**Ethical Concerns:**

["NO or VERY MINOR ethics concerns only"]

**Final Justification:**

The authors addressed my concerns and added new experiments. My initial review was positive and I raise my confidence now.

**Limitations:**

Yes

**Quality:**

4

**Strengths And Weaknesses:**

**Strengths**
- I very much like the careful design of the S² generator and the theoretical background provided around Takens’ theorem.
- The idea of synthetically generating time-series seems with their proposed approach seems to be highly effective.
- I like the broad experimentation carried out by the authors, and the consistent gains that their method seems to achieve.
- The ablations performed by the authors support the strength of each of the changes introduced.

**Weaknesses**

- If I understood correctly, there is no comparison against real-data pre-training with the same architecture. This would have been a valuable addition to the paper.
- There are no standard deviation results in the tables presented.

---

> ### Author Rebuttal · Authors · 2025-07-31
>
> We sincerely thank you for taking the time to review our paper and your constructive comments. We appreciate your recognition of our mechanism. Although deep learning technology has made great progress in time series analysis, we still hope to return to the essence of time series and introduce the complex dynamical system modeling perspective in machine learning to solve the current problem like data imbalance.
>
> ---
>
> **We will respond to your questions one by one below:**
>
> > **W1**: If I understood correctly, there is no comparison against real-data pre-training with the same architecture. This would have been a valuable addition to the paper.
>
> Thank you for the question. The paper's core is the Taken-theorem-based $S^2$ generator, which produces unlimited time-series and symbolic data. SymTime is designed to learn both modalities jointly, leveraging symbolic representations to boost time series modeling—this benefit is verified in our ablation. However, real-world data lack symbolic expressions, so we can not pre-train SymTime with only time series data through **Equation (6)**.
>
> But in order to conduct fair and effective comparative experiments, we adopt the configuration of the ablation experiment in **Section 4.4**, randomly and uniformly selecting $1/6$ (we use 50B time series points data in $S^2$ for pre-training) of real time series points from the **Time300B** dataset (Time-MoE), pre-training the model using **Equation (2)** as the target (only using mask time series modeling, not considering symbolic expressions), and fine-tuning on the ETTh1 and ETTh2 datasets. The experimental results (mean (std)) are shown below:
>
> | Type   | SymTime      |              | Real-Data    |              | w/o Symbol   |              |
> | ------ | ------------ | ------------ | ------------ | ------------ | ------------ | ------------ |
> | Metric | MSE          | MAE          | MSE          | MAE          | MSE          | MAE          |
> | ETTh1  | 0.442(0.008) | 0.448(0.006) | 0.446(0.012) | 0.453(0.007) | 0.450(0.0010) | 0.454(0.006) |
> | ETTh2  | 0.382(0.007) | 0.425(0.005) | 0.389(0.009) | 0.430(0.005) | 0.401(0.009) | 0.432(0.006) |
>
> We conduct multiple experiments by switching random seeds, with the results for each experiment being the average of forecasting lengths {96, 192, 336, 720}. The results show that pre-trained SymTime on real data outperforms models pre-trained solely on time series data from the $S^2$ dataset. However, it underperforms models pre-trained using both series and symbolic data. This result further demonstrates the relationship between symbolic expression data and time series, and that this pairing effectively enhances SymTime's performance. We did not intentionally exclude samples from the ETT dataset when sampling from Time300B, so our masked time series modeling may include our test samples. However, the performance of SymTime pre-trained on the synthetic $S^2$ dataset is not inferior to that of real data.
>
> **We will add these experimental results to the ablation experiments in the main text (Section 4.4) and provide a detailed description of our experimental setup.**
>
> > **W2**: There are no standard deviation results in the tables presented.
>
> Following TimesNet's benchmark across five tasks, we omitted SDs for brevity but reported their ranges in each table header (see **Tables 17–26** in Appendix). For example, **Table 17** notes SD ≤ 0.5 % for long-term forecasting.
>
> **We will highlight this note more prominently in the main text.**
>
> > **Q**: Would the synthetic data generation process be able to reproduce phenomena observed in real-world domain specific time-series datasets, e.g. volatility clustering in finance?
>
> Thank you for the insightful question and examples. The diversity of our $S^2$ generator stems from (1) the unrestricted variety of symbolic systems $f(\cdot)$ and (2) the diversity of sampled series $X$. Hence, $S^2$ can in principle generate any real-world domain. **Section 4.1** empirically confirms this.
>
> Financial series, as highly chaotic systems, exhibit strong heteroskedasticity and volatility clustering, typically modeled by ARCH or GARCH. Searching $S^2$, we find an $f(\cdot)$ that reproduces this behavior:
>
> $$
> y = x \cdot \exp\!\left(\frac{\sin^{2}(x-1)}{2}\right),
> $$
>
> with $x \sim \mathcal N(0,1)$ or any stationary input (in $S^2$ we can generate this input sampling series). The resulting $y$ shows clear volatility clustering akin to ARCH(1)/GARCH(1,1): large shocks are followed by calm periods.
>
> The effect can be observed with the short **Python** code below.
>
> ~~~python
> import numpy as np
> from matplotlib import pyplot as plt
>
> np.random.seed(42)
> x = np.random.normal(0, 1, 251)
>
> fig, ax = plt.subplots(2, 1, figsize=(12, 8))
> y = x * np.exp((np.sin((x - 1) ** 2)) / 2)
>
> ax[0].plot(y, color="royalblue")
> ax[0].set_title("Financial Time Series with Volatility Clustering")
> ax[1].plot(y[:-1] ** 2, "tomato")
> ax[1].set_title("Conditional Variance (Shows Clustering)")
> ~~~
>
> By observing this visualization, we can clearly see the small fluctuations after the large fluctuations in the time domain. This is well reflected in the real-time changes in the variance of the series below, where large and small variances appear alternately. Furthermore, real financial time series may contain trend information, and the complex system generated by our $S^2$ dataset is more complex than the above formula. It may contain expected trend information (derived from the autoregressive model in our sampled series). See the visualization in **Section B.2** for specific possible representations.
>
> The series is particularly similar to the simulated ARCH(1) model:
> $$
> r_t = \sigma_{t|t-1} \varepsilon_t,
> $$
> $$
> \sigma_{t|t-1} = \omega + \alpha r^2_{t-1},
> $$
> when $\omega=0.01$ and $\alpha=0.9$.
>
> To prevent explosion, we omitted autoregressive operators such as $y_t = \phi_1y_{t-1} + \phi_2y_{t - 2}$. (In order to prevent excessively large values from affecting the standardization during pre-training, we mentioned in **line 132** that excessively large sample values will be discarded.) But your example shows volatility clustering depends on past values, so we will reintroduce autoregression under tight constraints to yield richer series.
>
> **We believe this is a compelling and valuable case and add it to our paper to further demonstrate the validity of $S^2$.**

---

> > ### Comment · Reviewer_SPwj · 2025-08-05
> >
> > Thank you for the additional experiments and explanations provided by the authors. I will take them into account in my final score.

---

> > > ### Author Response · Authors · 2025-08-05
> > > **Thank you for your recognition and reply**
> > >
> > > **Dear Reviewer**,
> > >
> > > Thank you very much for your positive recognition of our work. We are sincerely grateful for the valuable time and effort you have dedicated to reviewing our paper and considering our rebuttal materials. Your constructive feedback throughout the review process has been instrumental, and we appreciate your acknowledgment of the additional experiments and explanations provided.
> > >
> > > We are delighted that you found merit in our contributions. Rest assured, we will carefully consider all points discussed during this review cycle as we work to further refine and improve the paper. We hope that your recognition of our perspectives on complex system modeling will further advance the field of time series analysis.
> > >
> > > Thank you again for your support and insightful comments.
> > >
> > > **Sincerely,
> > > The Authors**

---

### Official Review · Reviewer_1qct · 2025-07-01

**Clarity:** 3
**Significance:** 3
**Originality:** 3
**Rating:** 4
**Confidence:** 3

**Summary:**

The article focuses on improving model performance through data augmentation and the use of foundation models. Specifically, they propose a series-symbol data generation mechanism that alleviates issues of scarcity and imbalance in data. Additionally, they design a time series foundation model consisting of two encoders, where one encoder processes time series data and the other encoder processes symbol inputs. These two encoders are trained using contrastive learning to enhance the representation ability of time series. The approach has achieved good results in multiple time series tasks.
Experiments verify the ability of data augmentation in terms of stability, predictability, frequency domain analysis, and trendiness. The comparative experiments validate the performance of the foundation model across multiple downstream tasks.

**Questions:**

1. Does the symbol data generation mechanism have a complexity issue when dealing with long or non-stationary time series data?
2. Is the symbol generation mechanism required to be adjusted according to downstream tasks? Please provide detailed information on how symbols are selected.
3. How were the 6-layer Transformer encoder for handling time series data and the 6-layer DistilBERT for handling symbol data chosen? Does layer depth have any impact on performance?

**Ethical Concerns:**

["NO or VERY MINOR ethics concerns only"]

**Final Justification:**

These supplementary materials and the responses from the authors to me and other reviewers basically answered my questions. I think it is sufficient to rating this work, and I will maintain my scores.

**Limitations:**

Yes

**Quality:**

3

**Strengths And Weaknesses:**

S1: The author proposes a data generation mechanism that addresses issues of data scarcity and imbalance, and experimental results show that it can alleviate these problems.
S2: The pre-trained model demonstrates strong capabilities in multiple time-series tasks.

W1: Lack of implementation details for the symbol data generation mechanism.
W2: No explanation was given regarding the complexity issue of the symbol data generation mechanism when dealing with long series, as well as whether the symbol selection strategy needs to be adapted based on specific tasks.

---

> ### Author Rebuttal · Authors · 2025-07-31
>
> Thank you for your recognition of the data generation mechanism, model performance, and experimental results in this paper. Below I will answer your questions one by one.
>
> > **W1:** Lack of implementation details for the symbol data generation mechanism.
>
> Thank you for your valuable feedback. We will open a new chapter in the appendix to further explain each step and hyperparameter settings of the $S^2$ data generation mechanism in detail, and we have responded to your questions in detail in Q1-Q3.
>
> **We will open source our $S^2$ data generation code in the future.**
>
> > **W2:** No explanation was given regarding the complexity issue of the symbol data generation mechanism when dealing with long series.
>
> We define the specific symbol explanation and its  as follows. Then, we use the divide-and-conquer approach to analyze the complexity of our $S^2$ data generation mechanism.
>
> | symbol | explanation                                                  |
> | ------ | ------------------------------------------------------------ |
> | $L$    | The length of time series                                    |
> | $M$    | Number of input channels                                     |
> | $N$    | Number of output channels                                    |
> | $k$    | Total number of mixed distributions used                     |
> | $p$    | Autoregressive model order in ARMA model                     |
> | $q$    | Moving average model order in ARMA model                     |
> | $P$    | Probability of choosing a sampling method                    |
> | $b$    | Number of binary operators used to construct symbolic expressions |
> | $u$    | The number of unary operators used to construct symbolic expressions |
>
> 1. **Symbolic Expression Generation**: We construct symbolic expressions using a tree structure as a medium. When we have $b$ binary operators, we further insert $(b + 1)$ leaf nodes (the process from (a) to (b) in **Figure 3** in our paper). Therefore, after inserting $u$ unary operators (**Figure 3** (c)), the total number of nodes in the tree is $n = 2b + u + 1$. Because there are many ways to construct a tree, we consider the time complexity of constructing a balanced tree. Therefore, for $N$ symbols constructed, the specific complexity of this process is $O(N\times n\mathrm{log}n)$.
>
> 2. **Sampling series generation**: When we want to generate a sampling time series with $M$ channels, each channel has a probability of $P$ to be sampled using a mixture distribution and a probability of $(1-P)$ to be sampled using an ARMA model. When the sampling length of the series is $L$, the complexity of generating $k$ mixture distribution and ARMA ($p$, $q$) series is $O(kL)$ and $O(L(p+q))$. Therefore, the time complexity of this process can be quantified as $O \left ( ML \times [Pk + (1-P)(p+q)] \right )$.
>
> 3. **Sampling through symbolic expressions and series**: We simplify the specific operational details of this process and only consider the time complexity of operations with variables. For a series of length L, we have $N$ symbolic expressions to be sampled, and each symbol has an average of $\frac{M+1}{2}$ variables (Each symbolic expression may contain any number of variables from 1 to M, so here we take $\frac{M+1}{2}=\frac{(1+2+\cdots+M)}{M}$ as the average probability). Then the process can be quantified as $O(N \cdot \frac{M+1}{2}\cdot L)$.
>
> To sum up, since other variables that affect the $S^2$ sampling process are usually small, it can be intuitively understood that the time complexity of the entire sampling process is proportional to the length $L$.
>
> In **Q1** below, we further answer why this paper does not sample long time series.
>
> > **W2:** whether the symbol selection strategy needs to be adapted based on specific tasks.
>
> **We do not select any symbols**. Instead, to learn the richest representation of time series, we use all symbols to generate the $S^2$ dataset for unified pre-training of SymTime. See **Q2** for details.
>
> > **Q1:** Does the symbol data generation mechanism have a complexity issue when dealing with long or non-stationary time series data?
>
> Through **W2**, we find that the time complexity of data generation is indeed related to the length $L$. The specific explanation for avoiding sampling long and non-stationary series is as follows:
>
> 1. **Out of domain**: As described in **line 132**, the symbolic expression $f(\cdot)$ has a domain restriction. The longer the series, the more likely it is that the value will fall outside the domain, and the probability of sampling failure increases. When we fail to sample, we will resample.
> 2. **Stationarity**: Non-stationary series often exhibit strong autoregressive properties, which can easily lead to numerical inflation and sampling failure (**line 132**). Furthermore, stationary series have bounded prediction errors and are better characterized; non-stationary series have errors that tend to diverge and are poorly characterized. Therefore, to ensure sampling success and forecast performance, we prioritize generating stationary series.
> 3. **More channels**: Since we use mask time series modeling for representation learning, the benefits of long series on our model learning are not significant. We are more likely to generate data with more channels (with $M=6, N=12$) to help the model learn the feature between multivariate time series channels.
>
> In summary, in order to more effectively generate data with better predictability, we adopted the strategy described in the paper in the $S^2$ data generation mechanism.
>
> > **Q2:** Is the symbol generation mechanism required to be adjusted according to downstream tasks? Please provide detailed information on how symbols are selected.
>
> As described in **Experiment 4.1**, our goal is to generate as much data as possible through an unrestricted data generation mechanism to cover the broadest possible representation of time series, thereby training a time series foundation model (**Definition 1 in Line 54**). Using this dataset, we uniformly pre-train SymTime, enabling the model to learn the most comprehensive representation of time series data. To this end, we do not specifically tailor the symbolic expressions we use to specific downstream tasks or datasets. This approach demonstrates the generalization capabilities of SymTime as a foundation model.
>
> > **Q3-1:** How were the 6-layer Transformer encoder for handling time series data and the 6-layer DistilBERT for handling symbol data chosen?
>
> 1. **Time Series Encoder**: We drew on the frameworks of time series foundation models such as **Moirai**, **Timer**, and **Moment**, as well as model architectures in the computer vision such as **ViT** and **ALBEF**. Ultimately, we chose to use a 6-layer Transformer encoder to learn the basic representation of time series data.
> 2. **Symbolic Encoder**: This module aims to extract the representation of symbolic expressions during contrastive learning and enable the time series encoder to learn the pairing relationship between series  and symbols. Considering computational complexity, we decide to chose a smaller, but more linguistically robust, pre-trained LLM as the symbolic encoder. Therefore, the two most commonly used 6-layer Transformer models are **gpt2-small** and **DistilBERT**. DistilBERT was chosen due to its superior performance and the compatibility of its encoder architecture with the time series encoder.
>
> > **Q3-2:** Does layer depth have any impact on performance?
>
> Based on the ideas in **Q3-1**, we chose to determine the model architecture of SymTime. To further verify the impact of the number of model Transformer layers on downstream task performance, we set different parameters for the time series encoder and constructed the following two control groups:
>
> | name    | layers | $d_{\mathrm{model}}$ | $d_{\mathrm{ff}}$ |
> | :-----: | :----: | :------------------: | :---------------: |
> | SymTime | 6 | 512 | 2048 |
> | A | 6 | 768 | 3072 |
> | B | 3 | 386 | 1536 |
>
> We keep the symbol encoder unchanged and pre-train the control group using the pre-training configuration used in **C.3** of the original paper (since the parameters of the model in A are further expanded, we reduce the batch size from 128 to 96, while keeping the other configurations unchanged). Then, we fine-tune the model on the four ETT datasets. The average forecasting results for the lengths {96, 192, 336, 720} are shown below:
>
> | Dataset | SymTime |       | A     |       | B     |       |
> | ------- | ------- | ----- | ----- | ----- | ----- | ----- |
> | Metrics | MSE     | MAE   | MSE   | MAE   | MSE   | MAE   |
> | ETTm1   | 0.371   | 0.390 | 0.370 | 0.398 | 0.378 | 0.395 |
> | ETTm2   | 0.274   | 0.321 | 0.278 | 0.323 | 0.279 | 0.324 |
> | ETTh1   | 0.430   | 0.436 | 0.435 | 0.439 | 0.434 | 0.439 |
> | ETTh2   | 0.365   | 0.402 | 0.369 | 0.405 | 0.375 | 0.410 |
> | Average | 0.360   | 0.387 | 0.363 | 0.391 | 0.367 | 0.392 |
>
> It is clear from this that SymTime's original model parameter configuration achieves optimal results on the ETT dataset.
>
> However, when the number of parameters changes, the forecasting results will decrease to a certain extent, but the decrease is not significant. Therefore, as long as the model can meet the representation learning requirements during the pre-training phase, the performance improvement of the model mainly comes from our pre-training. A model that is too large will, to a certain extent, bring greater computational complexity.

---

> > ### Comment · Area_Chair_yyCR · 2025-08-05
> > **Any additional questions for the authors?**
> >
> > Dear reviewer,
> >
> > The authors have provided a response. Do you have any additional questions for them? If so, please ask them now. If not, please summarize your conclusions after reading their response.
> >
> > Best,
> >
> > AC

---

> ### Comment · Reviewer_1qct · 2025-08-07
>
> Thank you for providing detailed symbol explanation and additional experiments. These supplementary materials and your responses to other reviewers basically answered my questions. I think it is sufficient to rating your work, and I will maintain my scores.

---

> > ### Author Response · Authors · 2025-08-07
> >
> > **Dear Reviewer,**
> >
> > Thank you for acknowledging our supplementary materials and experimental additions. We sincerely appreciate your confirmation that the symbol explanations and extended results addressed your concerns, and we are grateful for your commitment to maintaining the evaluation score.
> >
> > Your focus on methodological clarity and empirical rigor has significantly strengthened our paper. We will further add the time complexity analysis of the S2 data generation algorithm to our paper.
> >
> > **Respectfully,**
> >
> > **The Authors**

---

### Official Review · Reviewer_yCnr · 2025-07-02

**Clarity:** 3
**Significance:** 3
**Originality:** 2
**Rating:** 5
**Confidence:** 4

**Summary:**

The paper introduces a novel approach to address data scarcity and imbalance in time series analysis. The authors propose a series-symbol data generation pipeline. Inspired by complex dynamic system theories, the authors generate high-quality univariate time series data paired with corresponding symbolic expressions, enabling the creation of diverse time series datasets. The authors present SymTime, a pre-trained encoder-only time series foundation model, leveraging contrastive learning and masked modeling. It shows notable scalability and generalization performance.

**Questions:**

1. Under the proposed symbolic system, what is the maximum scale of synthetic data that can be constructed?
2. Are there any suggestions to overcome the diminishing marginal returns when pre-training on an ever-larger scale of synthetic data?
3. What is the range of different statistical metrics in Figure 4? The generated data has relatively low diversity in some metrics (PE, FFT Mean, Seasonality). How does pre-training based on metrics of different degrees of diversity affect the performance of downstream tasks?
4. Can the proposed method generate multivariate time series? It may also provide insights about how to define how "multivariate" a set of time series is.

**Ethical Concerns:**

["NO or VERY MINOR ethics concerns only"]

**Final Justification:**

Based on the additional empirical results provided, I am confident that the framework has the potential to offer valuable insights for the development of time series foundation models. Therefore, I will maintain my positive evaluation. Additionally, I am pleased to see the release of this framework.

**Limitations:**

Please discuss whether omitted operators (e.g., exponent and higher power) limit the generation of certain dynamics.

**Quality:**

3

**Strengths And Weaknesses:**

**Quality**: The submission is technically sound, with a strong theoretical foundation and extensive experimental validation. SymTime’s architecture is well-motivated, leveraging contrastive learning and masked modeling on numerical and symbolic tokens. About this architecture option, I wonder if the author has ever tried generative modeling on a decoder-only Transformer like TimesFM and Timer. The benchmarks include five major TSA tasks (forecasting, classification, imputation, anomaly detection). Although it is comprehensive, I'd like to raise my score if the model can be evaluated more public leaderboard, such as GIFT-Eval and FEV.

**Clarity**: The mathematical formulation (e.g., tree-based symbolic sampling, ARMA processes) is well-explained. The paper is generally well-written and organized, but there are some typos that could be improved (e.g., "prmutation" -> "permutation"; "pretrain" -> "pre-train").

**Significance**: The paper provides new insights for (1) pre-training generalizable time series foundation models, (2) inspiring new benchmarks for time-series pre-training. As symbolically generated time-series data can effectively replace or augment real-world data for pre-training, it can be beneficial for the community if the author releases the data generation framework.

**Originality**: Different from previous works, which are pre-trained solely on synthetic data or partially synthetic data. The proposed symbolic framework can provide aligned synthetic data and symbolic formulation. In addition to enabling contrastive learning, it is a promising direction is to take symbolic formulation as the aligned textual description for multimodal time series models, which can help current models better understand the underlying dynamics of time series. In terms of the model design, the contributions of SymTime mainly focus on pre-training tasks with contrastive learning. The structure is similar to previous works (such as moment, moirai).

---

> ### Author Rebuttal · Authors · 2025-07-31
>
> We sincerely thank you for your recognition of the novelty of the data generation mechanism and the effectiveness of the SymTime model. Below we will answer your valuable questions one by one.
>
> > **Quality**: Although it is comprehensive, I'd like to raise my score if the model can be evaluated more public leaderboard, such as GIFT-Eval and FEV.
>
> Thank you for your recognition of the comprehensiveness of our experiments. We are happy to verify SymTime on the **GIFT-Eval benchmark**. We will split the training sets according to the requirements and fine-tune SymTime.
>
> However, due to the lots of sub-datasets included in this benchmark and time constraints, we have not yet completed fine-tuning. Once we have completed it, we will immediately inform you of the results and upload the online leaderboard. The results will also be added to **Section A.1** in our paper.
>
> > **Clarity**: The paper is generally well-written and organized, but there are some typos that could be improved.
>
> Thank you for your recognition of the clarity of our algorithm writing. We already have corrected the above errors and further checked the full paper.
>
> > **Significance**: As symbolically generated time-series data can effectively replace or augment real-world data for pre-training, it can be beneficial for the community if the author releases the data generation framework.
>
> We sincerely appreciate your recognition of our $S^2$ data generation mechanism. Deep learning have made significant progress in time series analysis, but we continue to strive to address existing challenges by incorporating the insights of complex dynamical systems into the essence of time series.
>
> **We promise to open source the code for our $S^2$ data generation mechanism in the future.**
>
> > **Q1:** Under the proposed symbolic system, what is the maximum scale of synthetic data that can be constructed?
>
> Theoretically, the $S^2$ data generation mechanism can generate unlimited sequence-symbol pairings. This diversity is primarily reflected in two aspects:
>
> 1. **It can generate an unrestricted variety of symbolic expressions $f(\cdot)$**: Due to the existence of random constants and a variety of symbols and operators, the symbols we can generate are infinite, and the time series generated by different symbols are also diverse.
> 2. **It can generate a variety of sampled sequences $X$**: Since we switch between different random seeds and specify different initializations at each sampling, $X$ generated from a mixture of uniform and normal distributions and randomly parameterized ARMA(p, q) models is also diverse. This also makes the data we can generate infinitely.
>
> > **Q2**: Are there any suggestions to overcome the diminishing marginal returns when pre-training on an ever-larger scale of synthetic data?
>
> This is a very valuable question for large-scale pre-training. Currently, many large-scale foundation models trained on real datasets face the problem of diminishing marginal returns. However, in the experiments in **Section 4.3** that verified the impact of pre-training dataset size on SymTime performance, we find that as we continuously increase the pre-training dataset, the model's performance continue to improve randomly. This phenomenon has little impact on SymTime. Therefore, we conduct a detailed analysis, and the specific reasons are as follows:
>
> When the $S^2$ dataset is generated, its representation is more "random," meaning that most adjacent samples may not come from the same distribution. As our data volume increases, our $S^2$ data will appear evenly across the representation space of time series data. However, real time series datasets often consist of multiple sub-datasets, each of which comes from the same distribution and has the same generation mechanism. Therefore, as the amount of real data increases, it may only be able to fill a certain position in the representation space (a specific position and range within this distribution).
>
> Therefore, for other synthetic datasets, <u>we believe that we should make the data more random when generating it</u>. Rather than restricting the possible distribution of the data, we should make it more diverse. This approach we believe can alleviate this problem.
>
> > **Q3-1**: What is the range of different statistical metrics in Figure 4? The generated data has relatively low diversity in some metrics (PE, FFT Mean, Seasonality).
>
> This is a very interesting question. Radviz visualizes high-D data in 2D using a **spring force balance**. After normalization, sample features act as "pulls" from anchor points evenly spaced around a circle. The sample's final position is determined by the equilibrium of all these pulls. Consequently, the 2D coordinates themselves lack inherent physical meaning, as they result from the combined influence of all features. This method is useful for observing clustering in reduced dimensions. The observed lack of diversity in indicators like PE, FFT Mean and Seasonality within the Radviz plot is an artifact of this balancing process across all indicators, not an indication that those specific features lack diversity in the actual data.
>
> > **Q3-2**: How does pre-training based on metrics of different degrees of diversity affect the performance of downstream tasks?
>
> To quantitatively explore this issue, we selecte **25B** of time series data from the 50B $S^2$ dataset, which ranked high for Seasonality (S), Forecastability (F), and Permutation Entropy (PE) respectively. So the subdataset can reflect the seasonality, predictability, and permutation complexity of the data, respectively. We pre-train SymTime using the configuration in C.3 and fine-tuned it on four ETT datasets for downstream tasks. The experimental results are shown below:
>
> |Dataset|F| |S| | PE | | SymTime | |
> | ------- | :---: | :---: | :---: | :---: | :---: | :---: | :-----: | :---: |
> |Metrics|MSE|MAE|MSE|MAE|MSE|MAE|MSE|MAE|
> |ETTm1|0.374|0.393|0.380|0.395|0.385|0.389|0.378|0.393|
> |ETTm2|0.283|0.329|0.278|0.325|0.290|0.331|0.278|0.325|
> |ETTh1|0.433|0.438|0.445|0.443|0.443|0.440|0.434|0.438|
> |ETTh2|0.386|0.411|0.382|0.410|0.408|0.419|0.370|0.408|
> |avg|0.369|0.393|0.371|0.395|0.382|0.395|0.365|0.391|
>
> We believe this is a very interesting experiment. The results show that metrics with varying degrees of diversity do have a certain impact on the performance of downstream tasks. We can preliminarily conclude that Forecasting and Forecasting efficiency metrics are more effective in promoting model representation learning than PE.
>
> > **Q4**: Can the proposed method generate multivariate time series? It may also provide insights about how to define how "multivariate" a set of time series is.
>
> Our $S^2$ data generation mechanism can generate a multivariate time series $Y = f(X)\in \mathbb R^{N\times L}$ by constructing a multivariate complex system $f(\cdot)$ and a multivariate sampling series $X\in \mathbb R^{M\times L}$, where $M$ and $N$ are the input and output channels respectively and $L$ is the length of time series.
>
> For example, we can generate a multivariate symbol and series as follows:
> $$
> f(x_1, x_2, x_3) = \begin{bmatrix}f_1(x_1, x_2, x_3)
>  \\ f_2(x_1, x_2, x_3)
>  \\ f_3(x_1, x_2, x_3)
> \end{bmatrix}
> $$
> $$
> X = \begin{bmatrix} x_1
>  \\ x_2
>  \\ x_3
>  \end{bmatrix}
> $$
> We simultaneously obtain time series data for the three input and three output channels using $Y=f(X)$. This allows us to combine sampling series from different channels through symbolic operations. Therefore, our data generation mechanism is capable of generating multivariate time series with channel correlation. Specifically, we present the generated symbolic expressions and time series data in **Section B.1 on page 21**, including multivariate time series data, from which we can observe the correlation between channels.
>
> > **Limitations**: Please discuss whether omitted operators (e.g., exponent and higher power) limit the generation of certain dynamics.
>
> Thank you for this valuable comment. In the current $S^2$, omitting certain operators does restrict the creation of some dynamic data. Earlier versions included a richer set of numerical operations—differentiation, integration, exponential functions with various bases, and higher-order powers. Extensive experiments showed that several of these operators severely impeded the data-generation algorithm; the main reasons are detailed below.
>
> 1. **Value explosion**: To maintain quality, we cap large magnitudes (l. 132). Integration, exponential and high-order powers readily cause overflow, so they were dropped; exp alone is retained for diversity.
> 2. **Numerical differentiation**: Symbolic differentiation introduces truncation/round-off trade-offs. Combined with reciprocal and absolute-value operators, functions such as $|x|$ or $\mathrm{sin}(\frac{1}{x})$ because non-smooth or high-frequency vibration at $x=0$, breaking differentiation.
> 3. **Numerical integration**: Randomly built expression trees often yield integrands with singularities (e.g., $\int_{0}^{1}\frac{1}{\sqrt{x}}\mathrm{d}x $) or force costly oscillatory integrals (e.g., $\int_{0}^{100}\mathrm{sin}(100x)\mathrm{d}x$); interval selection is non-trivial.
> 4. **Symbolic cost**: Differentiation and integration are slow and can trigger exponential memory growth. Many elementary functions lack closed-form antiderivatives (e.g., $\int \mathrm e ^{-x^2}\mathrm d x$).
>
> Considering factors like numerical stability, symbolic complexity, computational efficiency, and sampling success rate, we selectively omitted some symbolic operations.
>
> Nevertheless, the $S^2$ data generation framework proposed is essentially a complete theory. It already incorporates the vast majority of symbolic operations, and new operators or user-defined symbolic operations can be easily added. The omission of some operations due to the above factors does not undermine the validity of this framework. We consider this a valuable point and will add it in the paper.

---

> > ### Comment · Reviewer_yCnr · 2025-08-05
> >
> > Thanks to the authors' responses, which have effectively addressed the majority of my concerns. Based on the additional empirical results provided, I am confident that the framework has the potential to offer valuable insights for the development of time series foundation models. Therefore, I will maintain my positive evaluation.

---

> > > ### Author Response · Authors · 2025-08-06
> > >
> > > **Dear Reviewer,**
> > >
> > > Thank you for your **constructive feedback and continued support** throughout the review process. We deeply appreciate your recognition of our framework’s potential to contribute to **time-series foundation models** and your commitment to maintaining a positive assessment. Your insights have been invaluable in strengthening both our paper and our perspective on the broader implications of this work.
> > >
> > > We are sincerely grateful for the time and expertise you dedicated to evaluating our revisions.
> > >
> > > **Sincerely,**
> > >
> > > **The Authors**

---

### Official Review · Reviewer_Atv9 · 2025-07-02

**Clarity:** 2
**Significance:** 3
**Originality:** 3
**Rating:** 5
**Confidence:** 3

**Summary:**

This paper introduces a synthetic data generation method for training time series foundation models. At the core of the synthetic data generation method is a composition of symbolic expressions that gets randomly generated through sampling (sampling input/output dimensions, operators, affine transformations, etc.). The input to these symbolic expressions are time series sampled from mixed distributions and ARMA processes. The transformation of the symbolic expression then forms the synthetic time series. The paper also introduces SymTime, a time series foundation model which uses patched encodings, cross-entropy loss, and additional contrastive learning and momentum distillation terms.

The manuscript compares the generated time series dataset features with existing datasets from Monash to evaluate whether the generated time series are realistic. The authors then evaluate their models on forecasting, imputation, classification, and anomaly detection tasks. Finally, the paper conducts ablations on the pre-training objectives and analyses the obtained time series and symbol representations.

**Questions:**

How are positive/negative pairs selected in the contrastive learning step?

**Ethical Concerns:**

["NO or VERY MINOR ethics concerns only"]

**Final Justification:**

The authors addressed a major weakness in their evaluation of long-term forecasting benchmark, along with other clarifications and improvements as evident by the discussions with the other reviewers. Hence, I raise my score.

**Limitations:**

The authors appropriately address limitations.

**Quality:**

3

**Strengths And Weaknesses:**

The paper introduces an interesting idea for synthetic time series generation based on sampling of symbolic expressions. This offers a flexible way of generating synthetic time series. As the authors state in the paper, pretraining time series models requires synthetic data due to a relative scarcity (compared the text data) and therefore novel ideas for synthetic data generation are needed. Therefore, the explored topic in the paper has significance and is original. Another strength of the paper is that the thorough evaluation of the model on several downstream tasks (forecasting, imputation, classification, anomaly detection) along with ablations and analysis of the generated synthetic time series and SymTime representations.

One weakness of the paper is the clarity of writing. I found several parts of the paper a bit confusing and even after reading several times I am still not sure how they work. While I understood the data generation process after some re-reads, I still do not understand the contrastive learning part of SymTime. For example, how are positive and negative pairs sampled? I understand that the authors covered a lot of ground and had to accommodate this in the space constraints, but I believe a clearer explanation of the core concepts would strengthen the paper.

Another major weakness is the specific setup of the forecasting results and the claims that are made on SymTime’s performance in forecasting. The paper for the long-range forecasting dataset, a fixed lookback window of 96 is used for most comparing models. This makes the conclusion that SymTime outperforms the other baseline models on this dataset invalid because it only does so under this (artificial) conditions that make the baseline methods perform worse. For example, the MSE/MAE windows for PatchTST are much lower in its original paper on these datasets, if the proposed lookback window of 336 or 512 time steps is used. Thus, SymTime does not reduce the absolute error on these datasets compared to existing methods. I have a similar criticism on the short-term datasets. Here again, while the results for SymTime are good, the OWA for the state-of-the-art method is also lower (see Gasthaus et al. (2019), "Spline Quantile Function RNNs") than the error of the methods presented here.

Additionally, while the long-term forecasting benchmark has been used extensively in time series forecasting, the value of this benchmark to still measure progress in this field is questionable. This has been explored in recent position papers (https://arxiv.org/abs/2502.14045) and in the NeurIPS 2024 time series workshop (https://neurips.cc/virtual/2024/workshop/84712#wse-detail-108471, see Christoph Bergmeir's talk). I think it is important for the time series forecasting filed to move towards more extensive evaluation which is proposed in several recent paper (Gifteval: https://arxiv.org/abs/2410.10393; FEV (from the Chronos paper): https://openreview.net/forum?id=gerNCVqqtR).

I understand and appreciate that this work also considers other tasks (like classification and anomaly detection, where I have less insight on the SOTA). Therefore, the paper provides additional experimental evidence. However, I would ask the authors to at least run some of the baseline methods with their recommended settings (specifically PatchTST with 336/512 context length) make sure their comparison captures the error that SOTA models are capable of and therefore an accurate reflection of the SOTA.

---

> ### Author Rebuttal · Authors · 2025-07-31
>
> We sincerely appreciate your valuable feedback and your acknowledgement of the thoroughness of our experiments on downstream tasks. We will address your questions one by one.
>
> > **W1**: For example, how are positive and negative pairs sampled?
>
> Thank you for your question. In our $S^2$ data generation mechanism, we generate a three-tuple pair of data: (sample sequence, generated sequence, symbolic expression). Specifically, we first construct the symbolic expression $f(\cdot)$. Then, we construct the sample series $X$. Finally, by inputting the sample series into the symbolic expression, we obtain the generated series $Y=f(X)$. In this process, we believe that the generated $Y$ will have a natural correspondence with the symbol $f(\cdot)$ that generates it.
>
> Therefore, for a certain time series $Y$, we believe that the series and the symbolic expression $f(\cdot)$ that generates it are mutually positive samples, and that other symbolic expressions are mutually negative samples. Correspondingly, for a certain symbol $f(\cdot)$, the time series $Y$ generated by it is mutually positive, and other irrelevant symbols are mutually negative samples.
>
> In the abstract and **line 153**, we repeatedly mention that we generate paired data of sequences and symbols, and the pairing of the two makes them mutually positive samples. **We will make this clear in the main paper**.
>
> > **W2**: The paper for the long-range forecasting dataset, a fixed lookback window of 96 is used for most comparing models. This makes the conclusion that SymTime outperforms the other baseline models on this dataset invalid because it only does so under this (artificial) conditions that make the baseline methods perform worse.
> >
> > However, I would ask the authors to at least run some of the baseline methods with their recommended settings (specifically PatchTST with 336/512 context length) make sure their comparison captures the error that SOTA models are capable of and therefore an accurate reflection of the SOTA.
>
> Thank you for your valuable feedback. As you mentioned, for long-term forecasting, the vast majority of models use a look-back window of 96. Therefore, to ensure fairness, we adjusted the look-back length of all models to 96 in our initial comparison.
>
> However, We are pleased to accept your suggestion. To this end, we adjust the look-back window to 336 and 512, re-run the experiments, and compared them with a series of current baseline models. Below we show the average results for four lengths {96, 192, 336, 720} on 8 datasets:
>
>
> | 336 | SymTime |        | PatchTST |        | TimeMixer |        | TimesNet |        | Autoformer |        | DLinear |        | iTransformer |        |
> | -------- | :-----: | :----: | :------: | :----: | :-------: | :----: | :------: | :----: | :--------: | :----: | :-----: | :----: | :----------: | :----: |
> | Metrics  | MSE     | MAE      | MSE       | MAE      | MSE        | MAE     | MSE          | MAE    | MSE    | MAE    | MSE    | MAE    | MSE    | MAE    |
> | ETTm1    | 0.354   | 0.381    | 0.352     | 0.382    | 0.368      | 0.392   | 0.421        | 0.423  | 0.618  | 0.539  | 0.357  | 0.379  | 0.368  | 0.395  |
> | ETTm2    | 0.260   | 0.317    | 0.258     | 0.315    | 0.262      | 0.318   | 0.282        | 0.334  | 0.400  | 0.420  | 0.291  | 0.353  | 0.272  | 0.329  |
> | ETTh1    | 0.425   | 0.436    | 0.419     | 0.432    | 0.430      | 0.437   | 0.485        | 0.480  | 0.580  | 0.539  | 0.425  | 0.440  | 0.450  | 0.457  |
> | ETTh2    | 0.352   | 0.393    | 0.331     | 0.379    | 0.396      | 0.425   | 0.409        | 0.440  | 0.663  | 0.604  | 0.490  | 0.476  | 0.390  | 0.416  |
> | weather  | 0.238   | 0.273    | 0.258     | 0.292    | 0.235      | 0.273   | 0.250        | 0.286  | 0.441  | 0.450  | 0.245  | 0.298  | 0.238  | 0.272  |
> | ECL      | 0.165   | 0.267    | 0.165     | 0.294    | 0.169      | 0.260   | 0.197        | 0.297  | 0.236  | 0.346  | 0.170  | 0.269  | 0.163  | 0.257  |
> | Traffic  | 0.393   | 0.266    | 0.396     | 0.268    | 0.411      | 0.271   | 0.615        | 0.331  | 0.676  | 0.413  | 0.465  | 0.320  | 0.385  | 0.273  |
> | Exchange | 0.370   | 0.408    | 0.385     | 0.420    | 0.415      | 0.438   | 0.548        | 0.532  | 1.053  | 0.807  | 0.448  | 0.462  | 0.392  | 0.427  |
> | Average  | **0.3198** | **0.3427** | 0.3204    | 0.3478   | 0.3357     | 0.3519  | 0.4009       | 0.3903 | 0.5832 | 0.5149 | 0.3614 | 0.3747 | 0.3323 | 0.3533 |
>
> | 512 | SymTime |        | PatchTST |        | TimeMixer |        | TimesNet |        | Autoformer |        | DLinear |        | iTransformer |        |
> | :------: | :--------: | :--------: | :-------: | :------: | :--------: | :-----: | :----------: | :----: | :----: | :----: | :----: | :----: | :----: | :----: |
> | Metrics  | MSE        | MAE        | MSE       | MAE      | MSE        | MAE     | MSE          | MAE    | MSE    | MAE    | MSE    | MAE    | MSE    | MAE    |
> | ETTm1    | 0.356      | 0.380      | 0.352     | 0.382    | 0.371      | 0.392   | 0.425        | 0.430  | 0.556  | 0.518  | 0.358  | 0.380  | 0.367  | 0.397  |
> | ETTm2    | 0.265      | 0.320      | 0.256     | 0.317    | 0.263      | 0.323   | 0.294        | 0.344  | 0.371  | 0.416  | 0.275  | 0.340  | 0.273  | 0.331  |
> | ETTh1    | 0.414      | 0.432      | 0.413     | 0.434    | 0.429      | 0.444   | 0.481        | 0.486  | 0.627  | 0.579  | 0.418  | 0.438  | 0.446  | 0.460  |
> | ETTh2    | 0.365      | 0.405      | 0.357     | 0.409    | 0.373      | 0.410   | 0.397        | 0.432  | 0.687  | 0.609  | 0.499  | 0.478  | 0.388  | 0.417  |
> | weather  | 0.234      | 0.273      | 0.245     | 0.284    | 0.231      | 0.271   | 0.251        | 0.288  | 0.489  | 0.486  | 0.241  | 0.292  | 0.249  | 0.280  |
> | ECL      | 0.163      | 0.267      | 0.169     | 0.269    | 0.177      | 0.274   | 0.201        | 0.302  | 0.353  | 0.393  | 0.167  | 0.267  | 0.162  | 0.257  |
> | Traffic  | 0.395      | 0.268      | 0.399     | 0.272    | 0.410      | 0.270   | 0.624        | 0.334  | 0.705  | 0.435  | 0.433  | 0.305  | 0.383  | 0.273  |
> | Exchange | 0.384      | 0.412      | 0.398     | 0.423    | 0.517      | 0.497   | 0.718        | 0.608  | 0.944  | 0.768  | 0.500  | 0.494  | 0.427  | 0.467  |
> | Average  | **0.3221** | **0.3446** | 0.3237    | 0.3487   | 0.3464     | 0.3601  | 0.4239       | 0.4030 | 0.5914 | 0.5253 | 0.3614 | 0.3744 | 0.3368 | 0.3603 |
>
> The results show that increasing the look-back does improve forecasting performance for most models, likely because longer series contain more historical information. SymTime, PatchTST, and TimeMixer show particularly significant improvements, especially for datasets with larger numbers of channels, such as Weather, ECL, and Traffic. We summarize the possible reasons for the SymTime performance improvement:
>
> 1. SymTime is Transformer encoder framework, and has also undergone large-scale pre-training.
> 2. The longer look-back window allows us to construct more tokens, which is closer to the average number of tokens in our pre-training (144), thus better leveraging our pre-training performance.
>
> The detailed test results and experimental configuration of this experiment will be added to the paper in **Section A.1** (long-term forecasting benchmark).
>
> > **W3**: Additionally, while the long-term forecasting benchmark has been used extensively in time series forecasting, the value of this benchmark to still measure progress in this field is questionable. This has been explored in recent position papers and in the NeurIPS 2024 time series workshop. I think it is important for the time series forecasting filed to move towards more extensive evaluation which is proposed in several recent paper.
>
> We appreciate your insights on time series forecasting benchmarks. Our paper uses the benchmark from TimesNet, which evaluates models across 5 different TSA tasks to assess multi-task generalization capabilities. This benchmark is widely adopted by models like GPT4TS and Peri-midFormer. The goal of SymTime is to demonstrate strong applicability in a variety of tasks.
>
> While, we recognize newer zero-shot forecasting benchmarks like **GIFT-Eval** and **FEV** (used by mary models such as Sundial and Chronos), these primarily target zero-shot forecasting scenarios which differ from SymTime's objectives. Therefore, we prioritized the TimesNet benchmark.
>
> To further validate SymTime, we follow GIFT-Eval's training splits and performed fine-tuning. Testing across all GIFT-Eval sub-datasets is ongoing due to time constraints. We commit to reporting full results upon completion, submitting them to the online leaderboard, and will include comparative analyses with other benchmarks in our paper's future work section.
>
> Another key aspect of this paper lies in the $S^2$ paradigm, which offers the advantage of unrestricted data generation. We can construct unlimited high-quality time series datasets for model pre-training. When the data representations we generate are similar to those in real-world test scenarios, we believe that models pre-trained with $S^2$ will also achieve excellent zero-shot performance on real-world data.
>
> > **Q**: How are positive/negative pairs selected in the contrastive learning step?
>
> In **W1**, we point out that the time series data generated by the same symbol $f(\cdot)$ will be positive samples with the symbol expression, and negative samples with other symbols.

---

> > ### Comment · Reviewer_Atv9 · 2025-08-05
> >
> > Thank you for response and your added experiments. The increased lookback window results do not indicate significant improvements over PatchTST in the long-term forecasting task. I think this is a more realistic reflection of a real-world scenario as there would be no reason to truncate the lookback window. Thus, I would advocate to update the main results in the paper (including the main table/figures) to reflect the findings from the increased lookback experiment.
> >
> > Overall, this paper has strong empirical results in other downstream tasks and long-term forecasting is only one component. Even if SymTime "only" matches a competing method like PatchTST, this would be enough for acceptance given the only paper.
> >
> > I will consider raising my score if the authors commit to update the main figures/tables in the main text to reflect these findings at longer lookback horizons, as opposed to only updating section A.1.

---

> > > ### Author Response · Authors · 2025-08-05
> > > **We promise to update the prediction results and conclusions for lengths of 336 and 512 to the main text**
> > >
> > > **Dear Reviewer,**
> > >
> > > Thank you for your thoughtful feedback and for recognizing the empirical strengths of our work. We sincerely appreciate your time and expertise in evaluating our rebuttal and providing constructive suggestions.
> > >
> > > We fully agree with your observation that the extended lookback window results offer a more realistic reflection of long-term forecasting performance. **We will update the main figures and tables in the main text to reflect these findings over the longer review period.** We promise to add the experimental configurations of 336 and 512 lengths, as well as the experimental results and conclusions in the main text. We will also add this part of the complete experimental results to the appendix of the paper and draw a result tables similar to **Table 17 (long-term time series forecasting results)** in the paper to show this part of the results.
> > >
> > > We are grateful for your support and believe these revisions will strengthen the paper's transparency and rigor. Your guidance is invaluable to our work.
> > >
> > > **Sincerely,**
> > >
> > > **The Authors**

---

> > > > ### Comment · Reviewer_Atv9 · 2025-08-05
> > > >
> > > > Dear Authors,
> > > >
> > > > Thank you for your responds and your commitment to strengthen the empirical results. I will raise my score as my points have been addressed.

---

> > > > > ### Author Response · Authors · 2025-08-05
> > > > >
> > > > > **Dear Reviewer,**
> > > > >
> > > > > Thank you for your support and valuable feedback throughout the review process. We sincerely appreciate your recognition of our revisions and your decision to raise the score—this means a great deal to us. We are grateful for the time and expertise you dedicated to improving our paper. Your insights have undoubtedly strengthened our work.
> > > > >
> > > > > **Sincerely,**
> > > > >
> > > > > **The Authors**

---

### Decision · Program_Chairs · 2025-09-17

**Decision:**

Accept (poster)

**Comment:**

This paper studies synthetic data generation for time series. They use symbolic expressions to develop a data generator, generate a dataset, then show that models pre-trained on synthetic data can rival models pre-trained on real data when finetuned and tested on real data. The paper was received very well by reviewers, who appreciated the approach is innovative and interesting and that it seems to work well and will hopefully inspire future works. The negatives appear to largely be clarifications that can be addressed in the writing, as noted by the reviewers who were receptive to the authors' responses. One key remaining drawback identified during the discussion period was that the performance evaluations are limited to the TimesNet datasets. These datasets have been so overstudied that improvements on them mean very little, especially the forecasting datasets. So extending the evaluations to include others (e.g., GIFT-Eval) will boost the impact of this work.